# Inference of Online Newton Methods with Nesterov's Accelerated Sketching

Haoxuan Wang [* 1]   Xinchen Du [* 1]   Sen Na [1]

## Abstract

Reliable decision-making with streaming data requires principled uncertainty quantification of online methods. While first-order methods enable efficient iterate updates, their inference procedures still require updating proper (covariance) matrices, incurring $O(d^2)$ time and memory complexity, and are sensitive to ill-conditioning and noise heterogeneity of the problem. This costly inference task offers an opportunity for more robust second-order methods, which are, however, bottlenecked by solving Newton systems with $O(d^3)$ complexity. In this paper, we address this gap by studying an online Newton method with Hessian averaging, where the Newton direction at each step is approximately computed using a *sketch-and-project solver with Nesterov's acceleration*, matching $O(d^2)$ complexity of first-order methods. For the proposed method, we quantify its uncertainty arising from both random data and randomized computation. Under standard smoothness and moment conditions, we establish global almost-sure convergence, prove asymptotic normality of the last iterate with a limiting covariance characterized by a Lyapunov equation, and develop a fully online covariance estimator with non-asymptotic convergence guarantees. We also connect the resulting uncertainty quantification to that of exact and sketched Newton methods without Nesterov's acceleration. Extensive experiments on regression models demonstrate the superiority of the proposed method for online inference.

## 1. Introduction

We consider the stochastic optimization problem:

$$\min_{\boldsymbol{x} \in \mathbb{R}^d} f(\boldsymbol{x}) = \mathbb{E}_{\xi \sim \mathcal{P}}[F(\boldsymbol{x}; \xi)], \qquad (1.1)$$

* Equal contribution. [1] School of Industrial and Systems Engineering, Georgia Institute of Technology. Correspondence to: Sen Na <senna@gatech.edu>.

*Proceedings of the 43rd International Conference on Machine Learning*, Seoul, South Korea. PMLR 306, 2026. Copyright 2026 by the author(s).

where $f : \mathbb{R}^d \to \mathbb{R}$ is a stochastic objective and $F(\cdot; \xi) : \mathbb{R}^d \to \mathbb{R}$ is its realization with a random sample $\xi \sim \mathcal{P}$. Problem (1.1) is simple but serves as a fundamental building block for many decision-making tasks in statistics and machine learning, such as online recommendation (Li et al., 2010), precision medicine (Kosorok & Laber, 2019), energy control (Wallace & Ziemba, 2005), and portfolio allocation (Fan et al., 2012; Chen et al., 2022). In these applications, (1.1) is often interpreted as a parameter estimation problem, where $\boldsymbol{x}$ denotes the model parameter, $\xi$ corresponds to a data sample, and the population minimizer $\boldsymbol{x}^\star = \operatorname{argmin}_{\boldsymbol{x} \in \mathbb{R}^d} f(\boldsymbol{x})$ represents the underlying true model parameter.

The ubiquity of streaming data in modern applications has made online methods particularly attractive, where only one *single* sample is used per step. One of the most fundamental online methods is the stochastic gradient descent (SGD) (Robbins & Monro, 1951; Kiefer & Wolfowitz, 1952):

$$\boldsymbol{x}_{t+1} = \boldsymbol{x}_t - \varphi_t \nabla F(\boldsymbol{x}_t; \xi_t). \qquad (1.2)$$

Numerous studies have explored both non-asymptotic and asymptotic convergence properties (e.g., $\mathbb{E}[\|\boldsymbol{x}_t - \boldsymbol{x}^\star\|^2] \lesssim \varphi_t$ and $\boldsymbol{x}_t \overset{a.s.}{\to} \boldsymbol{x}^\star$) of SGD and many accelerated variates (Robbins & Siegmund, 1971; Kushner & Huang, 1979; Pelletier, 1998; Bertsekas & Tsitsiklis, 2000; Moulines & Bach, 2011). These results show the consistency of SGD as an estimate of $\boldsymbol{x}^\star$. However, point estimation alone is insufficient for reliable decision-making; one must also quantify the uncertainty of the estimation procedure, e.g., via standard errors and confidence intervals. This need has inspired a recent series of studies on online uncertainty quantification and inference of SGD. In particular, Ruppert (1988); Polyak & Juditsky (1992) proposed averaging SGD iterates, $\bar{\boldsymbol{x}}_t = \sum_{i=1}^{t} \boldsymbol{x}_i / t$, and established the asymptotic normality of $\bar{\boldsymbol{x}}_t$. This result has since been extended to broader gradient-based methods (Toulis & Airoldi, 2017; Liang & Su, 2019; Duchi & Ruan, 2021; Davis et al., 2024). Given the asymptotic normality, online inference largely reduces to estimating the associated limiting covariance matrix, for example via plug-in or batch-means procedures (Chen et al., 2020; Zhu et al., 2021; Zhu & Dong, 2021; Singh et al., 2023; Jiang et al., 2025). There also exist alternative inference procedures that bypass explicit covariance estimation but are also less statistically efficient, such as bootstrapping (Fang et al., 2018; Liu et al., 2023; Zhong et al., 2023; Lam & Wang, 2026) and random scaling

(Li et al., 2021; Lee et al., 2022; Du et al., 2025).

While first-order methods (1.2) admit efficient iterate updates, their inference procedures still require updating proper (covariance) matrices, incurring $O(d^2)$ time and memory complexity. Moreover, SGD is delicate to ill-conditioning and noise heterogeneity of the problem. It has been largely observed that inference of SGD can suffer from clear under-coverage even for problems of small dimensions as $d = 20$ (Chen et al., 2020; Zhu et al., 2021; Kuang et al., 2025). This costly and significantly challenging online inference task offers a nice opportunity to apply higher-order (quasi-)Newton methods, which exploit the objective curvature information via the update (Bottou et al., 2018; Byrd et al., 2016):

$$\boldsymbol{x}_{t+1} = \boldsymbol{x}_t + \varphi_t \Delta \boldsymbol{x}_t, \text{ with } B_t \Delta \boldsymbol{x}_t = -\nabla F(\boldsymbol{x}_t; \xi_t) \quad (1.3)$$

where $B_t \approx \nabla^2 f(\boldsymbol{x}_t)$ is an estimate or approximation of the objective Hessian. There has been a growing interest in quantifying the uncertainty of (1.3). Leluc & Portier (2023) treated $B_t^{-1}$ as a preconditioning matrix and established the asymptotic normality of the last iterate $\boldsymbol{x}_t$ under the convergence of $B_t$. They further showed that $\boldsymbol{x}_t$ attains *optimal statistical efficiency* when $B_t \to \nabla^2 f(\boldsymbol{x}^\star)$, corresponding to online Newton methods. Cénac et al. (2020); Boyer & Godichon-Baggioni (2023); Cénac et al. (2025); Godichon-Baggioni & Werge (2025) specialized (1.3) to regression problems and derived the asymptotic normality for the averaged Newton iterate $\bar{\boldsymbol{x}}_t$, and Na (2025); Gao et al. (2025) extended (1.3) to constrained problems.

Suppose the Hessian estimate is accessible (otherwise, it can be approximated by first- or zeroth-order information via quasi-Newton or finite-difference schemes (Dennis & Moré, 1977; Spall, 2000)). The major concern of (1.3) lies in solving the Newton system, which is prohibitively expensive with an $O(d^3)$ time complexity. As such, Na & Mahoney (2025) introduced an online sketched Newton method that approximately solves the Newton system via a *sketch-and-project* solver, thereby reducing the time complexity to $O(d^2)$. The authors quantified the uncertainty of sketched Newton method by establishing the last-iterate asymptotic normality. To bridge the gap between uncertainty quantification and practical inference, Kuang et al. (2025) further proposed an online, consistent limiting covariance matrix estimator that is constructed solely from sketched Newton iterates, without inverting any matrices. Together, these two works are the first attempts to make Newton methods practically viable for online inference tasks, achieving robust outperformance over SGD.

On the other hand, recent advances in randomized linear solvers have shown that the sketch-and-project solver used in aforementioned works, originating from Strohmer & Vershynin (2008); Gower & Richtárik (2015); Pilanci & Wainwright (2016; 2017); Dereziński & Rebrova (2024), can be further sped up via **Nesterov's acceleration**, without increasing

the order of per-iteration cost (Gower et al., 2018; Dereziński et al., 2025). Particularly, Dereziński et al. (2025) showed that the expected squared solution approximation error decays exponentially at the rate $1 - \mu_t$ for unaccelerated methods, while at $1 - \sqrt{\mu_t/\nu_t}$ for accelerated methods. Here, $\mu_t, \nu_t$ are problem parameters satisfying $1 \le \nu_t \le 1/\mu_t$ (that is, $\mu_t \le \sqrt{\mu_t/\nu_t} \le \sqrt{\mu_t}$). See (2.7) for details. Consequently, for the extreme case of $\nu_t = 1$, accelerated sketching improves the rate of the sketch-and-project solver from $1 - \mu_t$ to $1 - \sqrt{\mu_t}$, mirroring the effect of Nesterov's acceleration in SGD. This improved computational efficiency motivates a natural but largely unexplored question:

*How does accelerated sketching affect statistical efficiency of inference in online sketched Newton methods?*

In this work, we answer this question by establishing global almost-sure convergence and local asymptotic normality of online Newton methods with Nesterov's accelerated sketching. In contrast to the above sketching literature that focuses on computational efficiency of sketching solver, we focus on how accelerated sketching affects uncertainty quantification and inference of online Newton methods. Specifically, we quantify the uncertainty of (1.3) from both random data and accelerated sketching. Under standard moment conditions on the estimation noise, we first show that

$$1/\sqrt{\varphi_t} \cdot (\boldsymbol{x}_t - \boldsymbol{x}^\star) \xrightarrow{d} \mathcal{N}(\boldsymbol{0}, \Sigma^\star), \quad (1.4)$$

where the limiting covariance $\Sigma^\star$ is characterized by the solution to a Lyapunov equation that depends explicitly on the sketching distribution (cf. (4.2)). This characterization discloses a *computational-statistical trade-off* induced by accelerated sketching solver, with two special scenarios:

**(a)** When accelerated sketching is completely discarded, our results reduce to those of exact Newton method (Leluc & Portier, 2023), and $\Sigma^\star$ recovers the covariance of averaged SGD estimator (Polyak & Juditsky, 1992), which is known to be *statistically minimax optimal* (Davis et al., 2024).

**(b)** When accelerated sketching does not yield a provable accelerated rate (i.e., $\mu_t \nu_t = 1$), $\Sigma^\star$ recovers the covariance of the unaccelerated sketched Newton method (Kuang et al., 2025). Notably, no provable acceleration is much more general than specific unaccelerated sketching scheme used in that work; in particular, momentum may still be employed but does not offer an improved convergence rate.

To carry out practical inference based on normality (1.4), we further develop a fully online estimator of the covariance $\Sigma^\star$ and establish its consistency. The asymptotic normality together with consistent covariance estimation fully resolve the inference task for online sketched Newton methods with Nesterov's acceleration. We demonstrate theoretical findings on broad regression problems, illustrating the practicality and advantages of the proposed inference procedure.

**Technical challenges.** Our inference of online Newton with Nesterov's accelerated sketching overcomes several technical challenges building on existing works. **First**, covariance estimation relies on the non-asymptotic convergence rate for the fourth moment of the Newton iterate error $\mathbb{E}[\|\boldsymbol{x}_t - \boldsymbol{x}^\star\|^4]$ (cf. Lemma 4.5), which is a higher-order guarantee than commonly studied second-moment bound $\mathbb{E}[\|\boldsymbol{x}_t - \boldsymbol{x}^\star\|^2]$ (Leluc & Portier, 2023; Boyer & Godichon-Baggioni, 2023; Cénac et al., 2025; Godichon-Baggioni & Werge, 2025). **Second**, our method requires quantifying uncertainty from both data sampling and randomized linear solver, unlike existing inference methods in which randomness comes solely from data sampling (Chen et al., 2020; Zhu et al., 2021; Davis et al., 2024; Jiang et al., 2025). The sketching solver is executed for a fixed number of steps, so its algorithmic randomness does not vanish asymptotically and thus inevitably affects the limiting distribution (see (4.2)). **Third**, prior work Kuang et al. (2025); Na & Mahoney (2025) studied unaccelerated sketching, while we extend their analysis to Nesterov's accelerated sketching. The extension is highly technically involved:

**(a)** Nesterov's acceleration introduces an additional sequence of momentum co-states; thus, the solver evolves as a $2d$-dimensional state-co-state recursion, rather than an independent, projection-based state update as in standard sketch-and-project methods (Dereziński & Rebrova, 2024, (1.1)).

**(b)** Accelerated sketching involves a random, time-varying, asymmetric $2 \times 2$ block matrix acting as a linear operator (cf. (3.1)), while without acceleration, it reduces to a symmetric projection matrix in the $(1,1)$ block. Accordingly, many key properties no longer hold under acceleration. For example, while contraction of the $(1,1)$ block can be established in unaccelerated case due to favorable boundedness of projection matrices, this property fails for the full asymmetric operator. As such, we leverage the Cayley–Hamilton theorem and tools from the theory of similar matrices and Kronecker products to derive a recursion for the spectral radius and analyze the contraction of the $(1,1) + (1,2)$ block, corresponding to marginal state evolution (Lemmas 3.6, 3.7).

**(c)** Acceleration introduces conditionally deterministic, but random parameters $(\alpha_t, \beta_t, \gamma_t)$ governing the stepsize and momentum, induced by the linear operator in **(b)**. In contrast, these parameters are deterministic in unaccelerated settings. To show asymptotic normality, we have to explore their *non-asymptotic* convergence rates (e.g., $|\alpha_t - \alpha^\star| = O_p(\sqrt{\varphi_t})$), ensuring that the randomness of these parameters leads to only higher-order terms relative to the randomness from data sampling and sketching (Lemma 4.2). In other words, when $t$ is large, the sketching solver at iteration $t$ can be viewed as operating at $\boldsymbol{x}^\star$ with optimal parameters.

To our knowledge, all above technical challenges and our resolutions do not appear in existing works on either sketched Newton methods or online inference.

**Theoretical roadmap.** Section 3 first develops contraction guarantees for the accelerated sketching solver (Lemmas 3.6–3.7) and establishes global almost-sure convergence of the online Newton method (Theorem 3.8). Section 4.1 then proves convergence of the Hessian average, acceleration parameters, and sketching operator (Lemmas 4.1–4.2), leading to last-iterate asymptotic normality (Theorem 4.3). Finally, Section 4.2 establishes fourth-moment bounds (Lemma 4.5) and consistency of the online covariance estimator (Theorem 4.6), which enables practical statistical inference.

**Notation.** We let $\|\cdot\|$ denote the $\ell_2$ norm for vectors and spectral norm for matrices. For two sequences $\{a_t, b_t\}$, $a_t = O(b_t)$ implies that $|a_t| \leq c|b_t|$ for sufficient large $t$ and some positive constant $c$; $a_t = o(b_t)$ implies that $|a_t/b_t| \to 0$ as $t \to \infty$. We also denote $O_p, o_p$ the order in probability. We let $I$ denote the identity matrix, $\mathbf{0}$ denote the zero vector or matrix, and $\mathbf{1}$ denote the all-one vector; their dimensions are clear from the context. For a series of compatible matrices $\{A_i\}$, $\prod_{k=i}^{j} A_k = A_j A_{j-1} \cdots A_i$ if $j \geq i$ and $I$ if $j < i$. For two symmetric matrices $A$ and $B$, $A \succeq B$ means $A - B$ is positive semidefinite; and $\lambda_{\min}(A)$ ($\lambda_{\max}(A)$) denotes the smallest (largest) eigenvalue $A$. We also denote $f_t = f(\boldsymbol{x}_t)$ and $f^\star = f(\boldsymbol{x}^\star)$ (similar for $\nabla f_t, \nabla^2 f_t$, etc.).

## 2. Online Newton with Accelerated Sketching

We introduce online Newton method with Nesterov's accelerated sketching. The method simply adopts the update (1.3) with $B_t$ defined as the averaged Hessian estimate. Instead of solving the Newton system of (1.3) exactly, we employ an accelerated sketching solver to reduce time complexity.

### 2.1. Online Newton with Hessian averaging

At each outer iteration $t$, we draw a sample $\xi_t \sim \mathcal{P}$ and compute the gradient and Hessian estimates $g_t = \nabla F(\boldsymbol{x}_t; \xi_t)$ and $H_t = \nabla^2 F(\boldsymbol{x}_t; \xi_t)$. As standard in studies of Newton methods, we suppose the Hessian estimate $H_t$ is accessible; otherwise, it can be approximated by quasi-Newton or finite-difference schemes, and our analysis extends to those settings with proper adjustments (Leluc & Portier, 2023; Na, 2025). We define $B_t := \frac{1}{t} \sum_{i=0}^{t-1} H_i$ as the Hessian average based on samples $\{\xi_i\}_{i=0}^{t-1}$, admitting an online update:

$$B_t = (1 - 1/t) B_{t-1} + H_{t-1}/t. \qquad (2.1)$$

Inspired by the techniques of online variance reduction, Hessian averaging has been widely used in stochastic Newton methods to accelerate convergence rates (Bercu et al., 2020; Na et al., 2022; Leluc & Portier, 2023; Boyer & Godichon-Baggioni, 2023; Na & Mahoney, 2025).

With $B_t$ in (2.1), we then aim to solve the Newton system:

$$B_t \Delta \boldsymbol{x}_t = -g_t. \qquad (2.2)$$

**Algorithm 1** Sketch-and-Project with Nesterov's Acceleration: $\texttt{NASketch}(B, -g; \alpha, \beta, \gamma, \tau)$

---
1: **Target:** $B\Delta\boldsymbol{x} = -g$;
2: **Initialize:** set initial values $\boldsymbol{z}_0 = \boldsymbol{v}_0 = \boldsymbol{0}$;
3: **for** $j = 0, 1, \ldots \tau - 1$ **do**
4:    $\boldsymbol{y}_j = \alpha\boldsymbol{v}_j + (1-\alpha)\boldsymbol{z}_j$;
5:    Generate a sketching vector/matrix $S_j \sim S \in \mathbb{R}^{d\times s}$;
6:    $\boldsymbol{\omega}_j = BS_j(S_j^\top B^2 S_j)^\dagger S_j^\top(B\boldsymbol{y}_j + g)$;
7:    $\boldsymbol{z}_{j+1} = \boldsymbol{y}_j - \boldsymbol{\omega}_j$;
8:    $\boldsymbol{v}_{j+1} = \beta\boldsymbol{v}_j + (1-\beta)\boldsymbol{y}_j - \gamma\boldsymbol{\omega}_j$;
9: **end for**
10: **Output:** $\boldsymbol{z}_\tau$.

---

In general, solving (2.2) costs $O(d^3)$ complexity, making the method impractical. We compute $\Delta\boldsymbol{x}_t$ approximately in $O(d^2)$ via Nesterov's accelerated sketching.

### 2.2. Sketch-and-Project with Nesterov's acceleration

We now focus on (2.2). To ease notation, we drop the outer iteration index $t$ in this subsection and introduce the accelerated sketching solver $\texttt{NASketch}(B, -g; \alpha, \beta, \gamma, \tau)$, as summarized in Algorithm 1. Here, $(B, -g)$ denote the left-hand side matrix and right-hand side vector of linear system, respectively; $(\alpha, \beta, \gamma)$ are solver parameters related to the momentum stepsize; and $\tau$ is the number of sketching steps. The solver will output an approximation of $\Delta\boldsymbol{x}$. We refer a detailed introduction of the solver to Gower et al. (2018); Dereziński et al. (2025).

For each (inner) sketching iteration $j$ (at fixed outer iteration $t$), we draw a sketching vector/matrix $S_j \sim S \in \mathbb{R}^{d\times s}$ with $s \ll d$. Without acceleration, the vanilla sketch-and-project method admits the update:

$$\boldsymbol{z}_{j+1} = \boldsymbol{z}_j - BS_j(S_j^\top B^2 S_j)^\dagger S_j^\top(B\boldsymbol{z}_j + g), \quad (2.3)$$

where $\dagger$ denotes the Moore-Penrose pseudoinverse; $\boldsymbol{z}_j$ is the current state; and $BS_j(S_j^\top B^2 S_j)^\dagger S_j^\top(B\boldsymbol{z}_j + g)$ is the sketching direction, based on the observation that the above update is equivalent to solving a sketch-and-project problem:

$$\boldsymbol{z}_{j+1} = \arg\min \|\boldsymbol{z} - \boldsymbol{z}_j\|^2 \quad \text{s.t.} \quad S_j^\top B\boldsymbol{z} = -S_j^\top g.$$

With acceleration, the method involves an additional momentum co-state $\boldsymbol{v}_j$. Given the state-co-state pair $(\boldsymbol{z}_j, \boldsymbol{v}_j)$, we first compute the sketching direction at their midpoint $\boldsymbol{y}_j = \alpha\boldsymbol{v}_j + (1-\alpha)\boldsymbol{z}_j$ as

$$\boldsymbol{w}_j = BS_j(S_j^\top B^2 S_j)^\dagger S_j^\top(B\boldsymbol{y}_j + g).$$

Then, the pair $(\boldsymbol{z}_j, \boldsymbol{v}_j)$ is updated as

$$\begin{cases} \boldsymbol{z}_{j+1} = \boldsymbol{y}_j - \boldsymbol{\omega}_j, \\ \boldsymbol{v}_{j+1} = \beta\boldsymbol{v}_j + (1-\beta)\boldsymbol{y}_j - \gamma\boldsymbol{\omega}_j. \end{cases}$$

Clearly, by setting $\alpha = 0.5$, $\beta = 0$, $\gamma = 1$, accelerated sketching reduces to unaccelerated sketching (2.3).

**Parameter setup and accelerated rate.** Gower et al. (2018); Dereziński et al. (2025) proposed controlling $(\alpha, \beta, \gamma)$ by two problem parameters $(\mu, \nu)$ as

$$\alpha = 1/(1+\gamma\nu), \quad \beta = 1 - \sqrt{\mu/\nu}, \quad \gamma = 1/\sqrt{\mu\nu}, \quad (2.4)$$

where $(\mu, \nu)$ are defined as (or lower, upper bounds of)

$$\mu = \inf_{\boldsymbol{a}\neq\boldsymbol{0}} \frac{\boldsymbol{a}^\top Z\boldsymbol{a}}{\boldsymbol{a}^\top \boldsymbol{a}}, \qquad \nu = \sup_{\boldsymbol{a}\neq\boldsymbol{0}} \frac{\boldsymbol{a}^\top \mathbb{E}[\widetilde{Z}Z^{-1}\widetilde{Z}]\boldsymbol{a}}{\boldsymbol{a}^\top Z\boldsymbol{a}}, \quad (2.5)$$

with $\widetilde{Z} := BS(S^\top B^2 S)^\dagger S^\top B$ and $Z := \mathbb{E}[\widetilde{Z}] \in \mathbb{R}^{d\times d}$ being a projection matrix and its expectation, taken over the randomness of sketching only. (Recall that we drop the outer iteration index $t$; thus, the expectation is conditional on $\boldsymbol{x}_t$ and sample $\xi_t$). Gower et al. (2018, Lemma 2) shows:

$$1 \leq \nu \leq 1/\mu = \|Z^{-1}\|, \quad (2.6)$$

implying that $\alpha \in (0, 1)$, $\beta \in [0, 1)$, $\gamma \geq 1$.

In the online Newton method, these quantities are evaluated at the current Hessian estimate $B_t$ and therefore become $(\mu_t, \nu_t)$. Exact computation can be expensive, since it involves the sketching expectation defining $Z_t$ and spectral quantities of $Z_t$. Thus, (2.5) should be viewed as the theoretical parameterization under which the solver-level rate below is stated; in implementations, one may use computable approximations, periodically refreshed values, or fixed choices. The asymptotic normality and covariance estimation results in Section 4 rely on the convergence of the induced acceleration parameters and the resulting sketching operator, rather than on exact computation of $(\mu_t, \nu_t)$ at every iteration.

Under this parameterization, Dereziński et al. (2025, Eq. (1)) showed that accelerated sketching enjoys an exponential rate:

$$\mathbb{E}\|\boldsymbol{z}_j - \boldsymbol{z}\|^2 \leq 2(1 - \sqrt{\mu/\nu})^j \|\boldsymbol{z}_0 - \boldsymbol{z}\|^2, \quad (2.7)$$

where $\boldsymbol{z} = \Delta\boldsymbol{x} = -B^{-1}g$ is the exact solution and the expectation is again taken over randomness of sketching only. The rate $1 - \sqrt{\mu/\nu}$ in (2.7) improves the rate $1 - \mu$ of unaccelerated sketching (Gower & Richtárik, 2015). We note that the provable acceleration degrades if and only if $\gamma = 1$, i.e., $\mu\nu = 1$, which is substantially more general than the setting of unaccelerated sketching scheme (2.3) (momentum may be employed but does not offer an improved rate).

With the above sketching solver, online Newton performs:

$$\boldsymbol{x}_{t+1} = \boldsymbol{x}_t + \varphi_t \cdot \texttt{NASketch}(B_t, -g_t; \alpha_t, \beta_t, \gamma_t, \tau), \quad (2.8)$$

where $\varphi_t = C_\varphi/(t+1)^\varphi$ is a vanishing stepsize sequence.

# 3. Assumptions and Global Convergence

In this section, we introduce assumptions and prove global almost-sure convergence of online Newton with Nesterov's accelerated sketching. Unlike aforementioned sketching literature that focuses on the solver-level convergence rate in (2.7), we study the convergence of outer Newton iterates $\boldsymbol{x}_t$, and more importantly, their uncertainty quantification and inference in Section 4. This focus shift requires significantly different guarantees and analytical techniques even for Algorithm 1: Dereziński et al. (2025) uses Gaussian universality to derive a sharp bound on $\nu$, while we abstract Nesterov's acceleration via a random, asymmetric linear operator that is closely tied to the limiting covariance matrix $\Sigma^\star$ in (1.4). To analyze that operator, we employ the Cayley-Hamilton theorem along with tools from similar matrices and Kronecker products to derive a recursion for its spectral radius.

## 3.1. Assumptions

We define the filtration $\mathcal{F}_{t-0.5} = \sigma\left(\{\xi_i, \{S_{i,j}\}_j\}_{i=0}^{t-1} \cup \xi_t\right)$ that contains all the randomness of $\boldsymbol{x}_{0:t}$ and $\xi_t$; and $\mathcal{F}_t = \sigma\left(\{\xi_i, \{S_{i,j}\}_j\}_{i=0}^{t}\right)$ containing the randomness of $\boldsymbol{x}_{0:(t+1)}$. The first assumption regards the strong convexity and Lipschitz smoothness of the objective.

**Assumption 3.1.** We assume that $f$ and realizations $F(\cdot; \xi)$ are twice continuously differentiable such that for any $\boldsymbol{x}$, $\gamma_H \leq \lambda_{\min}\left(\nabla^2 F(\boldsymbol{x}; \xi)\right) \leq \lambda_{\max}\left(\nabla^2 F(\boldsymbol{x}; \xi)\right) \leq \Upsilon_H$ for some constants $\Upsilon_H > \gamma_H > 0$. We also assume $\nabla^2 f$ is $\Upsilon_L$-Lipschitz continuous, i.e., $\|\nabla^2 f(\boldsymbol{x}) - \nabla^2 f(\boldsymbol{x}')\| \leq \Upsilon_L \|\boldsymbol{x} - \boldsymbol{x}'\|$ for any $\boldsymbol{x}, \boldsymbol{x}'$.

Assumption 3.1 is standard in existing inference literature (Chen et al., 2020; Zhu et al., 2021; Kuang et al., 2025). The next assumption imposes a growth condition on the moment of gradient noise, which is also standard in those works.

**Assumption 3.2.** For any $t \geq 0$, we assume the gradient estimate is unbiased $\mathbb{E}[g_t \mid \mathcal{F}_{t-1}] = \nabla f_t$ and satisfies a growth condition on its $q_g$-th moment: for some $C_{g,1}, C_{g,2} > 0$,

$$\mathbb{E}[\|g_t - \nabla f_t\|^{q_g} \mid \mathcal{F}_{t-1}] \leq C_{g,1}\|\boldsymbol{x}_t - \boldsymbol{x}^\star\|^{q_g} + C_{g,2}.$$

We will specify $q_g$ in theorem statements. In particular, $q_g = 2$ suffices for global convergence, $q_g > 2$ for asymptotic normality, and $q_g = 4$ for online inference; all moment conditions are standard in aforementioned literature. Similarly, we present moment conditions on the Hessian noise.

**Assumption 3.3.** For any $t \geq 0$, we assume the Hessian estimate is unbiased $\mathbb{E}[H_t \mid \mathcal{F}_{t-1}] = \nabla^2 f_t$ and satisfies a growth condition on its $q_H$-th moment: for some $C_{H,1}, C_{H,2} > 0$,

$$\mathbb{E}[\|H_t - \nabla^2 f_t\|^{q_H} \mid \mathcal{F}_{t-1}] \leq C_{H,1}\|\boldsymbol{x}_t - \boldsymbol{x}^\star\|^{q_H} + C_{H,2}.$$

Similarly, $q_H$ is specified as follows: $q_H = 0$ for global almost-sure convergence, $q_H = 2$ for asymptotic normality,

and $q_H = 4$ for online covariance estimation. Assumptions 3.2 and 3.3 are standard in online Newton inference (Na & Mahoney, 2025; Kuang et al., 2025). For linear and logistic regressions, they can be verified by the arguments in Chen et al. (2020, Appendix A); in particular, $q_H = 4$ holds when the design covariates have bounded eighth moments. We next impose an assumption on the sketching distribution.

**Assumption 3.4.** For any $t \geq 0$, we assume $\{S_{t,j}\}_{j=0}^{\tau-1} \overset{iid}{\sim} S$ satisfying $Z_t := \mathbb{E}\left[B_t S(S^\top B_t^2 S)^\dagger S^\top B_t \mid \mathcal{F}_{t-1}\right] \succeq \gamma_S I$ for some $\gamma_S > 0$, and $\{S_{t,j}\}_{j=0}^{\tau-1}$ are independent of data sample $\xi_t$. Further, we assume the moment of the condition number of $S$ is finite: $\mathbb{E}[(\|S\|\|S^\dagger\|)^{q_S}] \leq \Upsilon_S$ for $\Upsilon_S > 0$.

As analyzed in Na & Mahoney (2025); Kuang et al. (2025), $q_S = 1$ suffices for asymptotic normality and $q_S = 2$ for inference. Note that the above expectations are taken over the randomness of the sketching matrix $S$. The lower bound on expected projection matrix $Z_t$ is commonly required to ensure convergence of sketching solvers, both with and without acceleration (Gower & Richtárik, 2015; Gower et al., 2018; Dereziński et al., 2025). All conditions in Assumption 3.4 easily hold for various sketching distributions, including Gaussian sketching $S \sim \mathcal{N}(\boldsymbol{0}, \Sigma)$ and Kaczmarz sketching $S \sim \mathrm{Unif}(\{\boldsymbol{e}_i\}_{i=1}^d)$ (Strohmer & Vershynin, 2008).

## 3.2. Guarantees of accelerated sketching

We state some results of Algorithm 1. Note from Section 2.2 that, for each $t$, we draw $\tau$ sketching matrices $\{S_{t,j}\}_{j=0}^{\tau-1} \sim S$. Define $\widetilde{Z}_{t,j} = B_t S_{t,j}(S_{t,j}^\top B_t^2 S_{t,j})^\dagger S_{t,j}^\top B_t \in \mathbb{R}^{d \times d}$ and denote their shared conditional expectation, taken over the randomness of sketching, by $Z_t = \mathbb{E}[\widetilde{Z}_{t,j}|\mathcal{F}_{t-1}]$. We also define parameters $(\alpha_t, \beta_t, \gamma_t, \mu_t, \nu_t)$ as in (2.4)-(2.5). Now, let us construct an asymmetric $2 \times 2$ block matrix

$$\widetilde{C}_{t,j} = \begin{pmatrix} 1 - \alpha_t & \alpha_t \\ (1-\alpha_t)(1-\beta_t) & \alpha_t + \beta_t - \alpha_t\beta_t \end{pmatrix} \otimes I_d$$
$$- \begin{pmatrix} 1 - \alpha_t & \alpha_t \\ (1-\alpha_t)\gamma_t & \alpha_t\gamma_t \end{pmatrix} \otimes \widetilde{Z}_{t,j} \in \mathbb{R}^{2d \times 2d}, (3.1)$$

where $\otimes$ is the Kronecker product. We will show in Lemma 3.5 that $\widetilde{C}_{t,j}$ captures the transition of state-co-state sequence $\{(\boldsymbol{z}_{t,j}, \boldsymbol{v}_{t,j})\}_j$ of Algorithm 1: for $t \geq 0, 0 \leq j \leq \tau - 1$,

$$(\boldsymbol{z}_{t,j+1} - \Delta\boldsymbol{x}_t, \boldsymbol{v}_{t,j+1} - \Delta\boldsymbol{x}_t) = \widetilde{C}_{t,j}(\boldsymbol{z}_{t,j} - \Delta\boldsymbol{x}_t, \boldsymbol{v}_{t,j} - \Delta\boldsymbol{x}_t).$$

Naturally, the product $\widetilde{C}_t := \prod_{j=0}^{\tau-1} \widetilde{C}_{t,j} \in \mathbb{R}^{2d \times 2d}$ denotes the transition operator mapping $(\boldsymbol{z}_{t,0}, \boldsymbol{v}_{t,0})$ to $(\boldsymbol{z}_{t,\tau}, \boldsymbol{v}_{t,\tau})$. We use $[\widetilde{C}_t]_{k,l}$ to denote its $(k, l)$-th block, and let

$$\widetilde{K}_t := [\widetilde{C}_t]_{1,1} + [\widetilde{C}_t]_{1,2} = \begin{pmatrix} I & \boldsymbol{0} \end{pmatrix} \widetilde{C}_t \begin{pmatrix} I \\ I \end{pmatrix} \in \mathbb{R}^{d \times d} (3.2)$$

be the matrix governing the evolution of marginal state $\boldsymbol{z}_{t,j}$. To analyze the expected error of Algorithm 1, we further define $C_t = \mathbb{E}[\widetilde{C}_t|\mathcal{F}_{t-1}]$ and $K_t = \mathbb{E}[\widetilde{K}_t|\mathcal{F}_{t-1}]$.

The first lemma makes the above statement rigorous.

**Lemma 3.5.** *For any $t \geq 0$, the construction of $\widetilde{K}_t$ in (3.2) admits the representation*

$$\boldsymbol{z}_{t,\tau} - \Delta\boldsymbol{x}_t = \widetilde{K}_t(\boldsymbol{z}_{t,0} - \Delta\boldsymbol{x}_t) \quad with \quad \Delta\boldsymbol{x}_t = -B_t^{-1}g_t.$$

*If $\mathbb{E}[g_t \mid \mathcal{F}_{t-1}] = \nabla f_t$, the conditional expectation of $\boldsymbol{z}_{t,\tau}$ is*

$$\mathbb{E}[\boldsymbol{z}_{t,\tau} \mid \mathcal{F}_{t-1}] = -(I - K_t)B_t^{-1}\nabla f_t.$$

*Thus, the residual $\boldsymbol{z}_{t,\tau} - (I - K_t)\Delta\boldsymbol{x}_t$ form a martingale difference sequence with respect to filtration $\{\mathcal{F}_t\}_{t\geq 0}$. Finally, if $\gamma_t = 1$, $\widetilde{K}_t$ has a simple form of $\widetilde{K}_t = \prod_{j=0}^{\tau-1}(I - \widetilde{Z}_{t,j})$.*

To control the $\tau$-step contraction of accelerated sketching solver, we analyze its associated asymmetric linear operator $C_t$ and marginal operator $K_t$. We observe that the accelerated sketching dynamics defined by $C_t, K_t$ decouple along eigendirections of $Z_t$ and reduce to a family of matrices:

$$G_t(z) := \begin{pmatrix} 1 - \alpha_t & \alpha_t \\ (1-\alpha_t)(1-\beta_t) & \alpha_t + \beta_t - \alpha_t\beta_t \end{pmatrix}$$
$$- \begin{pmatrix} 1 - \alpha_t & \alpha_t \\ (1-\alpha_t)\gamma_t & \alpha_t\gamma_t \end{pmatrix} z. \quad (3.3)$$

The following lemma makes the corresponding *spectral radius recursion* explicit via Cayley–Hamilton theorem.

**Lemma 3.6.** *Let $p_{t,\tau}(z) = \boldsymbol{e}_1^\top G_t(z)^\tau \mathbf{1}$ for a fixed $t \geq 0$ and $z \in [\mu_t, 1]$. Then, $p_{t,\tau}(z)$ satisfies the second-order recursion: for all $\tau \geq 2$,*

$$p_{t,\tau}(z) = Tr_t(z)\, p_{t,\tau-1}(z) - D_t(z)\, p_{t,\tau-2}(z), \quad (3.4)$$

*with initialization $p_{t,0}(z) = 1$ and $p_{t,1}(z) = 1 - z$, where $Tr_t(z) = \mathrm{trace}(G_t(z))$ and $D_t(z) = \det(G_t(z))$. Moreover, under Assumption 3.4, the spectral radius of $G_t(z)$, denoted as $\rho(G_t(z))$, is uniformly contractive:*

$$\sup_{z\in[\mu_t,1]} \rho(G_t(z)) \leq 1 - \sqrt{\mu_t/\nu_t} \leq 1 - \gamma_S =: \rho_\star < 1. \quad (3.5)$$

*Combining the recursion (3.4) with the contractive spectral radius bound (3.5) yields*

$$\sup_{z\in[\mu_t,1]} |p_{t,\tau}(z)| \leq 2\tau\, (1 - \sqrt{\mu_t/\nu_t})^{\tau-2}. \quad (3.6)$$

The quantity $p_{t,\tau}(z) = \boldsymbol{e}_1^\top G_t(z)^\tau \mathbf{1}$ measures the magnitude of $K_t$ along an eigendirection of $Z_t$ with eigenvalue $z$; thus, bounding $\sup_{z\in[\mu_t,1]} |p_{t,\tau}(z)|$ leads to a direct control of $\|K_t\|$, where the range of $z$ is ensured by Assumption 3.4. The recursion (3.4) and contraction (3.5) provide a tractable way of deriving the bound of $p_{t,\tau}(z)$ even if $G_t(z)$ is asymmetric; and the explicit bound in (3.6) quantifies how $\tau$ exponentially controls $\|K_t\|$, as stated in the next lemma.

**Lemma 3.7.** *Under Assumption 3.4, for any $t \geq 0$, we have*

$$\|K_t\| \leq 2\tau(1 - \sqrt{\mu_t/\nu_t})^{\tau-2}.$$

When $\gamma_t = 1$, Lemma 3.5 shows that the marginal operator $\widetilde{K}_t$ reduces to a product of $\tau$ projection matrices. Lemma 3.7 further shows that the exponential rate $1 - \sqrt{\mu_t/\nu_t}$ reduces to $1 - \mu_t$, which coincides with the rate for unaccelerated sketching (Gower & Richtárik, 2015). The constant factor $\tau$ in (3.6) and Lemma 3.7 is negligible; the overall behavior of $K_t$ is fully dominated by the exponential decay of $\tau$.

### 3.3. Global almost-sure convergence

With the results of accelerated sketching, we now establish global almost-sure convergence of the outer, online Newton iterates with Nesterov's accelerated sketching.

**Theorem 3.8.** *Consider the Newton scheme (2.8) with the stepsize $\varphi_t = C_\varphi/(t+1)^\varphi$ satisfying $C_\varphi > 0$, $\varphi \in (0.5, 1]$. Suppose Assumptions 3.1–3.4 hold with $q_g = 2$, $q_H = q_S = 0$, and $\tau$ satisfies $\tau(1 - \sqrt{\mu_t/\nu_t})^{\tau-2} \leq \gamma_H/(4\Upsilon_H)$ for all $t \geq 0$. Then, we have $\boldsymbol{x}_t \stackrel{a.s.}{\to} \boldsymbol{x}^\star$ as $t \to \infty$.*

We note that global convergence only requires gradient estimates to have bounded variance ($q_g = 2$), which is standard in the literature (Ghadimi & Lan, 2013). No moment conditions are needed for Hessian estimates or the condition number of the sketching distribution ($q_H = q_S = 0$). The condition on $\tau$ can be indeed satisfied uniformly by invoking (3.5); specifically, $\tau\rho_\star^{\tau-2} \leq \gamma_H/(4\Upsilon_H)$. This condition ensures that, at each step, the approximate Newton direction from accelerated sketching solver is not too far from the exact Newton direction, at least in an average sense.

## 4. Uncertainty Quantification and Inference

We quantify uncertainties of online Newton with Nesterov's accelerated sketching from both random sampling and randomized computation. We first establish asymptotic normality with a limiting covariance given by a Lyapunov equation depending on the solver operator. We then propose an online, consistent covariance estimator for practical inference goals.

### 4.1. Asymptotic rate and normality

We begin by showing the almost-sure convergence of the Hessian average $B_t$, the acceleration parameters $(\alpha_t, \beta_t, \gamma_t)$, and, more importantly, the marginal sketching operator $K_t$. Let $B^\star := \nabla^2 f(\boldsymbol{x}^\star)$ and define $(\alpha^\star, \beta^\star, \gamma^\star)$ as in (2.4) with $(\mu^\star, \nu^\star)$ in (2.5) evaluated at $B^\star$. Then, we let $K^\star := \mathbb{E}[\widetilde{K}^\star]$, where $\widetilde{K}^\star$ is defined analogously to $\widetilde{K}_t$ in (3.2) while evaluated at $(\alpha^\star, \beta^\star, \gamma^\star)$ and $B^\star$.

**Lemma 4.1.** *Under the conditions of Theorem 3.8 but strengthening $q_S = 0$ to $q_S = 1$, then we have $B_t \stackrel{a.s.}{\to} B^\star$, $(\alpha_t, \beta_t, \gamma_t) \stackrel{a.s.}{\to} (\alpha^\star, \beta^\star, \gamma^\star)$, and $K_t \stackrel{a.s.}{\to} K^\star$.*

We require finite expected condition number of $S$ since the definition of $(\alpha_t, \beta_t, \gamma_t)$ relies on $(\mu_t, \nu_t)$, involving the expected condition number. The almost-sure convergence of $K_t$ to $K^\star$, together with Lemma 3.7, ensures that our limiting covariance established in (4.2) is well defined, provided the number of sketching steps $\tau$ is chosen properly. The following lemma further provides the convergence rates.

**Lemma 4.2.** *Suppose Assumptions 3.1–3.4 hold with $q_g > 2$, $q_H = 2$, $q_S = 1$; $\tau$ satisfies $\tau(1 - \sqrt{\mu_t/\nu_t})^{\tau-2} \leq \gamma_H/(4\Upsilon_H)$ for all $t \geq 0$; and the stepsize parameters satisfy $\varphi \in (0.5, 1]$ and $C_\varphi > 0.5/\lambda_{\min}(I - K^\star)$ if $\varphi = 1$. Then, we have*

$$\max \left\{ \|B_t - B^\star\|, |\alpha_t - \alpha^\star|, |\beta_t - \beta^\star|, \right.$$
$$\left. |\gamma_t - \gamma^\star|, \|K_t - K^\star\| \right\} = O_p(\sqrt{\varphi_t}).$$

To establish convergence rates in Lemma 4.2, we have to leverage techniques for analyzing nonsmooth SGD (Davis et al. (2024), (10.1)) to introduce a stopping time to localize the analysis. See Appendix B.3 for the formal definition.

Before presenting the normality result, we introduce some additional notation. Let us define two sandwich matrices

$$\Omega^\star := (B^\star)^{-1} \mathbb{E}[\nabla F(\boldsymbol{x}^\star; \xi) \nabla^\top F(\boldsymbol{x}^\star; \xi)](B^\star)^{-1},$$
$$\Gamma^\star := \mathbb{E}[(I - \widetilde{K}^\star)\Omega^\star(I - \widetilde{K}^\star)^\top]. \quad (4.1)$$

In fact, $\Omega^\star$ is the limiting covariance of averaged SGD estimator (Ruppert, 1988; Polyak & Juditsky, 1992), which is asymptotically minimax optimal (Duchi & Ruan, 2021; Davis et al., 2024). However, due to randomized computation, $\Gamma^\star$ plays a crucial role in *our* limiting covariance. It depends on the accelerated sketching operator $\widetilde{K}^\star$ (cf. (3.2)), and its expectation is taken over $\tau$ sketching matrices $\{S_j\}_{j=0}^{\tau-1} \sim S$.

Now, with the notation in place, we present the following asymptotic normality of the last Newton iterate $\boldsymbol{x}_t$.

**Theorem 4.3.** *Under the conditions of Lemma 4.2, we have*

$$1/\sqrt{\varphi_t} \cdot (\boldsymbol{x}_t - \boldsymbol{x}^\star) \xrightarrow{d} \mathcal{N}(\boldsymbol{0}, \Sigma^\star),$$

*where $\Sigma^\star$ satisfies the following Lyapunov equation:*

$$[(I - K^\star) - \zeta I] \Sigma^\star + \Sigma^\star [(I - K^\star) - \zeta I] = \Gamma^\star, \quad (4.2)$$

*and $\zeta := \mathbf{1}_{\{\varphi=1\}}/2C_\varphi$.*

By standard properties of the Lyapunov equation (Khalil (2002, Theorem 4.6)), we know (4.2) always admits a unique positive semidefinite solution. This follows from the facts that $\Gamma^\star \succeq \boldsymbol{0}$ and $(I - K^\star) - \zeta I \succ \boldsymbol{0}$; the latter is guaranteed by the conditions on $\tau$ and $C_\varphi$ in Lemma 4.2.

To end this subsection, we draw connections between our limiting covariance $\Sigma^\star$ and those of exact and unaccelerated sketched Newton methods.

**Proposition 4.4.** *Consider $\Sigma^\star$ in (4.2) in two degenerate regimes.*

*(a) When the Newton system is exactly solved ($K^\star = \widetilde{K}^\star = \boldsymbol{0}$), we have $\Sigma^\star = 0.5\Omega^\star$ for $\varphi \in (0.5, 1)$ and $\Sigma^\star = \frac{C_\varphi \Omega^\star}{2C_\varphi - 1}$ for $\varphi = 1$. In the latter case, setting $\varphi = C_\varphi = 1$ leads to*

$$\sqrt{t} \cdot (\boldsymbol{x}_t - \boldsymbol{x}^\star) \xrightarrow{d} \mathcal{N}(\boldsymbol{0}, \Omega^\star).$$

*(b) When the accelerated rate is not provably achieved ($\gamma^\star = 1$), $\Sigma^\star$ is reduced to the solution to*

$$[(I - \mathcal{C}^\star) - \zeta I] \Sigma^\star + \Sigma^\star [(I - \mathcal{C}^\star) - \zeta I] = \mathcal{G}^\star, \quad (4.3)$$

*where $\mathcal{C}^\star := \mathbb{E}[\widetilde{\mathcal{C}}^\star]$ and $\mathcal{G}^\star := \mathbb{E}[(I - \widetilde{\mathcal{C}}^\star)\Omega^\star(I - \widetilde{\mathcal{C}}^\star)^\top]$ with $\widetilde{\mathcal{C}}^\star := \prod_{j=0}^{\tau-1}(I - B^\star S_j(S_j^\top (B^\star)^2 S_j)^\dagger S_j^\top B^\star)$.*

Case **(a)** matches the limiting covariance matrix of conditioned SGD method (Leluc & Portier, 2023), indicating that the Newton estimator attains optimal statistical efficiency on par with the averaged SGD estimator. Case **(b)** matches the limiting covariance matrix of unaccelerated sketched Newton method (Na & Mahoney, 2025); however, the latter corresponds only to Algorithm 1 under the parameter setup $(\alpha_t, \beta_t, \gamma_t) = (0.5, 0, 1)$. In contrast, our condition $\gamma^\star = 1$ is significantly weaker, achieved by our novel analysis.

### 4.2. Online covariance estimation

Building on the results of Section 4.1, we construct in this section an online estimator for the limiting covariance $\Sigma^\star$. With this covariance estimator, we can then perform online statistical inference, e.g., constructing asymptotically valid confidence intervals for the model parameters.

Inspired by prior work on covariance estimation (Chen et al., 2020; Zhu et al., 2021; Jiang et al., 2025; Kuang et al., 2025), we consider the weighted sample covariance estimator:

$$\widehat{\Sigma}_t := \frac{1}{t} \sum_{i=1}^{t} \frac{1}{\varphi_{i-1}}(\boldsymbol{x}_i - \bar{\boldsymbol{x}}_t)(\boldsymbol{x}_i - \bar{\boldsymbol{x}}_t)^\top \quad (4.4)$$

$$= \frac{1}{t} \sum_{i=1}^{t} \frac{\boldsymbol{x}_i \boldsymbol{x}_i^\top}{\varphi_{i-1}} - \frac{1}{t} \sum_{i=1}^{t} \frac{\boldsymbol{x}_i \bar{\boldsymbol{x}}_t^\top + \bar{\boldsymbol{x}}_t \boldsymbol{x}_i^\top}{\varphi_{i-1}} + \frac{1}{t} \sum_{i=1}^{t} \frac{\bar{\boldsymbol{x}}_t \bar{\boldsymbol{x}}_t^\top}{\varphi_{i-1}}.$$

The decomposition in the second line shows that all summations can be updated recursively, in the same spirit as (2.1). Hence, $\widehat{\Sigma}_t$ can be constructed efficiently in an online manner using only the accelerated Newton iterates, without requiring any additional matrix-vector products. Consequently, the inference procedure only needs to maintain a few running sums, rather than storing all past data samples or repeatedly solving auxiliary linear systems.

To show the consistency of $\widehat{\Sigma}_t$, we first need to provide non-asymptotic rates for the fourth moments of the iterate and Hessian average errors, $\mathbb{E}[\|\boldsymbol{x}_t - \boldsymbol{x}^\star\|^4]$ and $\mathbb{E}[\|B_t - B^\star\|^4]$.

**Lemma 4.5.** *Suppose Assumptions 3.1-3.4 hold with $q_g = q_H = 4, q_S = 2$; $\tau$ satisfies $\tau(1-\sqrt{\mu_t/\nu_t})^{\tau-2} \leq \gamma_H/(4\Upsilon_H)$ for all $t \geq 0$; and the stepsize parameters satisfy $C_\varphi > 0$ and $\varphi \in (0.5, 1)$. Then, we have*

$$\max\left\{\mathbb{E}[\|\boldsymbol{x}_t - \boldsymbol{x}^\star\|^4],\ \mathbb{E}[\|B_t - B^\star\|^4]\right\} = O(\varphi_t^2).$$

With the above lemma, we show the convergence rate of $\widehat{\Sigma}_t$ in the following theorem.

**Theorem 4.6.** *Under the conditions of Lemma 4.5, the covariance estimator $\widehat{\Sigma}_t$ defined in (4.4) satisfies*

$$\mathbb{E}[\|\widehat{\Sigma}_t - \Sigma^\star\|] = O(1/\sqrt{t\varphi_t}).$$

Since $t\varphi_t \to \infty$ as $t \to \infty$, Theorem 4.6 implies that $\widehat{\Sigma}_t$ is a consistent estimator of $\Sigma^\star$. An immediate consequence of this consistency is the construction of confidence intervals:

$$P\left(\boldsymbol{w}^\top\boldsymbol{x}^\star \in \left[\boldsymbol{w}^\top\boldsymbol{x}_t \pm z_{1-q/2}\{\varphi_t\boldsymbol{w}^\top\widehat{\Sigma}_t\boldsymbol{w}\}^{0.5}\right]\right) \to 1-q$$

as $t \to \infty$. Here, $z_{1-q/2}$ is the $(1-q/2)$-quantile of $\mathcal{N}(0,1)$.

# 5. Numerical Experiment

We empirically evaluate online Newton with Nesterov's accelerated sketching and compare it with exact Newton, unaccelerated sketched Newton, and the SGD baseline. Our main goal is to examine how computational acceleration affects online inference, especially its empirical coverage and statistical efficiency as reflected by confidence-interval length. We compare accelerated and unaccelerated sketching through covariance QQ plots, and include SGD in Tables 1 and 2 as a first-order benchmark. We also evaluate our inference procedure based on the covariance estimator $\widehat{\Sigma}_t$. Inference performance is assessed by the mean absolute error, the empirical coverage rate of the constructed confidence intervals, and their average lengths over 200 independent runs. Due to space limits, we present a subset of results, while complete experimental setups and results are in Appendix C.

Following prior literature (Chen et al., 2020; Zhu et al., 2024; Na & Mahoney, 2025; Kuang et al., 2025), we consider the same linear and logistic regressions with varying dimensions $d$ and covariance $\Sigma_a$ for random covariates. We also vary the sketching steps $\tau$ ($\tau = \infty$ corresponds to exact Newton) and sketching distributions (Kaczmarz and Gaussian). For accelerated sketching, Tables 1 and 2 report three refresh periods $N \in \{1, 500, 1000\}$ for recomputing the acceleration parameters $(\mu_t, \nu_t)$, where $N = 1$ denotes recomputation at every iteration and $N = 500, 1000$ are periodic updates. This comparison helps assess whether less frequent parameter refresh can reduce implementation overhead while maintaining comparable coverage and interval lengths.

• **Covariance of accelerated vs. unaccelerated sketching.** Since accelerated sketching enhances computational

efficiency of Newton methods, a natural question is *whether accelerated sketching introduces additional uncertainty to the estimation procedure*. We use QQ plots to compare the covariance matrices in (4.2) and (4.3). We examine several combinations of $(d, \tau)$, as reported in Figure 1. From Figure 1, we observe that the quantiles of accelerated sketched online Newton closely align with those of the unaccelerated scheme, indicating similar empirical distributions and comparable asymptotic covariance structures. This comparison is nontrivial because, although both solvers approximate the same Newton direction, acceleration changes the inner state–costate dynamics of the sketching solver. The observed alignment therefore suggests that accelerating the sketching solver does not degrade statistical efficiency or introduce additional leading-order estimation uncertainty, an aspect largely missing in the literature. Beyond comparing the covariance matrices, these QQ plots also provide empirical evidence for the normality of accelerated Newton iterates (Theorem 4.3).

• **Asymptotic validity of inference.** Table 1 shows that the empirical coverage rates of the confidence intervals constructed by accelerated sketched online Newton remain close to the nominal 95% level across most sketching settings and refresh periods $N \in \{1, 500, 1000\}$. In particular, the periodic choices $N = 500$ and $N = 1000$ maintain near-nominal coverage, supporting the discussion above that exact per-iteration recomputation of the acceleration parameters $(\mu_t, \nu_t)$ is not necessary for valid inference. In contrast, the SGD baseline exhibits clear undercoverage in several settings, consistent with prior observations for first-order online inference (Zhu et al., 2021; Kuang et al., 2025). This provides numerical support for our asymptotic normality and covariance estimation theory (Theorems 4.3 and 4.6), and motivates the use of second-order methods for robust online inference.

• **Mean absolute error.** Compared to exact Newton method ($\tau = \infty$), our inexact Newton method achieves comparable mean absolute errors in both linear and logistic regressions. This aligns with the fact that both methods are $\sqrt{\varphi_t}$-consistent, implying the same leading-order rate for the estimation error. Therefore, the observed error differences mainly reflect finite-sample and algorithmic effects, rather than a change in the asymptotic order. Notably, in several linear regression settings (especially when $\Sigma_a$ is Toeplitz or Equi-correlation), our inexact Newton method attains even smaller iterate errors than the exact method, with reductions varying from 8.8% to 37.5% as reported in Table 1.

• **Statistical efficiency of accelerated vs. exact Newton.** As shown in Table 1, in most settings, inexact Newton method yields slightly longer average confidence intervals than exact Newton method. This modest inflation is indeed expected, as the additional randomness introduced by the sketching solver can increase the variability of the resulting iterates and, consequently, the estimated asymptotic variance. The key point is that this inflation remains modest relative to the

| $d$ | $\Sigma_a$ | $\tau$ | Linear Regression (Kaczmarz) | | | Linear Regression (Gaussian) | | | Logistic Regression (Kaczmarz) | | | Logistic Regression (Gaussian) | | |
|---|---|---|---|---|---|---|---|---|---|---|---|---|---|---|
| | | | MAE $(10^{-2})$ | Ave Cov (%) | Ave Len $(10^{-2})$ | MAE $(10^{-2})$ | Ave Cov (%) | Ave Len $(10^{-2})$ | MAE $(10^{-2})$ | Ave Cov (%) | Ave Len $(10^{-2})$ | MAE $(10^{-2})$ | Ave Cov (%) | Ave Len $(10^{-2})$ |
| 40 | Identity | SGD | 25.49 | 62.50 | 0.65 | — | — | — | 7.79 | 96.50 | 0.64 | — | — | — |
| | | $\infty$ | 25.80 | 93.00 | 2.55 | 25.24 | 98.00 | 2.56 | 3.76 | 94.50 | 0.24 | 3.80 | 95.00 | 0.24 |
| | | 10 | 25.64 | 93.00 | 2.55 | 27.53 | 96.00 | 2.72 | 3.76 | 94.50 | 0.33 | 4.10 | 94.00 | 0.36 |
| | | | 25.68 | 96.50 | 2.54 | 27.66 | 92.00 | 2.72 | 3.83 | 95.50 | 0.33 | 4.03 | 97.00 | 0.36 |
| | | | 26.16 | 94.50 | 2.49 | 27.60 | 95.00 | 2.73 | 3.77 | 95.00 | 0.33 | 3.99 | 93.50 | 0.36 |
| | | 5 | 26.08 | 94.50 | 2.50 | 26.31 | 96.00 | 2.58 | 3.79 | 92.50 | 0.34 | 3.91 | 94.00 | 0.35 |
| | | | 25.94 | 92.50 | 2.44 | 26.56 | 95.50 | 2.54 | 3.83 | 95.00 | 0.34 | 3.90 | 93.00 | 0.35 |
| | | | 25.79 | 95.50 | 2.47 | 26.75 | 96.50 | 2.58 | 3.79 | 91.00 | 0.34 | 3.87 | 92.50 | 0.35 |
| | Toeplitz $r = 0.4$ | SGD | 25.96 | 68.50 | 0.65 | — | — | — | 6.40 | 95.00 | 0.63 | — | — | — |
| | | $\infty$ | 29.87 | 96.00 | 1.70 | 30.00 | 95.50 | 1.70 | 3.09 | 97.00 | 0.27 | 5.63 | 95.50 | 0.33 |
| | | 10 | 22.37 | 94.00 | 2.17 | 24.30 | 95.00 | 2.23 | 3.13 | 96.50 | 0.29 | 6.18 | 93.50 | 0.53 |
| | | | 22.92 | 93.50 | 2.18 | 24.26 | 91.00 | 2.25 | 3.13 | 92.00 | 0.30 | 5.98 | 95.50 | 0.53 |
| | | | 22.86 | 94.00 | 2.16 | 24.81 | 92.50 | 2.23 | 3.17 | 94.50 | 0.29 | 6.15 | 95.00 | 0.53 |
| | | 5 | 22.24 | 94.00 | 2.17 | 23.78 | 94.50 | 2.18 | 3.15 | 92.00 | 0.29 | 5.88 | 96.50 | 0.52 |
| | | | 22.30 | 95.50 | 2.17 | 23.53 | 94.00 | 2.20 | 3.13 | 94.50 | 0.30 | 5.82 | 97.00 | 0.52 |
| | | | 22.34 | 97.50 | 2.17 | 23.42 | 97.00 | 2.19 | 3.17 | 93.50 | 0.30 | 5.90 | 91.00 | 0.52 |
| | Equi-Corr $r = 0.4$ | SGD | 27.11 | 88.50 | 0.64 | — | — | — | 5.27 | 98.50 | 0.63 | — | — | — |
| | | $\infty$ | 20.11 | 92.00 | 0.61 | 19.76 | 95.00 | 0.61 | 2.67 | 93.50 | 0.19 | 2.59 | 96.00 | 0.19 |
| | | 10 | 13.55 | 92.00 | 1.03 | 18.85 | 95.50 | 1.09 | 2.67 | 93.50 | 0.24 | 2.81 | 91.50 | 0.26 |
| | | | 13.33 | 96.50 | 1.02 | 19.04 | 95.00 | 1.09 | 2.67 | 92.00 | 0.24 | 2.83 | 95.00 | 0.26 |
| | | | 13.23 | 92.50 | 1.03 | 18.98 | 94.00 | 1.09 | 2.67 | 93.50 | 0.24 | 2.84 | 95.00 | 0.26 |
| | | 5 | 12.56 | 98.00 | 1.05 | 18.03 | 92.50 | 1.12 | 2.68 | 92.00 | 0.24 | 2.72 | 96.50 | 0.25 |
| | | | 12.79 | 94.00 | 1.04 | 18.12 | 93.00 | 1.11 | 2.64 | 94.50 | 0.24 | 2.79 | 92.50 | 0.25 |
| | | | 12.87 | 94.50 | 1.04 | 18.24 | 94.00 | 1.12 | 2.69 | 94.00 | 0.24 | 2.76 | 95.50 | 0.25 |

*Table 1. A subset of evaluation results for the inference procedure under different parameter choices with $d = 40$. 'MAE' denotes the mean absolute iterate error over 200 independent runs, Ave Cov" denotes the empirical coverage rate, and Ave Len" denotes the average confidence-interval length. The row labeled SGD" reports the standard SGD baseline. Since SGD does not involve sketching, there is no distinction between the Kaczmarz and Gaussian variants; accordingly, the SGD results are reported only once for each regression model, and redundant entries are marked by —". For the sketching-based methods with $\tau \in \{5, 10\}$, each three-row block, from top to bottom, corresponds to a different refresh period $N \in \{1, 500, 1000\}$ for the acceleration parameters $(\mu_t, \nu_t)$. Here, $N = 1$ recovers the standard non-periodic setting.*

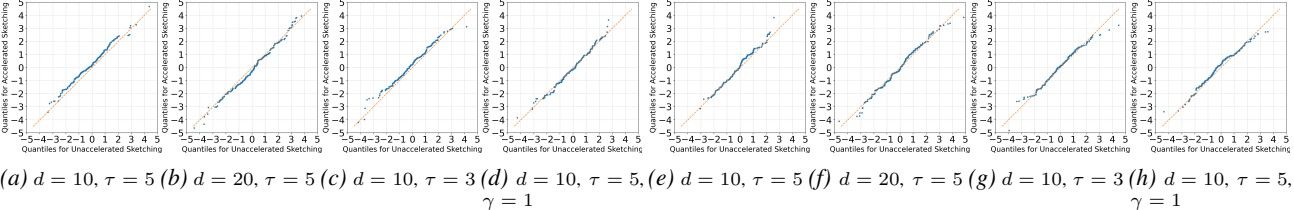

*(a) $d = 10, \tau = 5$ (b) $d = 20, \tau = 5$ (c) $d = 10, \tau = 3$ (d) $d = 10, \tau = 5,$ (e) $d = 10, \tau = 5$ (f) $d = 20, \tau = 5$ (g) $d = 10, \tau = 3$ (h) $d = 10, \tau = 5,$*
$\gamma = 1$ $\gamma = 1$
*Figure 1. A subset of QQ plots for linear regression with Kaczmarz (a-d) and Gaussian (e-h) accelerated v.s. unaccelerated sketching.*

computational savings gained by replacing exact Newton solves with accelerated sketching.

Overall, our inexact Newton method achieves near-nominal coverage, comparable (and in some cases improved) mean absolute errors, and only modest increases in averaged confidence interval length relative to the exact Newton method. These results suggest that our inference procedure preserves asymptotic validity, numerical accuracy, and statistical efficiency, while benefiting from faster theoretical convergence (compared to unaccelerated sketching) and lower computational costs (compared to the exact Newton). Furthermore, while Kuang et al. (2025) employs at least 10 sketching steps to achieve competitive performance in large dimensions, our method attains comparable or even superior results with at most 5 sketching steps. This reduction of sketching steps leads to substantial computational savings in practice, highlighting the practical advantages of Nesterov's acceleration for online Newton-based inference.

## 6. Conclusion

We studied uncertainty quantification and inference of online Newton methods with Nesterov's accelerated sketching. The proposed method has $O(d^2)$ time and memory complexity, matching that of first-order methods. We established global almost-sure convergence, derived asymptotic normality, and characterized the limiting covariance through a Lyapunov equation that explicitly captures the effect of the sketching solver. We further proposed a fully online covariance estimator with non-asymptotic guarantees, enabling practical inference without inverting any matrices. Our study clarifies the *computational-statistical trade-offs* induced by accelerated sketching and connects resulting inference behavior to that of exact and unaccelerated sketched Newton methods. Empirically, the method maintains near-nominal coverage while using only a small number of sketching steps, suggesting that acceleration can improve computational efficiency without noticeably sacrificing statistical efficiency.

## Impact Statement

This paper presents work on the theoretical and algorithmic foundations of online Newton methods with Nesterov's accelerated sketching, with a particular focus on uncertainty quantification and inference in streaming data settings. The results advance the fields of machine learning, optimization, statistics, and computation. There are many potential societal consequences of our work, none of which we feel must be specifically highlighted here.

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

# Appendix: Inference of Online Newton Methods with Nesterov's Accelerated Sketching

## A. Proofs of Section 3

### A.1. Proof of Lemma 3.5

First, we derive the recursion of $(z_{t,j} - \Delta x_t, v_{t,j} - \Delta x_t)$ for any fixed outer loop iteration $t$. In Section 2.2, we temporarily suppress the outer iteration index and write $\{z_j, v_j, y_j\}$. Below, we write $\{z_{t,j}, v_{t,j}, y_{t,j}\}$ to make the dependence on the outer index $t$ explicit. By Algorithm 1, we have

$$
\begin{aligned}
z_{t,j+1} - \Delta x_t &= y_{t,j} - \Delta x_t - \omega_{t,j} \\
&= \alpha_t(v_{t,j} - \Delta x_t) + (1 - \alpha_t)(z_{t,j} - \Delta x_t) - \widetilde{Z}_{t,j}(y_{t,j} - \Delta x_t) \\
&= (I - \widetilde{Z}_{t,j})[\alpha_t(v_{t,j} - \Delta x_t) + (1 - \alpha_t)(z_{t,j} - \Delta x_t)],
\end{aligned}
$$

and

$$
\begin{aligned}
v_{t,j+1} - \Delta x_t &= \beta_t(v_{t,j} - \Delta x_t) + (1 - \beta_t)(y_{t,j} - \Delta x_t) - \gamma_t \widetilde{Z}_{t,j}(y_{t,j} - \Delta x_t) \\
&= \beta_t(v_{t,j} - \Delta x_t) + (I - \beta_t I - \gamma_t \widetilde{Z}_{t,j})(y_{t,j} - \Delta x_t) \\
&= \beta_t(v_{t,j} - \Delta x_t) + (I - \beta_t I - \gamma_t \widetilde{Z}_{t,j})(\alpha_t(v_{t,j} - \Delta x_t) + (1 - \alpha_t)(z_{t,j} - \Delta x_t)) \\
&= [(\alpha_t + \beta_t - \alpha_t\beta_t)I - \alpha_t\gamma_t \widetilde{Z}_{t,j}](v_{t,j} - \Delta x_t) + [(1 - \alpha_t)(1 - \beta_t)I - (1 - \alpha_t)\gamma_t \widetilde{Z}_{t,j}](z_{t,j} - \Delta x_t).
\end{aligned}
$$

Combining the above two displays, we have the following recursion formula:

$$
\begin{pmatrix} z_{t,j+1} - \Delta x_t \\ v_{t,j+1} - \Delta x_t \end{pmatrix} = \underbrace{\begin{pmatrix} (1 - \alpha_t)(I - \widetilde{Z}_{t,j}) & \alpha_t(I - \widetilde{Z}_{t,j}) \\ (1 - \alpha_t)(1 - \beta_t)I - (1 - \alpha_t)\gamma_t \widetilde{Z}_{t,j} & (\alpha_t + \beta_t - \alpha_t\beta_t)I - \alpha_t\gamma_t \widetilde{Z}_{t,j} \end{pmatrix}}_{\widetilde{C}_{t,j}} \begin{pmatrix} z_{t,j} - \Delta x_t \\ v_{t,j} - \Delta x_t \end{pmatrix}. \quad \text{(A.1)}
$$

With the above formula (A.1), we have the following closed form of $(z_{t,\tau} - \Delta x_t, v_{t,\tau} - \Delta x_t)$:

$$
\begin{aligned}
\begin{pmatrix} z_{t,\tau} - \Delta x_t \\ v_{t,\tau} - \Delta x_t \end{pmatrix} &\overset{\text{(A.1)}}{=} \widetilde{C}_{t,\tau-1} \begin{pmatrix} x_{t,\tau-1} - \Delta x_t \\ v_{t,\tau-1} - \Delta x_t \end{pmatrix} = \cdots = \left( \prod_{j=0}^{\tau-1} \widetilde{C}_{t,j} \right) \begin{pmatrix} z_{t,0} - \Delta x_t \\ v_{t,0} - \Delta x_t \end{pmatrix} \\
&= -\left( \prod_{j=0}^{\tau-1} \widetilde{C}_{t,j} \right) \begin{pmatrix} \Delta x_t \\ \Delta x_t \end{pmatrix} = -\widetilde{C}_t \begin{pmatrix} \Delta x_t \\ \Delta x_t \end{pmatrix},
\end{aligned}
$$

where the second last equality holds since we set $z_{t,0} = v_{t,0} = 0$ in Algorithm 1. Thus, for any fixed $\tau > 0$, we have the following closed form for the sketched solver operator:

$$
\begin{pmatrix} z_{t,\tau} \\ v_{t,\tau} \end{pmatrix} = (I - \widetilde{C}_t) \begin{pmatrix} \Delta x_t \\ \Delta x_t \end{pmatrix}.
$$

By the definition of $\widetilde{K}_t$ in (3.2) and the fact that $\Delta x_t = -B_t^{-1}g_t$, we have

$$
\begin{pmatrix} I & 0 \end{pmatrix} \begin{pmatrix} z_{t,\tau} \\ v_{t,\tau} \end{pmatrix} = \begin{pmatrix} I & 0 \end{pmatrix} (I - \widetilde{C}_t) \begin{pmatrix} I \\ I \end{pmatrix} \Delta x_t \implies z_{t,\tau} = (I - \widetilde{K}_t)\Delta x_t = -(I - \widetilde{K}_t)B_t^{-1}g_t,
$$

which gives us the first part of Lemma 3.5.

For any $t \geq 0$, by the above conclusion that $z_{t,\tau} = (I - \widetilde{K}_t)B_t^{-1}g_t$, we have

$$
\begin{aligned}
\mathbb{E}[z_{t,\tau} \mid \mathcal{F}_{t-1}] &= -\mathbb{E}[(I - \widetilde{K}_t)B_t^{-1}g_t \mid \mathcal{F}_{t-1}] = -\mathbb{E}[\mathbb{E}[(I - \widetilde{K}_t)B_t^{-1}g_t \mid \mathcal{F}_{t-0.5}] \mid \mathcal{F}_{t-1}] \\
&= -\mathbb{E}[(I - K_t)B_t^{-1}g_t \mid \mathcal{F}_{t-1}] = -(I - K_t)B_t^{-1}\nabla f_t. \quad \text{(A.2)}
\end{aligned}
$$

Moreover, we have

$$
\begin{aligned}
\mathbb{E}[\boldsymbol{z}_{t,\tau} - (I - K_t)\Delta \boldsymbol{x}_t \mid \mathcal{F}_{t-1}] &\stackrel{(A.2)}{=} -(I - K_t)B_t^{-1}\nabla f_t - (I - K_t)\mathbb{E}[\Delta \boldsymbol{x}_t \mid \mathcal{F}_{t-1}] \\
&= -(I - K_t)B_t^{-1}\nabla f_t + (I - K_t)\mathbb{E}[B_t^{-1}g_t \mid \mathcal{F}_{t-1}] \\
&= -(I - K_t)B_t^{-1}\nabla f_t + (I - K_t)B_t^{-1}\nabla f_t \\
&= \mathbf{0}.
\end{aligned}
$$

We complete the proof of the second part of Lemma 3.5.

Finally, by the definition of $\widetilde{C}_{t,j}$ in (A.1), we claim that when $\gamma_t = 1$, for any $Y \in \mathbb{R}^{d \times d}$ and any $j \geq 0$, we have

$$
\widetilde{C}_{t,j}\left(\begin{pmatrix} 1 \\ 1 \end{pmatrix} \otimes Y\right) = \begin{pmatrix} 1 \\ 1 \end{pmatrix} \otimes \left((I_d - \widetilde{Z}_{t,j})Y\right). \tag{A.3}
$$

In fact, by the property of Kronecker product that $(A \otimes B)(C \times D) = (AC) \otimes (BD)$, we have

$$
\begin{aligned}
\widetilde{C}_{t,j}\left(\begin{pmatrix} 1 \\ 1 \end{pmatrix} \otimes Y\right) &= \begin{pmatrix} 1 - \alpha_t & \alpha_t \\ (1 - \alpha_t)(1 - \beta_t) & \alpha_t + \beta_t - \alpha_t\beta_t \end{pmatrix}\begin{pmatrix} 1 \\ 1 \end{pmatrix} \otimes Y - \begin{pmatrix} 1 - \alpha_t & \alpha_t \\ 1 - \alpha_t & \alpha_t \end{pmatrix}\begin{pmatrix} 1 \\ 1 \end{pmatrix} \otimes \widetilde{Z}_{t,j}Y \\
&= \begin{pmatrix} 1 \\ 1 \end{pmatrix} \otimes Y - \begin{pmatrix} 1 \\ 1 \end{pmatrix} \otimes \widetilde{Z}_{t,j}Y = \begin{pmatrix} 1 \\ 1 \end{pmatrix} \otimes \left((I_d - \widetilde{Z}_{t,j})Y\right).
\end{aligned}
$$

By the above property, we then obtain

$$
\widetilde{C}_t\begin{pmatrix} I_d \\ I_d \end{pmatrix} = \widetilde{C}_{t,\tau-1}\widetilde{C}_{t,\tau-2}\cdots\widetilde{C}_{t,0}\left(\begin{pmatrix} 1 \\ 1 \end{pmatrix} \otimes I_d\right) \stackrel{(A.3)}{=} \begin{pmatrix} 1 \\ 1 \end{pmatrix} \otimes \left(\prod_{j=0}^{\tau-1}(I_d - \widetilde{Z}_{t,j})\right).
$$

We then further obtain

$$
\widetilde{K}_t \stackrel{(3.2)}{=} \begin{pmatrix} I_d & \mathbf{0} \end{pmatrix} \widetilde{C}_t\begin{pmatrix} I_d \\ I_d \end{pmatrix} = \begin{pmatrix} I_d & \mathbf{0} \end{pmatrix}\left(\begin{pmatrix} 1 \\ 1 \end{pmatrix} \otimes \left(\prod_{j=0}^{\tau-1}(I_d - \widetilde{Z}_{t,j})\right)\right) = \begin{pmatrix} I & \mathbf{0} \end{pmatrix}\begin{pmatrix} \prod_{j=0}^{\tau-1}(I_d - \widetilde{Z}_{t,j}) \\ \prod_{j=0}^{\tau-1}(I_d - \widetilde{Z}_{t,j}) \end{pmatrix} = \prod_{j=0}^{\tau-1}(I_d - \widetilde{Z}_{t,j}).
$$

This completes the proof of the final part of Lemma 3.5.

### A.2. Proof of Lemma 3.6

• **Proof of** (3.4). Recall the definition of $G_t(z)$ in (3.3). By Cayley–Hamilton Theorem for a $2 \times 2$ matrix, we have

$$
G_t(z)^2 - \text{Tr}_t(z)G_t(z) + D_t(z)I = \mathbf{0},
$$

where $\text{Tr}_t(z) = \text{trace}(G_t(z))$, and $D_t(z) = \det(G_t(z))$. Left-multiplying both sides by $e_1^\top G_t(z)^{\tau-2}$ and right-multiplying by $\mathbf{1}$ yields

$$
p_{t,\tau}(z) = \text{Tr}_t(z)p_{t,\tau-1}(z) - D_t(z)p_{t,\tau-2}(z), \qquad \forall\, \tau \geq 2.
$$

It is a second-order recursion form, with initialization $p_{t,0}(z) = 1$ and $p_{t,1}(z) = e_1^\top G_t(z)\mathbf{1} = 1 - z$.

• **Proof of** (3.5). Recall that $\text{Tr}_t(z) = \text{trace}(G_t(z)) = 1 + \beta_t - \alpha_t\beta_t - z(1 - \alpha_t + \alpha_t\gamma_t)$ and $D_t(z) = \det(G_t(z)) = (1 - \alpha_t)\beta_t(1 - z)$. By the definition of $\alpha_t$, $\beta_t$ and $\gamma_t$ in (2.4), we know

$$
\text{Tr}_t(z) = \frac{2\nu_t - (\nu_t + 1)z}{\sqrt{\nu_t}(\sqrt{\mu_t} + \sqrt{\nu_t})}, \qquad D_t(z) = \frac{\sqrt{\nu_t} - \sqrt{\mu_t}}{\sqrt{\nu_t} + \sqrt{\mu_t}}(1 - z). \tag{A.4}
$$

Also eigenvalues of $G_t(z)$ are the roots of the following characteristic equation:

$$
\phi(\lambda) = \lambda^2 - \text{Tr}_t(z)\lambda + D_t(z) = 0.
$$

Hence we obtain (to ease notation, we drop the index $t$ for $\lambda$)

$$\lambda = \frac{\mathrm{Tr}_t(z) \pm \sqrt{\mathrm{Tr}_t^2(z) - 4D_t(z)}}{2} \overset{(A.4)}{=} \frac{2\nu_t - (\nu_t+1)z \pm \sqrt{(\nu_t+1)^2 z^2 - 4\nu_t(\mu_t+1)z + 4\mu_t\nu_t}}{2\sqrt{\nu_t}(\sqrt{\nu_t}+\sqrt{\mu_t})}. \tag{A.5}$$

Recall that $z \in [\mu_t, 1]$ and $1 \le \nu_t \le 1/\mu_t$ (cf. (2.6)). Since

$$16\nu_t^2(\mu_t+1)^2 - 16(\nu_t+1)^2\mu_t\nu_t = 16\nu_t[\nu_t(\mu_t+1)^2 - (\nu_t+1)^2\mu_t] = 16\nu_t(\nu_t-\mu_t)(1-\mu_t\nu_t) \ge 0,$$

we know $(\nu_t+1)^2 z^2 - 4\nu_t(\mu_t+1)z + 4\mu_t\nu_t$ as a function of $z$ must have real roots. Below we consider two cases.

- **Case 1:** $(\nu_t - \mu_t)(1 - \mu_t\nu_t) > 0$. In this case, $(\nu_t+1)^2 z^2 - 4\nu_t(\mu_t+1)z + 4\mu_t\nu_t$ has two distinctive real roots, which are denoted as $z_1$ and $z_2$. Plugging the two boundaries $z = \mu_t$ and $z = 1$ and observing that the function values are both nonnegative, we know $\mu_t \le z_1 \le \frac{2\nu_t(\mu_t+1)}{(\nu_t+1)^2} \le z_2 \le 1$. We take the derivative of the larger one in (A.5) and obtain

$$\frac{d\lambda}{dz} = \frac{-(\nu_t+1) + \frac{(\nu_t+1)^2 z - 2\nu_t(\mu_t+1)}{\sqrt{(\nu_t+1)^2 z^2 - 4\nu_t(\mu_t+1)z + 4\mu_t\nu_t}}}{2\sqrt{\nu_t}(\sqrt{\nu_t}+\sqrt{\mu_t})}. \tag{A.6}$$

Note that

$$\begin{aligned}
((\nu_t+1)^2 z &- 2\nu_t(\mu_t+1))^2 - (\nu_t+1)^2((\nu_t+1)^2 z^2 - 4\nu_t(\mu_t+1)z + 4\mu_t\nu_t) \\
&= (\nu_t+1)^4 z^2 - 4\nu_t(\nu_t+1)^2(\mu_t+1)^2 z + 4\nu_t^2(\mu_t+1)^2 - (\nu_t+1)^4 z^2 \\
&\quad + 4\nu_t(\nu_t+1)^2(\mu_t+1)^2 z - 4(\nu_t+1)^2\mu_t\nu_t \\
&= 4\nu_t^2(\mu_t+1)^2 - 4(\nu_t+1)^2\mu_t\nu_t \\
&= 4\nu_t(\nu_t(\mu_t+1)^2 - \mu_t(\nu_t+1)^2) \\
&= 4\nu_t(\nu_t\mu_t^2 + \nu_t - \mu_t\nu_t^2 - \mu_t) \\
&= 4\nu_t(\nu_t - \mu_t)(1 - \nu_t\mu_t) \ge 0. \tag{A.7}
\end{aligned}$$

As a result, $(\nu_t+1)^2 z - 2\nu_t(\mu_t+1) \ge (\nu_t+1)\sqrt{(\nu_t+1)^2 z^2 - 4\nu_t(\mu_t+1)z + 4\mu_t\nu_t}$ when $(\nu_t+1)^2 z - 2\nu_t(\mu_t+1) \ge 0$, i.e., when $z \ge 2\nu_t(\mu_t+1)/(\nu_t+1)^2$.

**Case 1a:** $z \in [\mu_t, z_1] \cup [z_2, 1]$. In this case, $\lambda$ is real. According to (A.6) and (A.7), we have, on $z \in [\mu_t, z_1]$, $d\lambda/dz < 0$; on $z \in [z_2, 1]$, $d\lambda/dz > 0$. Therefore, $\lambda$ decreases on $[\mu_t, z_1]$ and increases on $[z_2, 1]$. Hence, the supremum is achieved on $z = \mu_t$ or $z = 1$, i.e.,

$$\begin{aligned}
\sup_{z \in [\mu_t, z_1] \cup [z_2, 1]} \rho(G_t(z)) &\le \max\left\{ \frac{\mathrm{Tr}_t(1) + \sqrt{\mathrm{Tr}_t^2(1) - 4D_t(1)}}{2}, \frac{\mathrm{Tr}_t(\mu_t) + \sqrt{\mathrm{Tr}_t^2(\mu_t) - 4D_t(\mu_t)}}{2} \right\} \\
&\le \max\left\{ \frac{\nu_t - 1}{\sqrt{\nu_t}(\sqrt{\mu_t}+\sqrt{\nu_t})}, 1 - \sqrt{\frac{\mu_t}{\nu_t}} \right\} = 1 - \sqrt{\frac{\mu_t}{\nu_t}}.
\end{aligned}$$

**Case 1b:** $z \in (z_1, z_2)$. In this case, $\lambda$ is complex, and we have

$$\sup_{z \in [z_1, z_2]} \rho(G_t(z)) = \sup_{z \in [z_1, z_2]} \sqrt{D_t(z)} \overset{(A.4)}{\le} \sqrt{\frac{\sqrt{\nu_t} - \sqrt{\mu_t}}{\sqrt{\nu_t} + \sqrt{\mu_t}}}(1 - \mu_t),$$

where the last inequality holds since $D_t(z)$ is a linear function of $z$.

Combining **Case 1a** and **Case 1b**, and observing that

$$\left(1 - \sqrt{\frac{\mu_t}{\nu_t}}\right)^2 - \frac{\sqrt{\nu_t} - \sqrt{\mu_t}}{\sqrt{\nu_t} + \sqrt{\mu_t}}(1 - \mu_t) = \frac{(\sqrt{\nu_t} - \sqrt{\mu_t})\mu_t(\nu_t - 1)}{\nu_t(\sqrt{\nu_t} + \sqrt{\mu_t})} \ge 0,$$

we obtain $\sup_{z \in [\mu_t, 1]} \rho(G_t(z)) \le 1 - \sqrt{\mu_t/\nu_t}$ for all $t \ge 0$.

- **Case 2:** $(\nu_t - \mu_t)(1 - \mu_t \nu_t) = 0$. In this case, $(\nu_t + 1)^2 z^2 - 4\nu_t(\mu_t + 1)z + 4\mu_t \nu_t$ has two identical real roots $2\nu_t(\mu_t + 1)/(\nu_t + 1)^2$. Thus, (A.5) becomes

$$\lambda = \frac{\text{Tr}_t(z) \pm \sqrt{\text{Tr}_t^2(z) - 4D_t(z)}}{2} \overset{\text{(A.4)}}{=} \frac{2\nu_t - (\nu_t + 1)z \pm \left| (\nu_t + 1)z - \frac{2\nu_t(\mu_t+1)}{\nu_t+1} \right|}{2\sqrt{\nu_t}(\sqrt{\nu_t} + \sqrt{\mu_t})}.$$

We can see that the spectral radius of $G_t(z)$ is piecewise linear, and the supremum is attained on $z = \mu_t$ or $z = 1$. By plugging the two boundaries into the above display, we further conclude the supremum is attained on $z = \mu_t$. Thus, we have $\sup_{z \in [\mu_t, 1]} \rho(G_t(z)) \leq 1 - \sqrt{\mu_t/\nu_t}$ for all $t \geq 0$.

By Assumption 3.4, we have $\|Z_t^{-1}\| \leq 1/\gamma_S$. Therefore, by (2.6) and the definition of $\gamma_t$ in (2.4), it follows that

$$1 \overset{(2.6)}{\leq} \gamma_t \overset{(2.4)}{=} 1/\sqrt{\mu_t \nu_t} \overset{(2.6)}{\leq} 1/\sqrt{\mu_t} \overset{(2.6)}{\leq} 1/\sqrt{\gamma_S}. \tag{A.8}$$

This implies that $1 - \sqrt{\mu_t/\nu_t} \leq 1 - \mu_t \leq 1 - \gamma_S =: \rho^\star$. Combining **Case 1** and **Case 2** completes the proof of (3.5).

- **Proof of** (3.6). In the following, we will consider three cases.

•• **Case 1:** $G_t(z)$ **has two distinct real eigenvalues.** Let us denote the two distinct real eigenvalues of $G_t(z)$ as $\lambda_1$ and $\lambda_2$, and we know $\text{Tr}_t(z)^2 - 4D_t(z) > 0$. The solution for the recursion (3.4) gives us

$$p_{t,\tau}(z) = A\lambda_1^\tau + B\lambda_2^\tau.$$

We use the initial conditions to determine $A$ and $B$: $p_{t,0}(z) = A + B = 1$ and $p_{t,1}(z) = A\lambda_1 + B\lambda_2 = 1 - z$. By solving these equations, we have

$$A = \frac{1 - z - \lambda_2}{\lambda_1 - \lambda_2}, \qquad B = \frac{\lambda_1 - (1 - z)}{\lambda_1 - \lambda_2}.$$

Combining the above two displays, we get the solution of $p_{t,\tau}(z)$ as

$$\begin{aligned}
p_{t,\tau}(z) &= \left( \frac{1 - z - \lambda_2}{\lambda_1 - \lambda_2} \right) \lambda_1^\tau + \frac{\lambda_1 - (1 - z)}{\lambda_1 - \lambda_2} \lambda_2^\tau \\
&= (1 - z) \frac{\lambda_1^\tau - \lambda_2^\tau}{\lambda_1 - \lambda_2} - \lambda_1 \lambda_2 \frac{\lambda_1^{\tau-1} - \lambda_2^{\tau-1}}{\lambda_1 - \lambda_2} \\
&= (1 - z) \sum_{k=0}^{\tau-1} \lambda_1^{\tau-1-k} \lambda_2^k - \lambda_1 \lambda_2 \sum_{k=0}^{\tau-2} \lambda_1^{\tau-2-k} \lambda_2^k,
\end{aligned} \tag{A.9}$$

where the last equality uses the identity that $\frac{a^n - b^n}{a - b} = \sum_{k=0}^{n-1} a^{n-1-k} b^k$ for real numbers $a \neq b$. By (3.5), we know that $\max\{|\lambda_1|, |\lambda_2|\} \leq \sup_{z \in [\mu_t, 1]} \rho(G_t(z)) \leq 1 - \sqrt{\mu_t/\nu_t}$ for all $t \geq 0$. Then, (A.9) leads to the following derivation

$$\sup_{z \in [\mu_t, 1]} |p_{t,\tau}(z)| \overset{\text{(A.9)}}{\leq} \tau \left( 1 - \sqrt{\mu_t/\nu_t} \right)^{\tau-1} + (\tau - 1) \left( 1 - \sqrt{\mu_t/\nu_t} \right)^{\tau-2} \leq 2\tau \left( 1 - \sqrt{\mu_t/\nu_t} \right)^{\tau-2}. \tag{A.10}$$

•• **Case 2:** $G_t(z)$ **has two identical real eigenvalues.** In this case, the eigenvalue of $G_t(z)$ is given by $\lambda = \text{Tr}_t(z)/2$ and $\text{Tr}_t(z)^2 - 4D_t(z) = 0$. The solution for the recursion (3.4) gives us

$$p_{t,\tau}(z) = (A + B\tau)\lambda^\tau.$$

We use the initial conditions to determine $A$ and $B$: $p_{t,0}(z) = A = 1$ and $p_{t,1}(z) = (A+B)\lambda = 1 - z$. By solving these equations, we have

$$A = 1, \qquad B = \frac{1 - z}{\lambda} - 1.$$

Combining the above two displays, we get the solution of $p_{t,\tau}(z)$ as

$$p_{t,\tau}(z) = \left( 1 + \left( \frac{1 - z}{\lambda} - 1 \right) \tau \right) \lambda^\tau. \tag{A.11}$$

By (3.5), $|\lambda| \leq \sup_{z \in [\mu_t, 1]} \rho(G_t(z)) \leq 1 - \sqrt{\mu_t/\nu_t}$ for all $t \geq 0$. Then (A.11) leads to the following derivation

$$\sup_{z \in [\mu_t, 1]} |p_{t,\tau}(z)| \stackrel{(A.11)}{\leq} \tau \lambda^\tau + \tau \lambda^{\tau-1} \stackrel{(A.11)}{\leq} 2\tau \left(1 - \sqrt{\mu_t/\nu_t}\right)^{\tau-1}. \tag{A.12}$$

•• **Case 3: $G_t(z)$ has two conjugate complex eigenvalues.** Let us denote the two conjugate complex eigenvalues of $G_t(z)$ by $\lambda_+ = r(\cos\theta + i\sin\theta)$ and $\lambda_- = r(\cos\theta - i\sin\theta)$ with $r = \sqrt{D_t(z)}$ and $\cos\theta = \text{Tr}_t(z)/(2\sqrt{D_t(z)})$, and $\text{Tr}_t(z)^2 - 4D_t(z) < 0$. The solution for the recursion (3.4) gives us

$$p_{t,\tau}(z) = \left(\sqrt{D_t(z)}\right)^\tau (A\cos(\tau\theta) + B\sin(\tau\theta)).$$

We use the initial conditions to determine $A$ and $B$: $p_{t,0}(z) = A = 1$ and $p_{t,1}(z) = \sqrt{D_t(z)}(A\cos(\theta) + B\sin(\theta)) = 1 - z$. By solving these equations, we have

$$A = 1, \qquad B = \frac{2(1-z) - \text{Tr}_t(z)}{\sqrt{4D_t(z) - \text{Tr}_t^2(z)}} = \frac{2(1-z) - \text{Tr}_t(z)}{2\sqrt{D_t(z)}\sin\theta}.$$

Combining the above two displays, we get the solution of $p_{t,\tau}(z)$ as

$$p_{t,\tau}(z) = \left(\sqrt{D_t(z)}\right)^\tau \left(\cos(\tau\theta) + \frac{2(1-z) - \text{Tr}_t(z)}{\sqrt{4D_t(z) - \text{Tr}_t(z)^2}} \sin(\tau\theta)\right)$$

$$= r^\tau \cos(\tau\theta) + r^{\tau-1} \left(1 - z - \frac{\text{Tr}_t(z)}{2}\right) \frac{\sin(\tau\theta)}{\sin\theta}. \tag{A.13}$$

Since $\frac{|\sin n\theta|}{|\sin\theta|} \leq n$ for all positive integers $n$ and any $\theta \in (0, \pi)$,[1] (A.13) leads to the following derivation

$$\sup_{z \in [\mu_t, 1]} |p_{t,\tau}(z)| \stackrel{(A.13)}{\leq} r^\tau + (1 - z - 0.5\text{Tr}_t(z))\tau r^{\tau-1} = r^\tau + (0.5 - 0.5z + 0.5\alpha_t\beta_t - 0.5\beta_t - 0.5z\alpha_t + 0.5z\alpha_t\gamma_t)\tau r^{\tau-1}$$

$$\leq r^\tau + (0.5 + 0.5\alpha_t\beta_t)\tau r^{\tau-1} \leq 2\tau r^{\tau-1} \stackrel{(3.5)}{\leq} 2\tau \left(1 - \sqrt{\mu_t/\nu_t}\right)^{\tau-1}, \tag{A.14}$$

where the second equality is due to $\text{Tr}_t(z) = 1 + \beta_t - \alpha_t\beta_t - z(1 - \alpha_t + \alpha_t\gamma_t)$; the third inequality is due to $\alpha_t\gamma_t \leq 1$ by (2.4) and (2.6); and the fourth inequality is due to $0 < \alpha_t < 1$ and $0 \leq \beta_t < 1$.

Combining the results of **Case 1**, **Case 2**, **Case 3** in (A.10), (A.12), and (A.14) completes the proof.

### A.3. Proof of Lemma 3.7

Recall that $C_t = \mathbb{E}[\widetilde{C}_t \mid \mathcal{F}_{t-1}]$ and $\widetilde{C}_t = \prod_{j=0}^{\tau-1} \widetilde{C}_{t,j}$, where $\widetilde{C}_{t,j}$ is defined in (A.1). By the independence of $\{S_{t,j}\}_j$, we have $C_t = (M_t)^\tau$, where

$$M_t = \begin{pmatrix} 1 - \alpha_t & \alpha_t \\ (1 - \alpha_t)(1 - \beta_t) & \alpha_t + \beta_t - \alpha_t\beta_t \end{pmatrix} \otimes I_d - \begin{pmatrix} 1 - \alpha_t & \alpha_t \\ (1 - \alpha_t)\gamma_t & \alpha_t\gamma_t \end{pmatrix} \otimes Z_t. \tag{A.15}$$

We diagonalize each block of $M_t$ and get the following diagonalized block matrix:

$$\widehat{M}_t = \Omega_t^\top M_t \Omega_t = \begin{pmatrix} U_t^\top [M_t]_{1,1} U_t & U_t^\top [M_t]_{1,2} U_t \\ U_t^\top [M_t]_{2,1} U_t & U_t^\top [M_t]_{2,2} U_t \end{pmatrix}$$

$$= \begin{pmatrix} (1 - \alpha_t)(I - \Lambda_t) & \alpha_t(I - \Lambda_t) \\ (1 - \alpha_t)(1 - \beta_t)I - (1 - \alpha_t)\gamma_t\Lambda_t & (\alpha_t + \beta_t - \alpha_t\beta_t)I - \alpha_t\gamma_t\Lambda_t \end{pmatrix}, \tag{A.16}$$

---

[1]Consider the summation $\sum_{k=0}^{n-1} e^{i(n-1-2k)\theta} = e^{i(n-1)\theta} \sum_{k=0}^{n-1} e^{-2ki\theta} = e^{i(n-1)\theta} \cdot \frac{1 - e^{-2in\theta}}{1 - e^{-2i\theta}} = e^{i(n-1)\theta} \cdot e^{-i(n-1)\theta} \cdot \frac{e^{in\theta} - e^{-in\theta}}{e^{i\theta} - e^{-i\theta}} = \frac{2i\sin n\theta}{2i\sin\theta} = \frac{\sin n\theta}{\sin\theta}$, where the last equality uses Euler's formula, i.e. $e^{i\theta} - e^{-i\theta} = 2i\sin\theta$. Therefore, for all positive integers $n$ and $\theta \in (0, \pi)$, we have $\frac{|\sin n\theta|}{|\sin\theta|} \leq \sum_{k=0}^{n-1} |e^{i(n-1-2k)\theta}| \leq n$.

where $\Omega_t = \begin{pmatrix} U_t & \mathbf{0} \\ \mathbf{0} & U_t \end{pmatrix}$, $U_t \in \mathbb{R}^{d \times d}$ is orthogonal with $U_t U_t^\top = U_t^\top U_t = I$, and $\Lambda_t = \mathrm{diag}(z_{t,1}, z_{t,2}, \ldots, z_{t,d}) \in \mathbb{R}^{d \times d}$ is a diagonal matrix with each $(i,i)$-th entry $z_{t,i}$ being the $i$-th eigenvalue of $Z_t$. Raising $\widehat{M}_t$ to the power $\tau$ yields

$$\widehat{M}_t^\tau = \Omega_t^\top M_t^\tau \Omega_t = \begin{pmatrix} U_t^\top [M_t^\tau]_{1,1} U_t & U_t^\top [M_t^\tau]_{1,2} U_t \\ U_t^\top [M_t^\tau]_{2,1} U_t & U_t^\top [M_t^\tau]_{2,2} U_t \end{pmatrix} = \begin{pmatrix} U_t^\top [C_t]_{1,1} U_t & U_t^\top [C_t]_{1,2} U_t \\ U_t^\top [C_t]_{2,1} U_t & U_t^\top [C_t]_{2,2} U_t \end{pmatrix}.$$

Therefore, by the definition of $K_t$, we know that

$$U_t^\top K_t U_t = [\widehat{M}_t^\tau]_{1,1} + [\widehat{M}_t^\tau]_{1,2}. \tag{A.17}$$

To proceed, we introduce the following lemma.

**Lemma A.1.** *Each block of $\widehat{M}_t^\tau$ is diagonal, that is*

$$[\widehat{M}_t^\tau]_{1,1} = \begin{pmatrix} [G_{t,1}^\tau]_{1,1} & 0 & \cdots & 0 \\ 0 & [G_{t,2}^\tau]_{1,1} & \cdots & 0 \\ \vdots & \vdots & \ddots & \vdots \\ 0 & 0 & \cdots & [G_{t,d}^\tau]_{1,1} \end{pmatrix}, \quad [\widehat{M}_t^\tau]_{1,2} = \begin{pmatrix} [G_{t,1}^\tau]_{1,2} & 0 & \cdots & 0 \\ 0 & [G_{t,2}^\tau]_{1,2} & \cdots & 0 \\ \vdots & \vdots & \ddots & \vdots \\ 0 & 0 & \cdots & [G_{t,d}^\tau]_{1,2} \end{pmatrix}, \tag{A.18}$$

*where we define for each $i \in [d] := \{1, \ldots, d\}$, $G_{t,i} := G_t(z_{t,i}) \in \mathbb{R}^{2 \times 2}$ (cf. (3.3) for the definition of $G_t(\cdot)$).*

By (A.17) and Lemma A.1, for all $t \geq 0$,

$$\|K_t\| = \max_{i \in [d]} |[G_{t,i}^\tau]_{1,1} + [G_{t,i}^\tau]_{1,2}| = \max_{i \in [d]} |[G_t(z_{t,i})^\tau]_{1,1} + [G_t(z_{t,i})^\tau]_{1,2}|$$

$$\leq \sup_{z \in [\mu_t, 1]} |p_{t,\tau}(z)| \leq 2\tau \left(1 - \sqrt{\mu_t/\nu_t}\right)^{\tau - 2},$$

where the third inequality is from the definition of $p_{t,\tau}(z)$ and the last inequality is from Lemma 3.6. This completes the proof.

### A.4. Proof of Lemma A.1

For simplicity, let us rewrite $\widehat{M}_t$ in (A.16) as

$$\widehat{M}_t = \begin{pmatrix} A_t^{(1)} & A_t^{(2)} \\ A_t^{(3)} & A_t^{(4)} \end{pmatrix}.$$

We note that the blocks $A_t^{(1)}, A_t^{(2)}, A_t^{(3)}$, and $A_t^{(4)}$ are all diagonal, expressed as $A_t^{(k)} = \mathrm{diag}(a_{t,1}^{(k)}, \ldots, a_{t,d}^{(k)}) \in \mathbb{R}^{d \times d}$ for $k = 1, 2, 3, 4$. Then, for each $i \in [d]$, we have

$$G_{t,i} = \begin{pmatrix} a_{t,i}^{(1)} & a_{t,i}^{(2)} \\ a_{t,i}^{(3)} & a_{t,i}^{(4)} \end{pmatrix}. \tag{A.19}$$

We prove the claim by induction on $\tau$.

- *Base case ($\tau = 1$).* By (A.16), the representation (A.18) holds trivially.

- *Induction step.* Assume that (A.18) holds for $\tau$. For $\tau + 1$, we have

$$\widehat{M}_t^{\tau+1} = \widehat{M}_t^\tau \widehat{M}_t = \begin{pmatrix} [\widehat{M}_t^\tau]_{1,1} & [\widehat{M}_t^\tau]_{1,2} \\ [\widehat{M}_t^\tau]_{2,1} & [\widehat{M}_t^\tau]_{2,2} \end{pmatrix} \begin{pmatrix} A_t^{(1)} & A_t^{(2)} \\ A_t^{(3)} & A_t^{(4)} \end{pmatrix}$$

$$= \begin{pmatrix} [\widehat{M}_t^\tau]_{1,1} A_t^{(1)} + [\widehat{M}_t^\tau]_{1,2} A_t^{(3)} & [\widehat{M}_t^\tau]_{1,1} A_t^{(2)} + [\widehat{M}_t^\tau]_{1,2} A_t^{(4)} \\ [\widehat{M}_t^\tau]_{2,1} A_t^{(1)} + [\widehat{M}_t^\tau]_{2,2} A_t^{(3)} & [\widehat{M}_t^\tau]_{2,1} A_t^{(2)} + [\widehat{M}_t^\tau]_{2,2} A_t^{(4)} \end{pmatrix}.$$

Thus, we have

$$[\widehat{M}_t^{\tau+1}]_{1,1} = [\widehat{M}_t^{\tau}]_{1,1} A_t^{(1)} + [\widehat{M}_t^{\tau}]_{1,2} A_t^{(3)}, \qquad [\widehat{M}_t^{\tau+1}]_{1,2} = [\widehat{M}_t^{\tau}]_{1,1} A_t^{(2)} + [\widehat{M}_t^{\tau}]_{1,2} A_t^{(4)}.$$

Since $[\widehat{M}_t^{\tau}]_{1,1}$ and $[\widehat{M}_t^{\tau}]_{1,2}$ are diagonal by the induction hypothesis, and $A_t^{(1)}, A_t^{(2)}, A_t^{(3)}, A_t^{(4)}$ are diagonal, we know both $[\widehat{M}_t^{\tau+1}]_{1,1}$ and $[\widehat{M}_t^{\tau+1}]_{1,2}$ are diagonal. Furthermore, for any $i \in [d]$,

$$([\widehat{M}_t^{\tau+1}]_{1,1})_{i,i} = ([\widehat{M}_t^{\tau}]_{1,1})_{i,i} a_{t,i}^{(1)} + ([\widehat{M}_t^{\tau}]_{1,2})_{i,i} a_{t,i}^{(3)}, \quad ([\widehat{M}_t^{\tau+1}]_{1,2})_{i,i} = ([\widehat{M}_t^{\tau}]_{1,1})_{i,i} a_{t,i}^{(2)} + ([\widehat{M}_t^{\tau}]_{1,2})_{i,i} a_{t,i}^{(4)}.$$

$$\text{(A.20)}$$

On the other hand, by the definition of $G_{t,i} = G_t(z_{t,i})$ in (3.3), we have

$$G_{t,i}^{\tau+1} = G_{t,i}^{\tau} G_{t,i} \stackrel{\text{(A.19)}}{=} \begin{pmatrix} [G_{t,i}^{\tau}]_{1,1} & [G_{t,i}^{\tau}]_{1,2} \\ [G_{t,i}^{\tau}]_{2,1} & [G_{t,i}^{\tau}]_{2,2} \end{pmatrix} \begin{pmatrix} a_{t,i}^{(1)} & a_{t,i}^{(2)} \\ a_{t,i}^{(3)} & a_{t,i}^{(4)} \end{pmatrix}.$$

Thus, for each $i \in [d]$,

$$[G_{t,i}^{\tau+1}]_{1,1} = [G_{t,i}^{\tau}]_{1,1} a_{t,i}^{(1)} + [G_{t,i}^{\tau}]_{1,2} a_{t,i}^{(3)}, \qquad [G_{t,i}^{\tau+1}]_{1,2} = [G_{t,i}^{\tau}]_{1,1} a_{t,i}^{(2)} + [G_{t,i}^{\tau}]_{1,2} a_{t,i}^{(4)}. \qquad \text{(A.21)}$$

By the induction hypothesis, $[G_{t,i}^{\tau}]_{1,1} = ([\widehat{M}_t^{\tau}]_{1,1})_{i,i}$ and $[G_{t,i}^{\tau}]_{1,2} = ([\widehat{M}_t^{\tau}]_{1,2})_{i,i}$. Substituting these identities into (A.20) and (A.21) yields

$$([\widehat{M}_t^{\tau+1}]_{1,1})_{i,i} = [G_{t,i}^{\tau+1}]_{1,1}, \qquad ([\widehat{M}_t^{\tau+1}]_{1,2})_{i,i} = [G_{t,i}^{\tau+1}]_{1,2}, \quad \forall\, i \in [d].$$

This establishes (A.18) for $\tau + 1$ and completes the induction.

## A.5. Proof of Theorem 3.8

By Assumption 3.1, we know $f(\boldsymbol{x})$ is strongly convex. By the basic property of strong convexity (Nesterov, 2018), we have

$$\frac{\gamma_H}{2} \|\boldsymbol{x}_t - \boldsymbol{x}^\star\|^2 \le f_t - f^\star \le \frac{1}{2\gamma_H} \|\nabla f_t\|^2. \qquad \text{(A.22)}$$

Still, by the $\Upsilon_H$-Lipschitz continuity of $\nabla f(\boldsymbol{x})$ implied by Assumption 3.2, we know from Nesterov (2018) that

$$\frac{1}{2\Upsilon_H} \|\nabla f_t\|^2 \le f_t - f^\star \le \frac{\Upsilon_H}{2} \|\boldsymbol{x}_t - \boldsymbol{x}^\star\|^2. \qquad \text{(A.23)}$$

Furthermore, by the $\Upsilon_H$-Lipschitz continuity of $\nabla f(\boldsymbol{x})$, we have the Taylor expansion

$$f_{t+1} - f^\star \le (f_t - f^\star) + \varphi_t \nabla f_t^\top \boldsymbol{z}_{t,\tau} + \frac{\Upsilon_H}{2} \varphi_t^2 \|\boldsymbol{z}_{t,\tau}\|^2. \qquad \text{(A.24)}$$

We then take the conditional expectation $\mathbb{E}[\cdot \mid \mathcal{F}_{t-1}]$ on both sides of (A.24) and obtain

$$\mathbb{E}[f_{t+1} - f^\star \mid \mathcal{F}_{t-1}] \le (f_t - f^\star) + \varphi_t \mathbb{E}[\nabla f_t^\top \boldsymbol{z}_{t,\tau} \mid \mathcal{F}_{t-1}] + \frac{\Upsilon_H}{2} \varphi_t^2 \mathbb{E}[\|\boldsymbol{z}_{t,\tau}\|^2 \mid \mathcal{F}_{t-1}]. \qquad \text{(A.25)}$$

Also, by Assumption 3.1 as well as the construction of the Hessian approximate $B_t$, we have

$$\gamma_H \le \lambda_{\min}(B_t) \le \lambda_{\max}(B_t) \le \Upsilon_H. \qquad \text{(A.26)}$$

For the second term on the right-hand side of (A.25), we obtain

$$\begin{aligned} \varphi_t \mathbb{E}[\nabla f_t^\top \boldsymbol{z}_{t,\tau} \mid \mathcal{F}_{t-1}] &= \varphi_t \nabla f_t^\top \mathbb{E}[\boldsymbol{z}_{t,\tau} \mid \mathcal{F}_{t-1}] = -\varphi_t \nabla f_t^\top (I - K_t) B_t^{-1} \nabla f_t \\ &= -\varphi_t \nabla f_t^\top B_t^{-1} \nabla f_t + \varphi_t \nabla f_t^\top K_t B_t^{-1} \nabla f_t \\ &\stackrel{\text{(A.26)}}{\le} -\frac{\varphi_t}{\Upsilon_H} \|\nabla f_t\|^2 + \frac{\varphi_t}{\gamma_H} \|\nabla f_t\|^2 \|K_t\| \le -\frac{\varphi_t}{2\Upsilon_H} \|\nabla f_t\|^2, \end{aligned} \qquad \text{(A.27)}$$

where the second equality follows from Lemma 3.5 and the last inequality follows from Lemma 3.7 as well as the as the condition on $\tau$ such that $\|K_t\| \leq \gamma_H/2\Upsilon_H$. To proceed, we establish a uniform bound for $\|\widetilde{K}_t\|$. In particular, we have

$$
\begin{aligned}
\|\widetilde{K}_t\| &\overset{(3.2)}{=} \left\| \begin{pmatrix} I & \mathbf{0} \end{pmatrix} \widetilde{C}_t \begin{pmatrix} I \\ I \end{pmatrix} \right\| \leq \sqrt{2}\|\widetilde{C}_t\| \leq \sqrt{2} \cdot \prod_{j=0}^{\tau-1} \|\widetilde{C}_{t,j}\| \\
&\leq \sqrt{2} \cdot \prod_{j=0}^{\tau-1} \left( \|I - \widetilde{Z}_{t,j}\| + (1-\alpha_t)(1-\beta_t) + (1-\alpha_t)\gamma_t\|\widetilde{Z}_{t,j}\| + (\alpha_t+\beta_t-\alpha_t\beta_t) + \alpha_t\gamma_t\|\widetilde{Z}_{t,j}\| \right) \\
&\leq \sqrt{2} \cdot \prod_{j=0}^{\tau-1} \left( \|I - \widetilde{Z}_{t,j}\| + 1 + \gamma_t\|\widetilde{Z}_{t,j}\| \right) \\
&\leq \sqrt{2} \cdot \prod_{j=0}^{\tau-1} (1 + 1 + \gamma_t) \overset{(A.8)}{\leq} \sqrt{2}(2 + 1/\sqrt{\gamma_S})^\tau =: C_K.
\end{aligned}
\tag{A.28}
$$

We now turn to the third term on the right-hand side of (A.25), for which we have

$$
\begin{aligned}
\frac{\Upsilon_H}{2}\varphi_t^2 \mathbb{E}[\|\boldsymbol{z}_{t,\tau}\|^2 \mid \mathcal{F}_{t-1}] &\leq \frac{\Upsilon_H}{2}\varphi_t^2 \mathbb{E}[\|I - \widetilde{K}_t\|^2\|B_t^{-1}\|^2\|g_t\|^2 \mid \mathcal{F}_{t-1}] \quad \text{(by Lemma 3.5)} \\
&\overset{\substack{(A.26)\\(A.28)}}{\leq} \frac{\Upsilon_H}{2}\varphi_t^2 \frac{(1+C_K)^2}{\gamma_H^2} \mathbb{E}[\|g_t\|^2 \mid \mathcal{F}_{t-1}] \leq \frac{\Upsilon_H(1+C_K)^2}{2\gamma_H^2}\varphi_t^2 \left( \|\nabla f_t\|^2 + \mathbb{E}[\|g_t - \nabla f_t\|^2 \mid \mathcal{F}_{t-1}] \right) \\
&\leq \frac{\Upsilon_H(1+C_K)^2}{2\gamma_H^2}\varphi_t^2\|\nabla f_t\|^2 + \frac{\Upsilon_H(1+C_K)^2}{2\gamma_H^2}\varphi_t^2 \left( C_{g,1}\|\boldsymbol{x}_t - \boldsymbol{x}^\star\|^2 + C_{g,2} \right) \\
&\overset{(A.22)}{\leq} \frac{\Upsilon_H(1+C_K)^2}{2\gamma_H^2}\varphi_t^2\|\nabla f_t\|^2 + \frac{\Upsilon_H(1+C_K)^2 C_{g,1}}{2\gamma_H^4}\varphi_t^2\|\nabla f_t\|^2 + \frac{\Upsilon_H(1+C_K)^2 C_{g,2}}{2\gamma_H^2}\varphi_t^2 \\
&= \frac{(\gamma_H^2 + C_{g,1})\Upsilon_H(1+C_K)^2}{2\gamma_H^4}\varphi_t^2\|\nabla f_t\|^2 + \frac{\Upsilon_H(1+C_K)^2 C_{g,2}}{2\gamma_H^2}\varphi_t^2,
\end{aligned}
\tag{A.29}
$$

where the first inequality follows Lemma 3.5, and the fourth inequality follows from Assumption 3.2 with $q_g = 2$. Substituting (A.27) and (A.29) into (A.25) gives us

$$
\mathbb{E}[f_{t+1} - f^\star \mid \mathcal{F}_{t-1}] \leq f_t - f^\star - \frac{\varphi_t}{2\Upsilon_H}\|\nabla f_t\|^2 + \frac{(\gamma_H^2 + C_{g,1})\,\Upsilon_H(1+C_K)^2}{2\gamma_H^4}\varphi_t^2\|\nabla f_t\|^2 + \frac{\Upsilon_H(1+C_K)^2 C_{g,2}}{2\gamma_H^2}\varphi_t^2.
$$

Since $\varphi_t = C_\varphi/(t+1)^\varphi$ and $\varphi \in (1/2, 1]$, there exists $t_0 > 0$ such that for all $t \geq t_0$, $\frac{(\gamma_H^2+C_{g,1})\Upsilon_H(1+C_K)^2}{2\gamma_H^4}\varphi_t \leq \frac{1}{4\Upsilon_H}$. Then, for all $t \geq t_0$, we have

$$
\mathbb{E}[f_{t+1} - f^\star \mid \mathcal{F}_{t-1}] \leq f_t - f^\star - \frac{\varphi_t}{4\Upsilon_H}\|\nabla f_t\|^2 + \frac{\Upsilon_H(1+C_K)^2 C_{g,2}}{2\gamma_H^2}\varphi_t^2.
\tag{A.30}
$$

Since $\sum_t \varphi_t = \infty$ and $\sum_t \varphi_t^2 < \infty$, we apply the Robbins-Siegmund Theorem (Duflo, 2013, Theorem 1.3.12), and conclude that $f_t - f^\star$ converges to a finite random variable, and $\sum_{t=t_0}^\infty \varphi_t\|\nabla f_t\|^2 < \infty$ almost surely. This implies that $\liminf_{t\to\infty} \|\nabla f_t\| = 0$ almost surely; and thus $\liminf_{t\to\infty}(f_t - f^\star) = 0$ almost surely by (A.22). Since we have already known that $f_t - f^\star$ converges almost surely, the conclusion can be strengthened to $\lim_{t\to\infty} f_t - f^\star = 0$. Again, we apply (A.22) and obtain $\lim_{t\to\infty} \boldsymbol{x}_t = \boldsymbol{x}^\star$ almost surely. We complete the proof of Theorem 3.8.

## B. Proofs of Section 4

Throughout the section, we exchangeably denote $a_t = O(b_t)$ and $a_t \lesssim b_t$ to ease notation. We also use $\mathbf{1}_{\{\cdot\}}$ to denote the indicator function.

## B.1. Proof of Lemma 4.1

• **Convergence of $B_t$.** We first establish the almost-sure convergence of $B_t$. By the definition of $B_t$ in (2.1), we have

$$\|B_t - B^\star\| = \left\|\frac{1}{t}\sum_{i=0}^{t-1}(H_i - \nabla^2 f_i) + \frac{1}{t}\sum_{i=0}^{t-1}(\nabla^2 f_i - \nabla^2 f^\star)\right\|$$

$$\leq \left\|\frac{1}{t}\sum_{i=0}^{t-1}(H_i - \nabla^2 f_i)\right\| + \left\|\frac{1}{t}\sum_{i=0}^{t-1}(\nabla^2 f_i - \nabla^2 f^\star)\right\|$$

$$\leq \left\|\frac{1}{t}\sum_{i=0}^{t-1}(H_i - \nabla^2 f_i)\right\| + \frac{\Upsilon_L}{t}\sum_{i=0}^{t-1}\|\boldsymbol{x}_i - \boldsymbol{x}^\star\|, \tag{B.1}$$

where the last inequality follows from Assumption 3.1. The second term on the right-hand side of (B.1) converges to 0 almost surely, which is the direct consequence of Theorem 3.8 together with the Stolz–Cesàro theorem. For the first term on the right-hand side of (B.1), we note from Assumption 3.1 that $\mathbb{E}[\|H_i - \nabla^2 f_i\|_F^2] \lesssim 1$. Since $\mathbb{E}[H_t|\mathcal{F}_{t-1}] = \nabla^2 f_t$, we know $(H_i - \nabla^2 f_i, \mathcal{F}_i)_{i\geq 0}$ is a martingale difference sequence. Thus, $(\sum_{i=0}^{t-1}(H_i - \nabla^2 f_i)/(i+1), \mathcal{F}_{t-1})_{t\geq 1}$ is a martingale satisfying

$$\mathbb{E}\left[\left\|\sum_{i=0}^{t-1}\frac{1}{i+1}(H_i - \nabla^2 f_i)\right\|^2\right] \leq \mathbb{E}\left[\left\|\sum_{i=0}^{t-1}\frac{1}{i+1}(H_i - \nabla^2 f_i)\right\|_F^2\right] = \sum_{i=0}^{t-1}\mathbb{E}\left[\frac{1}{(i+1)^2}\|H_i - \nabla^2 f_i\|_F^2\right] < \infty,$$

where the equality uses the orthogonality of martingale differences under the Frobenius inner product, so that all cross terms have zero expectation. The above display suggests that $(\sum_{i=0}^{t-1}(H_i - \nabla^2 f_i)/(i+1))_{t\geq 1}$ is $L^2$-bounded, and hence $\sum_{i=0}^{t-1}\frac{1}{i+1}(H_i - \nabla^2 f_i)$ converges almost surely by Doob's martingale convergence theorem (Hall & Heyde, 2014, Corollary 2.2). Applying Kronecker's Lemma gives us $\frac{1}{t}\sum_{i=0}^{t-1}(H_i - \nabla^2 f_i) \overset{a.s.}{\to} 0$ as $t \to \infty$. This shows that both two terms in (B.1) converge to 0 almost surely; thus, $B_t \overset{a.s.}{\to} B^\star$ as $t \to \infty$.

• **Convergence of $(\alpha_t, \beta_t, \gamma_t)$.** Recalling the definitions of $\alpha, \beta, \gamma$ in (2.4) (the distinction between $(\alpha_t, \beta_t, \gamma_t)$ and $(\alpha^\star, \beta^\star, \gamma^\star)$ is solely due to the evaluation at $(\mu_t, \nu_t)$ v.s. $(\mu^\star, \nu^\star)$), we regard them as functions of $(\mu, \nu)$ as follows:

$$\alpha(\mu, \nu) = \frac{1}{1+\gamma\nu}, \quad \beta(\mu, \nu) = 1 - \sqrt{\frac{\mu}{\nu}}, \quad \gamma(\mu, \nu) = \frac{1}{\sqrt{\mu\nu}}.$$

Each of $\alpha, \beta, \gamma$ is continuously differentiable on $(0, \infty)^2$. As shown in (2.6) and Assumption 3.4, $(\mu_t, \nu_t) \in [\gamma_S, 1] \times [1, 1/\gamma_S]$, $\forall t \geq 0$. Since $\nabla\alpha, \nabla\beta, \nabla\gamma$ are continuous, $\nabla\alpha(\mu, \nu), \nabla\beta(\mu, \nu), \nabla\gamma(\mu, \nu)$ are bounded on the compact set $[\gamma_S, 1] \times [1, 1/\gamma_S]$. Moreover, since $[\gamma_S, 1] \times [1, 1/\gamma_S]$ is convex, we know from the mean value theorem that

$$|\alpha_t - \alpha^\star| \lesssim |\mu_t - \mu^\star| + |\nu_t - \nu^\star|, \quad |\beta_t - \beta^\star| \lesssim |\mu_t - \mu^\star| + |\nu_t - \nu^\star|, \quad |\gamma_t - \gamma^\star| \lesssim |\mu_t - \mu^\star| + |\nu_t - \nu^\star|. \tag{B.2}$$

Thus, it suffices to show $(\mu_t, \nu_t) \overset{a.s.}{\to} (\mu^\star, \nu^\star)$. To that end, let us start with defining several matrices as follows:

$$\widetilde{Z}_t := B_t S(S^\top B_t^2 S)^\dagger S^\top B_t, \qquad Z_t := \mathbb{E}[\widetilde{Z}_t \mid \mathcal{F}_{t-1}],$$
$$\widetilde{Z}^\star := B^\star S(S^\top (B^\star)^2 S)^\dagger S^\top B^\star, \qquad Z^\star := \mathbb{E}[\widetilde{Z}^\star]. \tag{B.3}$$

We next state a lemma characterizing the continuity of the projection matrix $\widetilde{Z}_t = B_t S(S^\top B_t^2 S)^\dagger S^\top B_t$ with respect to the Hessian estimate $B_t$.

**Lemma B.1.** *(Na & Mahoney, 2025, Lemma 5.2) Suppose $B_t, B^\star \in \mathbb{R}^{d\times d}$ are non-singular. Then for any $S \in \mathbb{R}^{d\times s}$, we have*

$$\|\widetilde{Z}_t - \widetilde{Z}^\star\| \leq \frac{2\|B_t - B^\star\|}{\sigma_{\min}(B^\star)} \cdot \|S\|\|S^\dagger\|,$$

*where $\sigma_{\min}(\cdot)$ denotes the least singular value.*

By Lemma B.1, the discrepancy between the projection matrices induced by the Hessian estimates can be bounded as

$$\mathbb{E}[\|\widetilde{Z}_t - \widetilde{Z}^\star\| \mid \mathcal{F}_{t-1}] \leq \frac{2\|B_t - B^\star\|}{\gamma_H}\mathbb{E}[\|S\| \|S^\dagger\|] \lesssim \|B_t - B^\star\|. \tag{B.4}$$

Here, the first inequality follows from Lemma B.1 together with the independence between $S$ and $\mathcal{F}_{t-1}$ implied by Assumption 3.4, while the second one is a direct consequence of Assumption 3.4 with $q_S = 1$.

•• **Convergence of $\mu_t$.** By the definitions of $\mu_t$ in (2.5) and $\mu^\star$ in Section 4.1, we have $\mu_t = \lambda_{\min}(Z_t)$ and $\mu^\star = \lambda_{\min}(Z^\star)$. Applying Weyl's perturbation theorem (Bhatia, 1997, Corollary III.2.6), we obtain

$$|\mu_t - \mu^\star| \leq \|Z_t - Z^\star\| \overset{(B.3)}{=} \|\mathbb{E}[\widetilde{Z}_t - \widetilde{Z}^\star \mid \mathcal{F}_{t-1}]\| \leq \mathbb{E}[\|\widetilde{Z}_t - \widetilde{Z}^\star\| \mid \mathcal{F}_{t-1}] \overset{(B.4)}{\lesssim} \|B_t - B^\star\|, \tag{B.5}$$

where the third inequality follows from Jensen's inequality. Since $B_t \overset{a.s.}{\to} B^\star$, we conclude that $\mu_t \overset{a.s.}{\to} \mu^\star$ as $t \to \infty$.

•• **Convergence of $\nu_t$.** Recall the definition of $\nu_t$ in (2.5) and define $A_t := \mathbb{E}[\widetilde{Z}_t Z_t^{-1} \widetilde{Z}_t \mid \mathcal{F}_{t-1}]$. The matrix $A_t$ is symmetric positive semidefinite, since $Z_t^{-1} \succeq I$ and both $\widetilde{Z}_t$ and $Z_t$ are symmetric. Letting $\boldsymbol{b} = Z_t^{1/2} \boldsymbol{a}$, we obtain

$$\begin{aligned} \nu_t &\overset{(2.5)}{=} \sup_{\boldsymbol{a} \in \mathbb{R}^d \setminus \{\boldsymbol{0}\}} \frac{\langle A_t \boldsymbol{a}, \boldsymbol{a} \rangle}{\langle Z_t \boldsymbol{a}, \boldsymbol{a} \rangle} = \sup_{\boldsymbol{b} \in \mathbb{R}^d \setminus \{\boldsymbol{0}\}} \frac{\langle A_t Z_t^{-1/2} \boldsymbol{b}, Z_t^{-1/2} \boldsymbol{b} \rangle}{\langle Z_t Z_t^{-1/2} \boldsymbol{b}, Z_t^{-1/2} \boldsymbol{b} \rangle} \\ &= \sup_{\boldsymbol{b} \in \mathbb{R}^d \setminus \{\boldsymbol{0}\}} \frac{\langle Z_t^{-1/2} A_t Z_t^{-1/2} \boldsymbol{b}, \boldsymbol{b} \rangle}{\langle \boldsymbol{b}, \boldsymbol{b} \rangle} = \lambda_{\max}(Z_t^{-1/2} A_t Z_t^{-1/2}) = \|Z_t^{-1/2} A_t Z_t^{-1/2}\|, \end{aligned}$$

where $\lambda_{\max}(\cdot)$ denotes the largest eigenvalue. Applying the same argument to $\nu^\star$ yields $\nu^\star = \|(Z^\star)^{-1/2} A^\star (Z^\star)^{-1/2}\|$ with $A^\star := \mathbb{E}[\widetilde{Z}^\star (Z^\star)^{-1} \widetilde{Z}^\star]$, where the expectation is taken only over the sketching distribution $S$. Consequently, we have

$$\begin{aligned} |\nu_t - \nu^\star| &= \left| \|Z_t^{-1/2} A_t Z_t^{-1/2}\| - \|(Z^\star)^{-1/2} A^\star (Z^\star)^{-1/2}\| \right| \\ &\leq \|Z_t^{-1/2} A_t Z_t^{-1/2} - (Z^\star)^{-1/2} A^\star (Z^\star)^{-1/2}\| \\ &\leq \|(Z_t^{-1/2} - (Z^\star)^{-1/2}) A_t Z_t^{-1/2}\| + \|(Z^\star)^{-1/2}(A_t - A^\star) Z_t^{-1/2}\| + \|(Z^\star)^{-1/2} A^\star (Z_t^{-1/2} - (Z^\star)^{-1/2})\|. \end{aligned} \tag{B.6}$$

To control the three terms on the right-hand side of (B.6), we first introduce the following preparation lemma.

**Lemma B.2.** *Under the conditions of Lemma 4.1, we have*

$$\|Z_t^{-1} - (Z^\star)^{-1}\| \leq \frac{1}{\gamma_S^2} \|Z_t - Z^\star\|, \qquad \|Z_t^{-1/2} - (Z^\star)^{-1/2}\| \leq \frac{1}{2\gamma_S^{3/2}} \|Z_t - Z^\star\|.$$

**First term in** (B.6). By Assumption 3.4 and the fact that $\widetilde{Z}_t$ is a projection matrix, we have $\|A_t\| \leq \gamma_S^{-1}$ and $\|Z_t^{-1/2}\| \leq \gamma_S^{-1/2}$. Therefore,

$$\begin{aligned} \|(Z_t^{-1/2} - (Z^\star)^{-1/2}) A_t Z_t^{-1/2}\| &\leq \|Z_t^{-1/2} - (Z^\star)^{-1/2}\| \|A_t\| \|Z_t^{-1/2}\| \\ &\leq \frac{1}{\gamma_S} \frac{1}{\sqrt{\gamma_S}} \frac{1}{2\gamma_S^{3/2}} \|Z_t - Z^\star\| = \frac{1}{2\gamma_S^3} \|Z_t - Z^\star\|, \end{aligned} \tag{B.7}$$

where the second last inequality follows from Lemma B.2.

**Third term in** (B.6). Similarly, since $\|A^\star\| \leq \gamma_S^{-1}$ and $\|(Z^\star)^{-1/2}\| \leq \gamma_S^{-1/2}$, we again apply Lemma B.2 and obtain

$$\|(Z^\star)^{-1/2} A^\star (Z_t^{-1/2} - (Z^\star)^{-1/2})\| \leq \frac{1}{\sqrt{\gamma_S}} \frac{1}{\gamma_S} \frac{1}{2\gamma_S^{3/2}} \|Z_t - Z^\star\| = \frac{1}{2\gamma_S^3} \|Z_t - Z^\star\|. \tag{B.8}$$

**Second term in** (B.6). Using $\|(Z^\star)^{-1/2}\| \leq \gamma_S^{-1/2}$ and $\|Z_t^{-1/2}\| \leq \gamma_S^{-1/2}$, we have

$$
\begin{aligned}
\|(Z^\star)^{-1/2}(A_t - A^\star)Z_t^{-1/2}\| &\leq \frac{1}{\gamma_S}\|A_t - A^\star\| \\
&= \frac{1}{\gamma_S}\left\|\mathbb{E}[\widetilde{Z}_t Z_t^{-1}\widetilde{Z}_t \mid \mathcal{F}_{t-1}] - \mathbb{E}[\widetilde{Z}^\star(Z^\star)^{-1}\widetilde{Z}^\star]\right\| \\
&= \frac{1}{\gamma_S}\left\|\mathbb{E}[\widetilde{Z}_t Z_t^{-1}\widetilde{Z}_t - \widetilde{Z}^\star(Z^\star)^{-1}\widetilde{Z}^\star \mid \mathcal{F}_{t-1}]\right\| \\
&= \frac{1}{\gamma_S}\left\|\mathbb{E}[(\widetilde{Z}_t - \widetilde{Z}^\star)Z_t^{-1}\widetilde{Z}_t + \widetilde{Z}^\star(Z_t^{-1} - (Z^\star)^{-1})\widetilde{Z}_t + \widetilde{Z}^\star(Z^\star)^{-1}(\widetilde{Z}_t - \widetilde{Z}^\star) \mid \mathcal{F}_{t-1}]\right\| \\
&\leq \frac{1}{\gamma_S}(\mathbb{E}[\|(\widetilde{Z}_t - \widetilde{Z}^\star)Z_t^{-1}\widetilde{Z}_t\| \mid \mathcal{F}_{t-1}] + \mathbb{E}[\|\widetilde{Z}^\star(Z_t^{-1} - (Z^\star)^{-1})\widetilde{Z}_t\| \mid \mathcal{F}_{t-1}] + \mathbb{E}[\|\widetilde{Z}^\star(Z^\star)^{-1}(\widetilde{Z}_t - \widetilde{Z}^\star)\| \mid \mathcal{F}_{t-1}]) \\
&\leq \frac{1}{\gamma_S}\left(\frac{1}{\gamma_S}\mathbb{E}[\|\widetilde{Z}_t - \widetilde{Z}^\star\| \mid \mathcal{F}_{t-1}] + \|Z_t^{-1} - (Z^\star)^{-1}\| + \frac{1}{\gamma_S}\mathbb{E}[\|\widetilde{Z}_t - \widetilde{Z}^\star\| \mid \mathcal{F}_{t-1}]\right) \\
&\leq \frac{1}{\gamma_S}\left(\frac{1}{\gamma_S}\mathbb{E}[\|\widetilde{Z}_t - \widetilde{Z}^\star\| \mid \mathcal{F}_{t-1}] + \frac{1}{\gamma_S^2}\|Z_t - Z^\star\| + \frac{1}{\gamma_S}\mathbb{E}[\|\widetilde{Z}_t - \widetilde{Z}^\star\| \mid \mathcal{F}_{t-1}]\right) \\
&= \frac{1}{\gamma_S^3}\|Z_t - Z^\star\| + \frac{2}{\gamma_S^2}\mathbb{E}[\|\widetilde{Z}_t - \widetilde{Z}^\star\| \mid \mathcal{F}_{t-1}]. \tag{B.9}
\end{aligned}
$$

Plugging (B.7), (B.8), (B.9) into (B.6) gives us

$$
\begin{aligned}
|\nu_t - \nu^\star| &\leq \frac{2}{\gamma_S^3}\|Z_t - Z^\star\| + \frac{2}{\gamma_S^2}\mathbb{E}[\|\widetilde{Z}_t - \widetilde{Z}^\star\| \mid \mathcal{F}_{t-1}] \\
&= \frac{2}{\gamma_S^3}\left\|\mathbb{E}[\widetilde{Z}_t - \widetilde{Z}^\star \mid \mathcal{F}_{t-1}]\right\| + \frac{2}{\gamma_S^2}\mathbb{E}[\|\widetilde{Z}_t - \widetilde{Z}^\star\| \mid \mathcal{F}_{t-1}] \\
&\leq \frac{2}{\gamma_S^3}\mathbb{E}[\|\widetilde{Z}_t - \widetilde{Z}^\star\| \mid \mathcal{F}_{t-1}] + \frac{2}{\gamma_S^2}\mathbb{E}[\|\widetilde{Z}_t - \widetilde{Z}^\star\| \mid \mathcal{F}_{t-1}] \\
&\lesssim \mathbb{E}[\|\widetilde{Z}_t - \widetilde{Z}^\star\| \mid \mathcal{F}_{t-1}] \overset{(B.4)}{\lesssim} \|B_t - B^\star\|. \tag{B.10}
\end{aligned}
$$

Since $B_t \overset{a.s.}{\to} B^\star$, we conclude that $\nu_t \overset{a.s.}{\to} \nu^\star$ as $t \to \infty$. Combining (B.5) and (B.10) with (B.2) yields

$$
|\alpha_t - \alpha^\star| \lesssim \|B_t - B^\star\|, \qquad |\beta_t - \beta^\star| \lesssim \|B_t - B^\star\|, \qquad |\gamma_t - \gamma^\star| \lesssim \|B_t - B^\star\|, \tag{B.11}
$$

which implies that $(\alpha_t, \beta_t, \gamma_t) \overset{a.s.}{\to} (\alpha^\star, \beta^\star, \gamma^\star)$ as $t \to \infty$.

• **Convergence of $K_t$.** Recall the constructions of $K_t$ and $K^\star$ from $C_t$ and $C^\star$ in (3.2). To establish the convergence of $K_t$, it suffices to study the convergence of $C_t$. Due to the independence among the sketching matrices, both $C_t$ and $C^\star$ admit the formulas $C_t = (M_t)^\tau$ and $C^\star = (M^\star)^\tau$, where $M_t$ is defined in (A.15) and $M^\star$ is defined analogously by replacing $\alpha_t, \beta_t, \gamma_t$ and $Z_t$ with their counterparts evaluated at $\boldsymbol{x}^\star$. We first bound $\|M_t - M^\star\|$ as follows:

$$
\begin{aligned}
\|M_t - M^\star\| &\leq \|(1 - \alpha_t)(I - Z_t) - (1 - \alpha^\star)(I - Z^\star)\| + \|\alpha_t(I - Z_t) - \alpha^\star(I - Z^\star)\| \\
&\quad + \|(1 - \alpha_t)(1 - \beta_t)I - (1 - \alpha_t)\gamma_t Z_t - (1 - \alpha^\star)(1 - \beta^\star)I + (1 - \alpha^\star)\gamma^\star Z^\star\| \\
&\quad + \|(\alpha_t + \beta_t - \alpha_t\beta_t)I - \alpha_t\gamma_t Z_t - (\alpha^\star + \beta^\star - \alpha^\star\beta^\star)I + \alpha^\star\gamma^\star Z^\star\| \\
&\lesssim |\alpha_t - \alpha^\star| + |\beta_t - \beta^\star| + |\gamma_t - \gamma^\star| + \|Z_t - Z^\star\| \\
&\overset{(B.11)}{\lesssim} \|B_t - B^\star\| + \mathbb{E}[\|\widetilde{Z}_t - \widetilde{Z}^\star\| \mid \mathcal{F}_{t-1}] \overset{(B.4)}{\lesssim} \|B_t - B^\star\|. \tag{B.12}
\end{aligned}
$$

Here, the second inequality uses the bounds $\alpha_t < 1$, $\beta_t < 1$, $\gamma_t \leq 1/\sqrt{\gamma_S}$, and $\|Z_t\| \leq 1$, and the third inequality uses the fact that $\|Z_t - Z^\star\| = \|\mathbb{E}[\widetilde{Z}_t - \widetilde{Z}^\star \mid \mathcal{F}_{t-1}]\| \leq \mathbb{E}[\|\widetilde{Z}_t - \widetilde{Z}^\star\| \mid \mathcal{F}_{t-1}]$. Combining (B.12) with a telescoping expansion, we obtain

$$
\|C_t - C^\star\| = \|(M_t)^\tau - (M^\star)^\tau\| = \left\|\sum_{k=0}^{\tau-1} M_t^{\tau-1-k}(M_t - M^\star)(M^\star)^k\right\| \leq \left(\sum_{k=0}^{\tau-1}\|M_t\|^{\tau-1-k}\|M^\star\|^k\right)\|M_t - M^\star\|.
$$

For $M_t$, we find the following uniform upper bound:

$$\|M_t\| \overset{(A.15)}{\leq} \|I - Z_t\| + (1-\alpha_t)(1-\beta_t) + (1-\alpha_t)\gamma_t\|Z_t\| + (\alpha_t + \beta_t - \alpha_t\beta_t) + \alpha_t\gamma_t\|Z_t\|$$

$$\leq \|I - Z_t\| + 1 + \gamma_t\|Z_t\| \leq 1 + 1 + \gamma_t \overset{(A.8)}{\leq} 2 + 1/\sqrt{\gamma_S}.$$

By the almost-sure convergence of $B_t$, $Z_t$, and $(\alpha_t, \beta_t, \gamma_t)$, the same bound holds for $\|M^\star\|$. Consequently, the above two displays lead to

$$\|C_t - C^\star\| \leq (2 + 1/\sqrt{\gamma_S})^{\tau-1} \cdot \tau \cdot \|M_t - M^\star\| \lesssim \|M_t - M^\star\| \overset{(B.12)}{\lesssim} \|B_t - B^\star\|. \tag{B.13}$$

Finally, by the definitions of $K_t$ and $K^\star$, we have

$$\|K_t - K^\star\| = \left\| (I \quad 0)(C_t - C^\star)\begin{pmatrix} I \\ I \end{pmatrix} \right\| \leq \sqrt{2} \cdot \|C_t - C^\star\| \overset{(B.13)}{\lesssim} \|B_t - B^\star\|. \tag{B.14}$$

Since $B_t \overset{a.s.}{\to} B^\star$, we conclude that $K_t \overset{a.s.}{\to} K^\star$ as $t \to \infty$.

## B.2. Proof of Lemma B.2

For the first result, we have

$$\|Z_t^{-1} - (Z^\star)^{-1}\| = \|Z_t^{-1}(Z_t - Z^\star)(Z^\star)^{-1}\| \leq \|Z_t^{-1}\|\|Z_t - Z^\star\|\|(Z^\star)^{-1}\| \leq \frac{1}{\gamma_S^2}\|Z_t - Z^\star\|,$$

where the last inequality follows from $Z_t \succeq \gamma_S I$ and $Z^\star \succeq \gamma_S I$ in Assumption 3.4, together with the almost sure convergence $Z_t \overset{a.s.}{\longrightarrow} Z^\star$ established in Theorem 3.8 and Lemma B.1.

For the second result, we proceed in two steps.

• **Step 1.** We show that for any positive definite matrix $X$,

$$X^{-1/2} = \frac{1}{\pi}\int_0^\infty t^{-1/2}(X + tI)^{-1}\,dt.$$

Indeed, for any scalar $x > 0$, the following identity holds

$$x^{-1/2} = \frac{1}{\pi}\int_0^\infty t^{-1/2}(x + t)^{-1}\,dt.$$

To verify this, let $t = xu^2$, so that $dt = 2xu\,du$. Then

$$\frac{1}{\pi}\int_0^\infty t^{-1/2}(x+t)^{-1}dt = \frac{1}{\pi}\int_0^\infty (xu^2)^{-1/2}(x + xu^2)^{-1}2xu\,du$$

$$= \frac{1}{\pi}\int_0^\infty \frac{2}{x^{1/2}(1+u^2)}du = \frac{2}{\pi x^{1/2}}\cdot\arctan(u)\big|_0^\infty = x^{-1/2}. \tag{B.15}$$

We now extend this identity to matrices. Since $X$ is positive definite, it admits an eigendecomposition $X = UDU^\top$, where $D$ is diagonal and $U$ is orthogonal. Then $X^{-1/2} = UD^{-1/2}U^\top$, and

$$\frac{1}{\pi}\int_0^\infty t^{-1/2}(X+tI)^{-1}dt = \frac{1}{\pi}\int_0^\infty t^{-1/2}(UDU^\top + tUIU^\top)^{-1}dt = \frac{1}{\pi}\int_0^\infty t^{-1/2}(U(D+tI)U^\top)^{-1}dt$$

$$= \frac{1}{\pi}\int_0^\infty t^{-1/2}U(D+tI)^{-1}U^\top dt = U\left(\frac{1}{\pi}\int_0^\infty t^{-1/2}(D+tI)^{-1}dt\right)U^\top. \tag{B.16}$$

Since $(D + tI)^{-1}$ is diagonal, the integral is evaluated entry-wise. By (B.15), the $i$th diagonal entry equals $\lambda_i^{-1/2}$, where $\lambda_i$ is the $i$-th eigenvalue of $X$. Thus, we have

$$\frac{1}{\pi}\int_0^\infty t^{-1/2}(X+tI)^{-1}dt \overset{(B.16)}{=} U\left(\frac{1}{\pi}\int_0^\infty t^{-1/2}(D+tI)^{-1}dt\right)U^\top = UD^{-1/2}U^\top = X^{-1/2}.$$

- **Step 2.** By the representation from Step 1, we can write

$$\|Z_t^{-1/2} - (Z^\star)^{-1/2}\| = \left\| \frac{1}{\pi} \int_0^\infty y^{-1/2} \left[ (Z_t + yI)^{-1} - (Z^\star + yI)^{-1} \right] dy \right\|$$

$$= \left\| \frac{1}{\pi} \int_0^\infty y^{-1/2} (Z_t + yI)^{-1} (Z^\star - Z_t)(Z^\star + yI)^{-1} dy \right\|$$

$$\leq \frac{\|Z^\star - Z_t\|}{\pi} \int_0^\infty y^{-1/2} \|(Z_t + yI)^{-1}\| \|(Z^\star + yI)^{-1}\| dy$$

$$\leq \frac{\|Z^\star - Z_t\|}{\pi} \int_0^\infty y^{-1/2} \frac{1}{(\gamma_S + y)^2} dy,$$

where the last inequality uses $Z_t \succeq \gamma_S I$ and $Z^\star \succeq \gamma_S I$. To evaluate the integral, set $y = \gamma_S u^2$, which yields

$$\int_0^\infty y^{-1/2} \frac{1}{(\gamma_S + y)^2} dy = \int_0^\infty \frac{(\gamma_S u^2)^{-1/2}}{(\gamma_S + \gamma_S u^2)^2} (2\gamma_S u \, du) = \frac{2}{\gamma_S^{3/2}} \int_0^\infty \frac{1}{(1 + u^2)^2} du.$$

Letting $u = \tan\theta$ further gives us

$$\int_0^\infty \frac{1}{(1+u^2)^2} du = \int_0^{\frac{\pi}{2}} \frac{\sec^2\theta}{(1 + \tan^2\theta)^2} d\theta = \int_0^{\frac{\pi}{2}} \frac{1}{\sec^2\theta} d\theta = \int_0^{\frac{\pi}{2}} \cos^2\theta \, d\theta = \int_0^{\frac{\pi}{2}} \frac{1 + \cos 2\theta}{2} d\theta = \frac{\pi}{4}.$$

Combining the above three displays, we conclude that

$$\|Z_t^{-1/2} - (Z^\star)^{-1/2}\| \leq \frac{\|Z^\star - Z_t\|}{\pi} \int_0^\infty y^{-1/2} \frac{1}{(\gamma_S + y)^2} dy = \frac{1}{2\gamma_S^{3/2}} \|Z^\star - Z_t\|.$$

This completes the proof.

### B.3. Proof of Lemma 4.2

To prove this lemma, we have to localize the analysis by introducing the following stopping time: for any given $k \geq 0, r > 0$,

$$\tau_{k,r} := \inf \{ t \geq k : \|\boldsymbol{x}_t - \boldsymbol{x}^\star\| > r \ \ \text{OR} \ \ \|B_t - B^\star\| > r \}. \tag{B.17}$$

It is a stopping time since for any $t \geq k$,

$$\{\tau_{k,r} \leq t\} = \bigcup_{s=k}^t \left( \{\|\boldsymbol{x}_s - \boldsymbol{x}^\star\| > r\} \cup \{\|B_s - B^\star\| > r\} \right) \in \mathcal{F}_{t-1},$$

where the last inclusion uses the fact that $(\boldsymbol{x}_s, B_s)$ is $\mathcal{F}_{s-1}$-measurable for any $s \geq 1$. Furthermore, since $\boldsymbol{x}_t \overset{a.s.}{\to} \boldsymbol{x}^\star$ and $B_t \overset{a.s.}{\to} B^\star$ given in Theorem 3.8 and Lemma 4.1, respectively, we know there exists a random variable $T_r$ with $\mathbb{P}(T_r < \infty) = 1$ such that, for any realization of the sample sequence, $\|\boldsymbol{x}_t - \boldsymbol{x}^\star\| \leq r$ and $\|B_t - B^\star\| \leq r$ for any $t \geq T_r$. For this realization, we know that if $k \geq T_r$, then $\tau_{k,r} = \infty$. Therefore, we conclude that

$$\mathbb{P}(\tau_{k,r} < \infty) \leq \mathbb{P}(T_r > k) \to 0 \quad \text{as} \quad k \to \infty.$$

This suggests that for any given $\epsilon, r > 0$, we can choose $k = k(\epsilon, r)$ (depending on $\epsilon, r$) such that $\mathbb{P}(\tau_{k,r} < \infty) \leq 0.5\epsilon$. Furthermore, for any $t \geq k$, we have

$$\mathbb{P}(\tau_{k,r} \leq t) \leq \mathbb{P}(\tau_{k,r} < \infty) < \frac{\epsilon}{2}. \tag{B.18}$$

Next, we introduce how we choose $r > 0$. To this end, we introduce the following decomposition of the iterate error $\Delta_t := \boldsymbol{x}_t - \boldsymbol{x}^\star$.

**Lemma B.3.** *The iterate error $\Delta_t$ of the algorithm scheme* (2.8) *can be decomposed as*

$$\Delta_{t+1} = (I - \varphi_t \mathcal{R})\Delta_t + \varphi_t \boldsymbol{\theta}_t + \varphi_t \boldsymbol{\delta}_t, \tag{B.19}$$

*where $\mathcal{R} := I - K^\star$, $(\boldsymbol{\theta}_t, \mathcal{F}_t)_{t \geq 0}$ is a martingale difference sequence and $\boldsymbol{\delta}_t$ is the error term, which are defined by*

$$\boldsymbol{\theta}_t := -(I - K_t) B_t^{-1} (g_t - \nabla f_t) + (\boldsymbol{z}_{t,\tau} - (I - K_t)\Delta \boldsymbol{x}_t), \tag{B.20a}$$

$$\boldsymbol{\delta}_t := (K_t - K^\star)\Delta_t - (I - K_t)\left[ (B^\star)^{-1}(\nabla f_t - B^\star \Delta_t) \right] - (I - K_t)\left[ B_t^{-1} - (B^\star)^{-1} \right] \nabla f_t. \tag{B.20b}$$

In addition, by Lemma 3.7 and the condition on $\tau$ that $\tau(1-\sqrt{\mu_t/\nu_t})^{\tau-2} \le \gamma_H/(4\Upsilon_H)$, we know $\|K_t\| \le \gamma_H/(2\Upsilon_H) < 0.5$. By the convergence of $K_t \overset{a.s.}{\to} K^\star$ in Lemma 4.1, $\|K^\star\| \le \gamma_H/(2\Upsilon_H) < 0.5$. By the definition of $\boldsymbol{\delta}_t$ in (B.20b), we have

$$
\begin{aligned}
\|\boldsymbol{\delta}_t\| &\le \frac{3}{2}\left(\|(B^\star)^{-1}\| \cdot \|\nabla f_t - B^\star(\boldsymbol{x}_t - \boldsymbol{x}^\star)\| + \|B_t^{-1} - (B^\star)^{-1}\| \cdot \|\nabla f_t\|\right) + \|K_t - K^\star\| \cdot \|\boldsymbol{x}_t - \boldsymbol{x}^\star\| \\
&\overset{(B.14)}{\lesssim} \frac{3\Upsilon_L}{2\gamma_H}\|\boldsymbol{x}_t - \boldsymbol{x}^\star\|^2 + \frac{3}{2\gamma_H^2}\|B_t - B^\star\| \cdot \|\nabla f_t\| + \|B_t - B^\star\| \cdot \|\boldsymbol{x}_t - \boldsymbol{x}^\star\|,
\end{aligned}
\tag{B.21}
$$

where the first inequality is due to $\|K_t\| < 0.5$ and the second inequality also uses Assumptions 3.1 and 3.2. By the $\Upsilon_H$-Lipschitz continuity of $\nabla f(\boldsymbol{x})$ as implied by Assumption 3.1, (B.21) directly implies that there exists $C_\delta > 0$ such that

$$
\|\boldsymbol{\delta}_t\| \le C_\delta \|\boldsymbol{x}_t - \boldsymbol{x}^\star\|^2 + C_\delta \|B_t - B^\star\| \cdot \|\boldsymbol{x}_t - \boldsymbol{x}^\star\|, \quad \forall t \ge 0.
\tag{B.22}
$$

Let $\lambda_m := \lambda_{\min}(\mathcal{R}) > 0.5$. By the condition on $C_\varphi$ that $2C_\varphi \lambda_m > 1$ when $\varphi = 1$, we choose $\eta > 0$ such that $2C_\varphi \lambda_m(1-\eta) > 1$. Then, for such an $\eta > 0$, we choose $r > 0$ small enough to satisfy

$$
2C_\delta r \le \eta \lambda_m \quad \text{and} \quad 4C_\delta^2 r^2 \le \eta \lambda_m.
\tag{B.23}
$$

With the above chosen $r > 0$ and any given $\epsilon > 0$, by (B.18), we can choose $k = k(\epsilon)$ large enough so that $\mathbb{P}(\tau_{k,r} \le t) < 0.5\epsilon$ for any $t \ge k$. Here, $r$ has been chosen as a constant and hence the dependency of $k$ on $r$ is suppressed.

With this result, we next establish the convergence rates in probability of $B_t$, $\alpha_t$, $\beta_t$, $\gamma_t$, and $K_t$ separately.

• **Convergence rate of $B_t$.** We follow the above discussion. To obtain $O_p(\sqrt{\varphi_t})$ rate for $\|B_t - B^\star\|$, for any given $\epsilon > 0$ and any $M > 0$, we investigate the following quantity

$$
\begin{aligned}
\mathbb{P}(\|B_t - B^\star\| > M\sqrt{\varphi_t}) &= \mathbb{P}\left(\|B_t - B^\star\|^2 > M^2\varphi_t\right) \\
&= \mathbb{P}\left(\{\|B_t - B^\star\|^2 > M^2\varphi_t\} \cap \{\tau_{k,r} > t\}\right) + \mathbb{P}\left(\{\|B_t - B^\star\|^2 > M^2\varphi_t\} \cap \{\tau_{k,r} \le t\}\right) \\
&\le \mathbb{P}\left(\|B_t - B^\star\|^2 \mathbf{1}_{\{\tau_{k,r}>t\}} > M^2\varphi_t\right) + \mathbb{P}(\tau_{k,r} \le t) \\
&\le \frac{\mathbb{E}[\|B_t - B^\star\|^2 \mathbf{1}_{\{\tau_{k,r}>t\}}]}{M^2\varphi_t} + \frac{\epsilon}{2}.
\end{aligned}
\tag{B.24}
$$

By the above derivation, we see that it suffices to show $\mathbb{E}[\|B_t - B^\star\|^2 \mathbf{1}_{\{\tau_{k,r}>t\}}] \le C_B\,\varphi_t$ for $t$ large enough with some constant $C_B > 0$. Then, we can choose $M = M(\epsilon) = \sqrt{2C_B/\epsilon}$ such that $\mathbb{P}\left(\|B_t - B^\star\| > M\sqrt{\varphi_t}\right) \le \epsilon$, which further implies $\|B_t - B^\star\| = O_p(\sqrt{\varphi_t})$.

By (B.1), we obtain

$$
\begin{aligned}
\mathbb{E}[\|B_t - B^\star\|^2 \mathbf{1}_{\{\tau_{k,r}>t\}}] &\le 2\mathbb{E}\left[\left\|\frac{1}{t}\sum_{i=0}^{t-1}(H_i - \nabla^2 f_i)\right\|^2\right] + 2\mathbb{E}\left[\left(\frac{\Upsilon_L}{t}\sum_{i=0}^{t-1}\|\boldsymbol{x}_i - \boldsymbol{x}^\star\|\right)^2 \mathbf{1}_{\{\tau_{k,r}>t\}}\right] \\
&\le 2\mathbb{E}\left[\left\|\frac{1}{t}\sum_{i=0}^{t-1}(H_i - \nabla^2 f_i)\right\|^2\right] + \frac{2\Upsilon_L^2}{t^2}\mathbb{E}\left[\left(\sum_{i=0}^{t-1}\|\boldsymbol{x}_i - \boldsymbol{x}^\star\|\mathbf{1}_{\{\tau_{k,r}>i\}}\right)^2\right] \\
&\le 2\mathbb{E}\left[\left\|\frac{1}{t}\sum_{i=0}^{t-1}(H_i - \nabla^2 f_i)\right\|^2\right] + \frac{2\Upsilon_L^2}{t^2}\left(\sum_{i=0}^{t-1}\left(\mathbb{E}\left[\|\boldsymbol{x}_i - \boldsymbol{x}^\star\|^2\mathbf{1}_{\{\tau_{k,r}>i\}}\right]\right)^{1/2}\right)^2,
\end{aligned}
\tag{B.25}
$$

where the second inequality uses the fact $\{\tau_{k,r} > t\} \subseteq \{\tau_{k,r} > i\}$ for $i \le t$, and the last inequality uses the Cauchy-Schwarz inequality.

•• **Convergence rate of the first term in** (B.25)**.** To control the first term on the right-hand side, we first establish the

bound for $\mathbb{E}[\|\boldsymbol{x}_t - \boldsymbol{x}^\star\|^2]$ as follows. We take full expectation on both sides of (A.30) and obtain for all $t$ large enough,

$$\mathbb{E}[f_{t+1} - f^\star] \le \mathbb{E}[f_t - f^\star] - \frac{\varphi_t}{4\Upsilon_H}\mathbb{E}[\|\nabla f_t\|^2] + \frac{\Upsilon_H(1+C_K)^2 C_{g,2}}{2\gamma_H^2}\varphi_t^2$$

$$\overset{(A.22)}{\le} \mathbb{E}[f_t - f^\star] - \frac{\gamma_H \varphi_t}{2\Upsilon_H}\mathbb{E}[f_t - f^\star] + \frac{\Upsilon_H(1+C_K)^2 C_{g,2}}{2\gamma_H^2}\varphi_t^2$$

$$= \left(1 - \frac{\gamma_H \varphi_t}{2\Upsilon_H}\right)\mathbb{E}[f_t - f^\star] + \frac{\Upsilon_H(1+C_K)^2 C_{g,2}}{2\gamma_H^2}\varphi_t^2.$$

By Leluc & Portier (2023, Lemma 3), we conclude $\limsup_{t\to\infty}\mathbb{E}[f_t - f^\star] = 0$ and hence $\mathbb{E}[f_t - f^\star] \lesssim 1$. Applying (A.22) gives us

$$\mathbb{E}[\|\boldsymbol{x}_t - \boldsymbol{x}^\star\|^2] \le \frac{2}{\gamma_H}\mathbb{E}[f_t - f^\star] \lesssim 1. \tag{B.26}$$

We now return to the first term on the right-hand side of (B.25) and obtain

$$\mathbb{E}\left[\left\|\frac{1}{t}\sum_{i=0}^{t-1}(H_i - \nabla^2 f_i)\right\|^2\right] \le \mathbb{E}\left[\left\|\frac{1}{t}\sum_{i=0}^{t-1}(H_i - \nabla^2 f_i)\right\|_F^2\right]$$

$$= \frac{1}{t^2}\sum_{i=0}^{t-1}\mathbb{E}\left[\|H_i - \nabla^2 f_i\|_F^2\right] \le \frac{d}{t^2}\sum_{i=0}^{t-1}\mathbb{E}\left[\|H_i - \nabla^2 f_i\|^2\right]$$

$$= \frac{d}{t^2}\sum_{i=0}^{t-1}\mathbb{E}[\mathbb{E}[\|H_i - \nabla^2 f_i\|^2 \mid \mathcal{F}_{i-1}]] \le \frac{d}{t^2}\sum_{i=0}^{t-1}\mathbb{E}[C_{H,1}\|\boldsymbol{x}_i - \boldsymbol{x}^\star\|^2 + C_{H,2}] \overset{(B.26)}{\lesssim} \frac{1}{t}. \tag{B.27}$$

Here, the second equality uses the conditional orthogonality of martingale differences (via the tower property) under the Frobenius inner product and the second-to-last inequality is implied by Assumption 3.3 with $q_H = 2$.

•• **Convergence rate of the second term in** (B.25). To control the second term on the right-hand side, we first establish the bound for $u_t := \mathbb{E}\left[\|\Delta_t\|^2 \mathbf{1}_{\{\tau_{k,r} > t\}}\right]$, where we recall that $\Delta_t = \boldsymbol{x}_t - \boldsymbol{x}^\star$. By the one-step recursion of $\Delta_t$ in Lemma B.3, $\forall t \ge k$, we have

$$\mathbb{E}\left[\|\Delta_{t+1}\|^2 \mathbf{1}_{\{\tau_{k,r} > t\}} \mid \mathcal{F}_{t-1}\right] = \mathbb{E}\left[\|(I - \varphi_t \mathcal{R})\Delta_t + \varphi_t \boldsymbol{\theta}_t + \varphi_t \boldsymbol{\delta}_t\|^2 \mathbf{1}_{\{\tau_{k,r} > t\}} \mid \mathcal{F}_{t-1}\right]$$

$$\le (1 - \varphi_t \lambda_m)^2 \|\Delta_t\|^2 \mathbf{1}_{\{\tau_{k,r} > t\}} + \varphi_t^2 \mathbb{E}\left[\|\boldsymbol{\theta}_t + \boldsymbol{\delta}_t\|^2 \mathbf{1}_{\tau_{k,r} > t} \mid \mathcal{F}_{t-1}\right] + 2\varphi_t \left\langle (I - \varphi_t \mathcal{R})\Delta_t, \, \boldsymbol{\delta}_t \mathbf{1}_{\{\tau_{k,r} > t\}}\right\rangle$$

$$\le (1 - \varphi_t \lambda_m)^2 \|\Delta_t\|^2 \mathbf{1}_{\{\tau_{k,r} > t\}} + 2\varphi_t^2 \mathbb{E}\left[\|\boldsymbol{\theta}_t\|^2 \mid \mathcal{F}_{t-1}\right] + 2\varphi_t^2 \|\boldsymbol{\delta}_t\|^2 \mathbf{1}_{\{\tau_{k,r} > t\}}$$

$$+ 2\varphi_t \left\langle (I - \varphi_t \mathcal{R})\Delta_t, \, \boldsymbol{\delta}_t \mathbf{1}_{\{\tau_{k,r} > t\}}\right\rangle, \tag{B.28}$$

Here, the second inequality holds since $(\boldsymbol{\theta}_t, \mathcal{F}_t)$ is a martingale difference sequence and $\mathbf{1}_{\{\tau_{k,r} > t\}}$ is $\mathcal{F}_{t-1}$-measurable; the third inequality follows from the Young's inequality. Below we deal with each term on the right-hand side of (B.28) and then derive the one-step recursion for $u_t = \mathbb{E}\left[\|\Delta_t\|^2 \mathbf{1}_{\{\tau_{k,r} > t\}}\right]$.

For the **second** term in (B.28), we state the following lemma.

**Lemma B.4.** *Under the conditions of Lemma 4.2, for $q = q_g > 2$, there exist $C_{q,1}, C_{q,2} > 0$ such that*

$$\mathbb{E}\left[\|\boldsymbol{\theta}_t\|^q \mid \mathcal{F}_{t-1}\right] \le C_{q,1}\|\boldsymbol{x}_t - \boldsymbol{x}^\star\|^q + C_{q,2}.$$

By Lemma B.4 with $q = q_g > 2$, we have

$$\mathbb{E}\left[\|\boldsymbol{\theta}_t\|^2 \mid \mathcal{F}_{t-1}\right] \le \left(\mathbb{E}\left[\|\boldsymbol{\theta}_t\|^{q_g} \mid \mathcal{F}_{t-1}\right]\right)^{2/q_g} \le \left(C_{q_g,1}\|\Delta_t\|^{q_g} + C_{q_g,2}\right)^{2/q_g} \le C_{q_g,1}^{2/q_g}\|\Delta_t\|^2 + C_{q_g,2}^{2/q_g}, \tag{B.29}$$

Here, the first step follows from the Jensen's inequality; and the last inequality uses that $2/q_g \in (0,1)$ and hence $(a+b)^{2/q_g} \le a^{2/q_g} + b^{2/q_g}$ for $a, b \ge 0$.

For the **third** term in (B.28), we first obtain from (B.22) that

$$\|\boldsymbol{\delta}_t\| \mathbf{1}_{\{\tau_{k,r} > t\}} \le C_\delta r \|\Delta_t\| \mathbf{1}_{\{\tau_{k,r} > t\}} + C_\delta r \|\Delta_t\| \mathbf{1}_{\{\tau_{k,r} > t\}} = 2C_\delta r \|\Delta_t\| \mathbf{1}_{\{\tau_{k,r} > t\}}, \tag{B.30}$$

where the first inequality applies the definition of the stopping time $\tau_{k,r}$ in (B.17). Then, by the choice of $r$ mentioned in (B.23), we have

$$\|\boldsymbol{\delta}_t\|^2 \mathbf{1}_{\{\tau_{k,r}>t\}} \le 4C_\delta^2 r^2 \|\Delta_t\|^2 \mathbf{1}_{\{\tau_{k,r}>t\}} \le \eta\lambda_m \|\Delta_t\|^2 \mathbf{1}_{\{\tau_{k,r}>t\}}, \tag{B.31}$$

where we recall that $\eta$ is chosen so that $2C_\varphi \lambda_m (1-\eta) > 1$.

For the **last** term in (B.28), we have

$$\langle (I - \varphi_t \mathcal{R})\Delta_t, \ \boldsymbol{\delta}_t \mathbf{1}_{\{\tau_{k,r}>t\}} \rangle \overset{(B.30)}{\le} \|I - \varphi_t \mathcal{R}\| \|\Delta_t\| \cdot 2C_\delta r \|\Delta_t\| \mathbf{1}_{\{\tau_{k,r}>t\}} \overset{(B.23)}{\le} \eta\lambda_m \|\Delta_t\|^2 \mathbf{1}_{\{\tau_{k,r}>t\}}, \tag{B.32}$$

where the second inequality also uses the fact that $0 \preceq I - \varphi_t \mathcal{R} = (1-\varphi_t)I + \varphi_t K^\star \preceq I$ for $t$ large enough.

Combining (B.29), (B.31) and (B.32) with (B.28), we have

$$\mathbb{E}\big[\|\Delta_{t+1}\|^2 \mathbf{1}_{\{\tau_{k,r}>t\}} \big| \mathcal{F}_{t-1}\big] \le (1 - \varphi_t \lambda_m)^2 \|\Delta_t\|^2 \mathbf{1}_{\{\tau_{k,r}>t\}} + 2\varphi_t^2 (C_{q_g,1}^{2/q_g} \|\Delta_t\|^2 + C_{q_g,2}^{2/q_g})$$
$$+ \varphi_t^2 \cdot 2\eta\lambda_m \|\Delta_t\|^2 \mathbf{1}_{\{\tau_{k,r}>t\}} + \varphi_t \cdot 2\eta\lambda_m \|\Delta_t\|^2 \mathbf{1}_{\{\tau_{k,r}>t\}},$$
$$\implies u_{t+1} = \mathbb{E}\big[\|\Delta_{t+1}\|^2 \mathbf{1}_{\{\tau_{k,r}>t+1\}}\big] \le \mathbb{E}\big[\|\Delta_{t+1}\|^2 \mathbf{1}_{\{\tau_{k,r}>t\}}\big] \le (1 - \varphi_t \lambda_m)^2 u_t + 2\varphi_t^2 (C_{q_g,1}^{2/q_g} \mathbb{E}\big[\|\Delta_t\|^2\big] + C_{q_g,2}^{2/q_g})$$
$$+ \varphi_t^2 \cdot 2\eta\lambda_m \mathbb{E}\big[\|\Delta_t\|^2\big] + \varphi_t \cdot 2\eta\lambda_m u_t$$
$$= \big[1 - \varphi_t \cdot 2(1-\eta)\lambda_m\big] u_t + C_u \cdot \varphi_t^2.$$

Here, such a constant $C_u > 0$ exists since $u_t \le \mathbb{E}[\|\Delta_t\|^2] \lesssim 1$ as established in (B.26).

By Leluc & Portier (2023, Lemma 3), we obtain for $\varphi \in (0.5, 1]$

$$\limsup_{t\to\infty} \frac{u_t}{\varphi_t} \lesssim \begin{cases} \frac{1}{2C_\varphi (1-\eta)\lambda_m} \cdot C_u, & \varphi \in (0.5, 1), \\ \frac{1}{2C_\varphi (1-\eta)\lambda_m - 1} \cdot C_u, & \varphi = 1, \end{cases} \implies u_t = \mathbb{E}\big[\|\Delta_t\|^2 \mathbf{1}_{\{\tau_{k,r}>t\}}\big] \lesssim \varphi_t. \tag{B.33}$$

Combining (B.33) with the second term on the right-hand side of (B.25), we obtain

$$\frac{2\Upsilon_L^2}{t^2} \left(\sum_{i=0}^{t-1} \big(\mathbb{E}\big[\|\boldsymbol{x}_i - \boldsymbol{x}^\star\|^2 \mathbf{1}_{\{\tau_{k,r}>i\}}\big]\big)^{1/2}\right)^2 \overset{(B.33)}{\lesssim} \frac{1}{t^2}\left(\sum_{i=0}^{t-1} \sqrt{\varphi_i}\right)^2 = \frac{C_\varphi}{t^2}\left(\sum_{i=0}^{t-1} \frac{1}{(i+1)^{\varphi/2}}\right)^2$$
$$\lesssim \frac{C_\varphi}{t^2}\left(\int_0^t \frac{1}{(x+1)^{\varphi/2}}\,dx\right)^2 \le \frac{C_\varphi}{(t+1)^\varphi} = \varphi_t. \tag{B.34}$$

Plugging (B.27) and (B.34) into (B.25) yields

$$\mathbb{E}[\|B_t - B^\star\|^2 \mathbf{1}_{\{\tau_{k,r}>t\}}] \lesssim 1/t + \varphi_t \lesssim \varphi_t, \qquad \varphi \in (0.5, 1].$$

As discussed around (B.24), the above display implies

$$\|B_t - B^\star\| = O_p(\sqrt{\varphi_t}). \tag{B.35}$$

- **Convergence rates of** $(\alpha_t, \beta_t, \gamma_t)$**.** The result immediately follows from (B.11) and (B.35).

- **Convergence rate of** $K_t$**.** The result immediately follows from (B.14) and (B.35).

This completes the proof of Lemma 4.2.

## B.4. Proof of Lemma B.3

By Lemma 3.5, $z_{t,\tau} - (I - K_t)\Delta x_t$ forms a martingale difference sequence with respect to filtration $\{\mathcal{F}_t\}_{t \geq 0}$. Recalling that $B^\star = \nabla^2 f(x^\star)$, we decompose the error $\Delta_{t+1} = x_{t+1} - x^\star$ as follows

$$
\begin{aligned}
\Delta_{t+1} = x_{t+1} - x^\star &= x_t - x^\star + \varphi_t z_{t,\tau} \\
&= x_t - x^\star + \varphi_t(I - K_t)\Delta x_t + \varphi_t(z_{t,\tau} - (I - K_t)\Delta x_t) \\
&= x_t - x^\star - \varphi_t(I - K_t)B_t^{-1}g_t + \varphi_t(z_{t,\tau} - (I - K_t)\Delta x_t) \\
&= x_t - x^\star - \varphi_t(I - K_t)B_t^{-1}\nabla f_t - \varphi_t(I - K_t)B_t^{-1}(g_t - \nabla f_t) + \varphi_t(z_{t,\tau} - (I - K_t)\Delta x_t) \\
&= x_t - x^\star - \varphi_t(I - K_t)(B^\star)^{-1}\nabla f_t - \varphi_t(I - K_t)[B_t^{-1} - (B^\star)^{-1}]\nabla f_t + \varphi_t\theta_t \\
&= \{I - \varphi_t(I - K_t)\}(x_t - x^\star) - \varphi_t(I - K_t)[(B^\star)^{-1}(\nabla f_t - B^\star(x_t - x^\star))] - \varphi_t(I - K_t)[B_t^{-1} - (B^\star)^{-1}]\nabla f_t + \varphi_t\theta_t \\
&= \{I - \varphi_t(I - K^\star)\}\Delta_t + \varphi_t(K_t - K^\star)\Delta_t - \varphi_t(I - K_t)[(B^\star)^{-1}(\nabla f_t - B^\star\Delta_t)] - \varphi_t(I - K_t)[B_t^{-1} - (B^\star)^{-1}]\nabla f_t + \varphi_t\theta_t \\
&= \{I - \varphi_t(I - K^\star)\}\Delta_t + \varphi_t\delta_t + \varphi_t\theta_t,
\end{aligned}
$$

where

$$
\theta_t = -(I - K_t)B_t^{-1}(g_t - \nabla f_t) + (z_{t,\tau} - (I - K_t)\Delta x_t)
$$

forms a martingale difference with respect to filtration $\{\mathcal{F}_t\}_{t \geq 0}$, and

$$
\delta_t = (K_t - K^\star)\Delta_t - (I - K_t)[(B^\star)^{-1}(\nabla f_t - B^\star\Delta_t)] - (I - K_t)[B_t^{-1} - (B^\star)^{-1}]\nabla f_t
$$

collects the higher-order approximation errors. This completes the proof of Lemma B.3.

## B.5. Proof of Lemma B.4

For $q_g > 2$ in Assumption 3.2, we let $q = q_g$ and have

$$
\begin{aligned}
\mathbb{E}[\|\theta_t\|^q \mid \mathcal{F}_{t-1}] &\overset{\text{(B.20a)}}{\leq} 2^{q-1}\left\{\mathbb{E}[\|(I - K_t)B_t^{-1}(g_t - \nabla f_t)\|^q \mid \mathcal{F}_{t-1}] + \mathbb{E}[\|z_{t,\tau} - (I - K_t)\Delta x_t\|^q \mid \mathcal{F}_{t-1}]\right\} \\
&\leq 2^{q-1}\left(\mathbb{E}[\|(I - K_t)B_t^{-1}\|^q\|g_t - \nabla f_t\|^q \mid \mathcal{F}_{t-1}] + \mathbb{E}[\|(\widetilde{K}_t - K_t)\Delta x_t\|^q \mid \mathcal{F}_{t-1}]\right) \quad \text{(by Lemma 3.5)} \\
&\overset{\text{(A.28)}}{\leq} 2^{q-1}\left\{\left(\frac{3}{2\gamma_H}\right)^q \mathbb{E}[\|g_t - \nabla f_t\|^q \mid \mathcal{F}_{t-1}] + \left(C_K + \frac{1}{2}\right)^q \mathbb{E}[\|\Delta x_t\|^q \mid \mathcal{F}_{t-1}]\right\} \quad \text{(also } \|K_t\| \leq 0.5) \\
&\leq 2^{q-1}\left\{\left(\frac{3}{2\gamma_H}\right)^q (C_{g,1}\|x_t - x^\star\|^q + C_{g,2}) + \left(C_H + \frac{1}{2}\right)^q \mathbb{E}[\|B_t^{-1}g_t\|^q \mid \mathcal{F}_{t-1}]\right\} \\
&\overset{\text{(A.26)}}{\leq} \frac{1}{2\gamma_H^q}\left\{3^q(C_{g,1}\|x_t - x^\star\|^q + C_{g,2}) + (2C_H + 1)^q \mathbb{E}[\|g_t\|^q \mid \mathcal{F}_{t-1}]\right\} \\
&\leq \frac{1}{2\gamma_H^q}\left\{3^q(C_{g,1}\|x_t - x^\star\|^q + C_{g,2}) + (2C_H + 1)^q \cdot 2^{q-1}(\|\nabla f_t\|^q + \mathbb{E}[\|g_t - \nabla f_t\|^q \mid \mathcal{F}_{t-1}])\right\} \\
&\leq \frac{1}{2\gamma_H^q}\left\{3^q(C_{g,1}\|x_t - x^\star\|^q + C_{g,2}) + \frac{1}{2}(4C_H + 2)^q(\Upsilon_H^q\|x_t - x^\star\|^q + C_{g,1}\|x_t - x^\star\|^q + C_{g,2})\right\} \\
&=: C_{q,1}\|x_t - x^\star\|^q + C_{q,2}.
\end{aligned}
$$

Here, the second inequality holds due to Lemma 3.5; and the fourth and the second last inequalities are both direct applications of Assumption 3.2. This completes the proof.

## B.6. Proof of Theorem 4.3

Let $\Phi_t := \sum_{i=0}^{t-1}\varphi_i$ and $\lambda_M := \lambda_{\max}(I - K^\star)$. By Lemma 4.1, we have $K_t \overset{a.s.}{\to} K^\star$. Moreover, since $\|K^\star\| \leq \gamma_H/(2\Upsilon_H) < 0.5$, we know the matrix $\mathcal{R} = I - K^\star$ is positive definite with $\lambda_m = \lambda_{\min}(\mathcal{R}) > 0.5$. Define

$$
\Gamma_t := \mathbb{E}[\theta_{t+1}\theta_{t+1}^\top \mid \mathcal{F}_t], \qquad \mathcal{A}_t := \{\|\Gamma_t\| \leq 2\|\Gamma^\star\| \vee 1\}. \tag{B.36}
$$

With these definitions, we can rewrite (B.19) as

$$
\Delta_{t+1} = (I - \varphi_t\mathcal{R})\Delta_t + \varphi_t\theta_t\mathbf{1}_{\mathcal{A}_{t-1}} + \varphi_t\delta_t + \varphi_t\theta_t\mathbf{1}_{\mathcal{A}_{t-1}^c}. \tag{B.37}
$$

Applying (B.37) recursively gives us

$$\Delta_t = \prod_{i=0}^{t-1}(I - \varphi_i \mathcal{R})\Delta_0 + \sum_{i=0}^{t-1}\prod_{j=i+1}^{t-1}(I - \varphi_j \mathcal{R})\varphi_i[\boldsymbol{\theta}_i \mathbf{1}_{\mathcal{A}_{i-1}} + \boldsymbol{\delta}_i + \boldsymbol{\theta}_i \mathbf{1}_{\mathcal{A}_{i-1}^c}],$$

which implies

$$\frac{1}{\sqrt{\varphi_{t-1}}}\Delta_t = \frac{1}{\sqrt{\varphi_{t-1}}}\Pi_{t,0}\Delta_0 + \frac{1}{\sqrt{\varphi_{t-1}}}\sum_{i=0}^{t-1}\varphi_i\Pi_{t,i+1}\boldsymbol{\theta}_i\mathbf{1}_{\mathcal{A}_{i-1}} + \frac{1}{\sqrt{\varphi_{t-1}}}\sum_{i=0}^{t-1}\varphi_i\Pi_{t,i+1}\boldsymbol{\delta}_i + \frac{1}{\sqrt{\varphi_{t-1}}}\sum_{i=0}^{t-1}\varphi_i\Pi_{t,i+1}\boldsymbol{\theta}_i\mathbf{1}_{\mathcal{A}_{i-1}^c}$$

$$=: \mathcal{I}_{1,t} + \mathcal{I}_{2,t} + \mathcal{I}_{3,t} + \mathcal{I}_{4,t}. \tag{B.38}$$

Here, we define $\Pi_{n,m} := \prod_{i=m}^{n-1}(I - \varphi_i \mathcal{R})$ for $m < n$ and $I$ otherwise. In what follows, we analyze each term on the right-hand side of (B.38) and establish the following claims:

(a) $\mathcal{I}_{1,t} \xrightarrow{a.s.} \mathbf{0}$, as $t \to \infty$;

(b) $\mathcal{I}_{2,t} \xrightarrow{d} \mathcal{N}(\mathbf{0}, \Sigma^\star)$, as $t \to \infty$, where $\Sigma^\star$ satisfies $(\mathcal{R} - \zeta I)\Sigma^\star + \Sigma^\star(\mathcal{R} - \zeta I) = \Gamma^\star$ and $\zeta := \mathbf{1}_{\{\varphi=1\}}/(2C_\varphi)$;

(c) $\mathcal{I}_{3,t} \xrightarrow{p} \mathbf{0}$, as $t \to \infty$;

(d) $\mathcal{I}_{4,t} \xrightarrow{a.s.} \mathbf{0}$, as $t \to \infty$.

By the four claims above, Slutsky's theorem, and the fact that $\varphi_t/\varphi_{t-1} \to 1$, we immediately finish the proof of Theorem 4.3.

• **Proof of (a).** Since $\varphi_t \to 0$ as $t \to \infty$, there exists $j \in \mathbb{N}$ such that $1 - \varphi_i\lambda_M > 0$, $\forall i > j$. Therefore, we have

$$\|\mathcal{I}_{1,t}\| \leq \frac{1}{\sqrt{\varphi_{t-1}}}\left\|\prod_{i=0}^{t-1}(I - \varphi_i\mathcal{R})\right\|\|\Delta_0\|$$

$$\leq \frac{1}{\sqrt{\varphi_{t-1}}}\left\|\prod_{i=0}^{j}(I - \varphi_i\mathcal{R})\right\|\|\Delta_0\|\prod_{i=j+1}^{t-1}(1 - \varphi_i\lambda_m)$$

$$\leq \frac{1}{\sqrt{\varphi_{t-1}}}\left\|\prod_{i=0}^{j}(I - \varphi_i\mathcal{R})\right\|\|\Delta_0\|\prod_{i=j+1}^{t-1}\exp(-\varphi_i\lambda_m)$$

$$= \left\|\prod_{i=0}^{j}(I - \varphi_i\mathcal{R})\right\|\|\Delta_0\|\exp(\lambda_m\Phi_{j+1})\cdot\exp\left(-\lambda_m\Phi_t - 0.5\log\varphi_{t-1}\right).$$

Note that $\left\|\prod_{i=0}^{j}(I - \varphi_i K)\right\|\|\Delta_0\|\exp(\lambda_m\Phi_{j+1})$ is independent of $t$ and let

$$d_t := -\lambda_m\Phi_t - 0.5\log\varphi_{t-1}, \tag{B.39}$$

for which we have the following:

$$d_t \lesssim \begin{cases} -t^{1-\varphi} + \log t, & \varphi \in \left(\frac{1}{2}, 1\right), \\ -(\lambda_m C_\varphi - 0.5)\log t, & \varphi = 1. \end{cases}$$

Since $C_\varphi > 1/(2\lambda_m)$, we obtain $d_t \to -\infty$ and $\mathcal{I}_{1,t} \xrightarrow{a.s.} \mathbf{0}$ as $t \to \infty$.

• **Proof of (b).** Recall that $\mathcal{I}_{2,t} = 1/\sqrt{\varphi_{t-1}}\sum_{i=0}^{t-1}\varphi_i\Pi_{t,i+1}\boldsymbol{\theta}_i\mathbf{1}_{\mathcal{A}_{i-1}}$. Since $\mathcal{A}_{i-1}$ is $\mathcal{F}_{i-1}$-measurable, $\mathcal{I}_{2,t}$ is a sum of martingale increments and we can show its limiting distribution via the following central limit theorem for martingale arrays. For simplicity, we let

$$W_{t,i} = \frac{\varphi_i}{\sqrt{\varphi_{t-1}}}\Pi_{t,i+1}\boldsymbol{\theta}_i\mathbf{1}_{\mathcal{A}_{i-1}}.$$

**Lemma B.5.** *(Hall & Heyde, 2014, Corollary 3.1) Let $(W_{t,i})$ be a triangular array of random vectors such that*

$$\mathbb{E}\left[W_{t,i}|\mathcal{F}_{i-1}\right] = \mathbf{0}, \tag{B.40}$$

$$\sum_{i=0}^{t-1} \mathbb{E}\left[W_{t,i}W_{t,i}^\top|\mathcal{F}_{i-1}\right] \xrightarrow{p} V^* \succeq \mathbf{0}, \tag{B.41}$$

$$\sum_{i=0}^{t-1} \mathbb{E}\left[\|W_{t,i}\|^2\mathbf{1}_{\{\|W_{t,i}\|>\epsilon\}}\big|\mathcal{F}_{i-1}\right] \xrightarrow{p} 0, \quad \forall\,\epsilon > 0, \tag{B.42}$$

*then* $\sum_{i=0}^{t-1} W_{t,i} \xrightarrow{d} \mathcal{N}(\mathbf{0}, V^*)$ *as* $t \to \infty$.

To proceed, we also need the following lemma.

**Lemma B.6.** *Under the conditions of Theorem 4.3, we have the following convergence as* $t \to \infty$

$$\Gamma_t = \mathbb{E}[\boldsymbol{\theta}_{t+1}\boldsymbol{\theta}_{t+1}^\top \mid \mathcal{F}_t] \xrightarrow{a.s.} \mathbb{E}[(I - \widetilde{K}^\star)\Omega^\star(I - \widetilde{K}^\star)^\top] = \Gamma^\star.$$

•• **Verifying** (B.40). By Lemma 3.5, we obtain

$$\mathbb{E}[W_{t,i}|\mathcal{F}_{i-1}] = \frac{\varphi_i}{\sqrt{\varphi_{t-1}}}\Pi_{t,i+1}\mathbb{E}[\boldsymbol{\theta}_i|\mathcal{F}_{i-1}]\mathbf{1}_{\mathcal{A}_{i-1}} = \mathbf{0}.$$

•• **Verifying** (B.41). Let $\Sigma_t := \sum_{i=0}^{t-1}\mathbb{E}\left[W_{t,i}W_{t,i}^\top|\mathcal{F}_{i-1}\right]$ be the quadratic variation of $\mathcal{I}_{2,t}$. First, we show that $(\Sigma_t)$ is bounded as follows:

$$\|\Sigma_t\| = \left\|\sum_{i=0}^{t-1}\frac{\varphi_i^2}{\varphi_{t-1}}\Pi_{t,i+1}\mathbb{E}\left[\boldsymbol{\theta}_i\boldsymbol{\theta}_i^T|\mathcal{F}_{i-1}\right]\Pi_{t,i+1}\mathbf{1}_{\mathcal{A}_{i-1}}\right\|$$

$$\leq \frac{1}{\varphi_{t-1}}\sum_{i=0}^{t-1}\varphi_i^2\|\Pi_{t,i+1}\|^2\|\Gamma_{i-1}\|\mathbf{1}_{\mathcal{A}_{i-1}} \overset{\text{(B.36)}}{\leq} \frac{1}{\varphi_{t-1}}\sum_{i=0}^{t-1}\varphi_i^2\|\Pi_{t,i+1}\|^2\cdot(2\|\Gamma^\star\|\vee 1).$$

Similar to the argument in the proof of **(a)**, there exists $j \in \mathbb{N}$ such that $1 - \varphi_i\lambda_M > 0$, $\forall\,i > j$. Therefore, for $t$ large enough, we have

$$\|\Sigma_t\| \lesssim \underbrace{\frac{1}{\varphi_{t-1}}\sum_{i=0}^{j}\varphi_i^2\|\Pi_{j+1,i+1}\|^2\cdot\prod_{k=j+1}^{t-1}(1-\varphi_k\lambda_m)^2}_{a_t} + \underbrace{\frac{1}{\varphi_{t-1}}\sum_{i=j+1}^{t-1}\varphi_i^2\prod_{k=i+1}^{t-1}(1-\varphi_k\lambda_m)^2}_{b_t}.$$

Then we analyze the two sequences $(a_t)$ and $(b_t)$ separately. For $(a_t)$, we have

$$a_t = \frac{1}{\varphi_{t-1}}\sum_{i=0}^{j}\varphi_i^2\|\Pi_{j+1,i+1}\|^2\cdot\prod_{k=j+1}^{t-1}(1-\varphi_k\lambda_m)^2$$

$$\leq \frac{1}{\varphi_{t-1}}\left(\sum_{i=0}^{j}\varphi_i^2\right)\left(\max_{0\leq i\leq j}\|\Pi_{j+1,i+1}\|^2\right)\cdot\prod_{k=j+1}^{t-1}(1-\varphi_k\lambda_m)^2 \lesssim \frac{1}{\varphi_{t-1}}\prod_{k=j+1}^{t-1}\exp(-2\varphi_k\lambda_m) \overset{\text{(B.39)}}{\lesssim} \exp(2d_t).$$

As shown in the proof of **(a)**, $d_t \to -\infty$ as $t \to \infty$. Thus, $a_t \to 0$ as $t \to \infty$. For $(b_t)$, we have the following recursive relation:

$$b_t = \frac{1}{\varphi_{t-1}}\sum_{i=j+1}^{t-1}\varphi_i^2\prod_{k=i+1}^{t-1}(1-\varphi_k\lambda_m)^2 = \frac{1}{\varphi_{t-1}}\left[\varphi_{t-1}^2 + \sum_{i=j+1}^{t-2}\varphi_i^2\prod_{k=i+1}^{t-2}(1-\varphi_k\lambda_m)^2\cdot(1-\varphi_{t-1}\lambda_m)^2\right]$$

$$\implies \varphi_{t-1}b_t = (1-\varphi_{t-1}\lambda_m)^2\varphi_{t-2}b_{t-1} + \varphi_{t-1}^2.$$

By Leluc & Portier (2023, Lemma 3), we directly obtain

$$\limsup_{t \to \infty} \frac{\varphi_{t-1} b_t}{1/t^\varphi} \leq \begin{cases} \frac{1}{2C_\varphi \lambda_m} \cdot C_\varphi^2, & \varphi \in (0.5, 1), \\ \frac{1}{2C_\varphi \lambda_m - 1} \cdot C_\varphi^2, & \varphi = 1, \end{cases} \implies \limsup_{t \to \infty} b_t \leq \begin{cases} \frac{1}{2\lambda_m}, & \varphi \in (0.5, 1), \\ \frac{C_\varphi}{2C_\varphi \lambda_m - 1}, & \varphi = 1. \end{cases}$$

Thus, $(b_t)$ is bounded for $\varphi \in (0.5, 1]$. Combining the above four displays, we obtain that $\|\Sigma_t\|$ is bounded by a deterministic upper bound almost surely. Furthermore, by the definition of $\Sigma_t$, we have the following recursive relation:

$$\begin{aligned} \varphi_{t-1} \Sigma_t &= \sum_{i=0}^{t-1} \varphi_i^2 \Pi_{t,i+1} \Gamma_{i-1} \Pi_{t,i+1} \mathbf{1}_{\mathcal{A}_{i-1}} \\ &= \varphi_{t-1}^2 \Gamma_{t-2} \mathbf{1}_{\mathcal{A}_{t-2}} + (I - \varphi_{t-1}\mathcal{R}) \left( \sum_{i=0}^{t-2} \varphi_i^2 \Pi_{t,i+1} \Gamma_{i-1} \Pi_{t,i+1} \mathbf{1}_{\mathcal{A}_{i-1}} \right) (I - \varphi_{t-1}\mathcal{R}) \\ &= \varphi_{t-1}^2 \Gamma_{t-2} \mathbf{1}_{\mathcal{A}_{t-2}} + \varphi_{t-2} (I - \varphi_{t-1}\mathcal{R}) \Sigma_{t-1} (I - \varphi_{t-1}\mathcal{R}) \\ &= \varphi_{t-1}^2 \Gamma_{t-2} \mathbf{1}_{\mathcal{A}_{t-2}} + \varphi_{t-2} \left[ \Sigma_{t-1} - \varphi_{t-1}\mathcal{R}\Sigma_{t-1} - \varphi_{t-1}\Sigma_{t-1}\mathcal{R} + O\left(\varphi_{t-1}^2\right) \right], \end{aligned}$$

where the last equality uses the result that $(\Sigma_t)$ is deterministically bounded as we have shown earlier. Dividing by $\varphi_{t-1}$ on both sides, we obtain

$$\Sigma_t = \Sigma_{t-1} - \varphi_{t-1} \left( \mathcal{R}\Sigma_{t-1} + \Sigma_{t-1}\mathcal{R} - \Gamma_{t-2}\mathbf{1}_{\mathcal{A}_{t-2}} \right) + \frac{\varphi_{t-2} - \varphi_{t-1}}{\varphi_{t-1}} \Sigma_{t-1} + O\left( \varphi_{t-2}\varphi_{t-1} + |\varphi_{t-2} - \varphi_{t-1}| \right),$$

which gives us the following:

$$\Sigma_t = \begin{cases} \Sigma_{t-1} - \varphi_{t-1} \left( \mathcal{R}\Sigma_{t-1} + \Sigma_{t-1}\mathcal{R} - \Gamma_{t-2}\mathbf{1}_{\mathcal{A}_{t-2}} \right) + o(\varphi_{t-1}), & \varphi \in (0.5, 1), \\ \Sigma_{t-1} - \varphi_{t-1} \left[ \left( \mathcal{R} - \frac{I}{2C_\varphi} \right) \Sigma_{t-1} + \Sigma_{t-1} \left( \mathcal{R} - \frac{I}{2C_\varphi} \right) - \Gamma_{t-2}\mathbf{1}_{\mathcal{A}_{t-2}} \right] + o(\varphi_{t-1}), & \varphi = 1. \end{cases}$$

Recall that $\zeta = \mathbf{1}_{\{\varphi=1\}}/(2C_\varphi)$ and define $\mathcal{R}_\zeta := \mathcal{R} - \zeta I$. We can combine the above two cases for $\varphi \in (0.5, 1]$ as

$$\Sigma_t = \Sigma_{t-1} - \varphi_{t-1} \left( \mathcal{R}_\zeta \Sigma_{t-1} + \Sigma_{t-1}\mathcal{R}_\zeta - \Gamma_{t-2}\mathbf{1}_{\mathcal{A}_{t-2}} \right) + o(\varphi_{t-1}).$$

Then, we vectorize this equation by letting $s_t = \text{vec}(\Sigma_t)$, and obtain

$$\begin{aligned} s_t &= s_{t-1} - \varphi_{t-1} \left[ \text{vec}(\mathcal{R}_\zeta \Sigma_{t-1} + \Sigma_{t-1}\mathcal{R}_\zeta) - \text{vec}(\Gamma_{t-2}\mathbf{1}_{\mathcal{A}_{t-2}}) \right] + o(\varphi_{t-1}) \\ &= s_{t-1} - \varphi_{t-1} \left[ (I \otimes \mathcal{R}_\zeta + \mathcal{R}_\zeta \otimes I) s_{t-1} - \text{vec}(\Gamma^\star) \right] + \varphi_{t-1} \text{vec}(\Gamma_{t-2}\mathbf{1}_{\mathcal{A}_{t-2}} - \Gamma^\star) + o(\varphi_{t-1}) \\ &= s_{t-1} - \varphi_{t-1}(Q \, s_{t-1} - \text{vec}(\Gamma^\star)) + \varepsilon_{t-1}\varphi_{t-1}, \end{aligned}$$

where $Q = I \otimes \mathcal{R}_\zeta + \mathcal{R}_\zeta \otimes I$ and $\varepsilon_t \overset{a.s.}{\to} \mathbf{0}$, which is due to $\Gamma_t \overset{a.s.}{\to} \Gamma^\star$ given in Lemma B.6. Let $s^\star$ be the solution of equation $Qs - \text{vec}(\Gamma^\star) = \mathbf{0}$. The existence and uniqueness of such a solution are guaranteed since $\mathcal{R}_\zeta = \mathcal{R} - \zeta I$ is positive definite. Since all eigenvalues of $\mathcal{R}_\zeta$ are strictly positive, the sum of any pair of eigenvalues is non-zero, implying that $Q$ is positive definite. For $t$ large enough, we have $1 - 2\varphi_{t-1}\lambda_M > 0$ and hence $\|I - \varphi_{t-1}Q\| = 1 - 2\varphi_{t-1}\lambda_m$. Then, we can rewrite the recursive relation as

$$s_t - s^\star = (s_{t-1} - s^\star) - \varphi_{t-1}Q(s_{t-1} - s^\star) + \varepsilon_{t-1}\varphi_{t-1} = (I - \varphi_{t-1}Q)(s_{t-1} - s^\star) + \varepsilon_{t-1}\varphi_{t-1},$$

which implies

$$\|s_t - s^\star\| \leq \|I - \varphi_{t-1}Q\| \|s_{t-1} - s^\star\| + \|\varepsilon_{t-1}\|\varphi_{t-1} = (1 - 2\varphi_{t-1}\lambda_m)\|s_{t-1} - s^\star\| + \|\varepsilon_{t-1}\|\varphi_{t-1}.$$

By Leluc & Portier (2023, Lemma 3), for $\varphi \in (0.5, 1]$, almost surely we have

$$\limsup_{t \to \infty} \|s_t - s^\star\| \leq \frac{1}{2C_\varphi \lambda_m} \cdot 0, \implies \lim_{t \to \infty} \|s_t - s^\star\| = 0,$$

which is equivalent to $\Sigma_t \overset{a.s.}{\to} \Sigma^*$, where $\Sigma^*$ is the unique solution of the Lyapunov equation

$$(\mathcal{R} - \zeta I)\Sigma^\star + \Sigma^\star(\mathcal{R} - \zeta I) = \Gamma^\star.$$

•• **Verifying** (B.42). For simplicity, let

$$\sigma_t^2 := \sum_{i=0}^{t-1} \mathbb{E}\left[\|W_{t,i}\|^2 \mathbf{1}_{\{\|W_{t,i}\|>\epsilon\}}\big|\mathcal{F}_{i-1}\right]$$

for any given $\epsilon > 0$. Thus, it suffices to show that $\sigma_t^2 \overset{a.s.}{\to} 0$ as $t \to \infty$. Under Assumption 3.2 with $q_g > 2$, we write $q_g = 2+\delta$ for some $\delta > 0$. Then, we have

$$\sigma_t^2 \le \sum_{i=0}^{t-1} \frac{\varphi_i^2}{\varphi_{t-1}} \mathbb{E}\left[\|\Pi_{t,i+1}\|^2 \mathbf{1}_{\{\varphi_i\|\Pi_{t,i+1}\boldsymbol{\theta}_i\|>\epsilon\sqrt{\varphi_{t-1}}\}}\Big|\mathcal{F}_{i-1}\right] \le \frac{1}{\epsilon^\delta} \sum_{i=0}^{t-1} \frac{\varphi_i^{2+\delta}}{\varphi_{t-1}^{1+\delta/2}} \mathbb{E}\left[\|\Pi_{t,i+1}\boldsymbol{\theta}_i\|^{2+\delta}\big|\mathcal{F}_{i-1}\right]$$

$$\le \frac{1}{\epsilon^\delta} \sum_{i=0}^{t-1} \left(\frac{\varphi_i}{\sqrt{\varphi_{t-1}}}\|\Pi_{t,i+1}\|\right)^{2+\delta} \mathbb{E}\left[\|\boldsymbol{\theta}_i\|^{2+\delta}\big|\mathcal{F}_{i-1}\right]. \tag{B.43}$$

By Lemma B.4, there exist $C_{2+\delta,1}, C_{2+\delta,2} > 0$ such that

$$\mathbb{E}\left[\|\boldsymbol{\theta}_t\|^{2+\delta}\big|\mathcal{F}_{t-1}\right] \le C_{2+\delta,1}\|\boldsymbol{x}_t - \boldsymbol{x}^\star\|^{2+\delta} + C_{2+\delta,2}.$$

Since $\boldsymbol{x}_t \overset{a.s.}{\to} \boldsymbol{x}^*$ as $t \to \infty$, we can conclude that the sequence $\left(\mathbb{E}\left[\|\boldsymbol{\theta}_t\|^{2+\delta}\big|\mathcal{F}_{t-1}\right]\right)$ is bounded almost surely. Therefore, in order to show $\sigma_t^2 \overset{a.s.}{\to} 0$ as $t \to \infty$, we only need to show

$$S_t := \sum_{i=0}^{t-1} \left(\frac{\varphi_i}{\sqrt{\varphi_{t-1}}}\|\Pi_{t,i+1}\|\right)^{2+\delta} \to 0 \quad \text{as } t \to \infty.$$

Again, similar to the argument in the proof of **(a)**, there exists $j \in \mathbb{N}$ such that $1 - \varphi_i\lambda_M > 0$, $\forall i > j$. Therefore, for $t$ large enough, we have

$$S_t \le \underbrace{\frac{1}{\varphi_{t-1}^{1+\delta/2}} \sum_{i=0}^{j} \varphi_i^{2+\delta} \|\Pi_{j+1,i+1}\|^{2+\delta} \cdot \prod_{k=j+1}^{t-1}(1-\varphi_k\lambda_m)^{2+\delta}}_{a_t'} + \underbrace{\frac{1}{\varphi_{t-1}^{1+\delta/2}} \sum_{i=j+1}^{t-1} \varphi_i^{2+\delta} \prod_{k=i+1}^{t-1}(1-\varphi_k\lambda_m)^{2+\delta}}_{b_t'}.$$

We analyze the two sequences $(a_t')$ and $(b_t')$ separately. For $(a_t')$, we have

$$a_t' = \frac{1}{\varphi_{t-1}^{1+\delta/2}} \sum_{i=0}^{j} \varphi_i^{2+\delta} \|\Pi_{j+1,i+1}\|^{2+\delta} \cdot \prod_{k=j+1}^{t-1}(1-\varphi_k\lambda_m)^{2+\delta}$$

$$\le \frac{1}{\varphi_{t-1}^{1+\delta/2}} \left(\sum_{i=0}^{j}\varphi_i^{2+\delta}\right) \left(\max_{0\le i\le j}\|\Pi_{j+1,i+1}\|^{2+\delta}\right) \cdot \prod_{k=j+1}^{t-1}(1-\varphi_k\lambda_m)^{2+\delta}$$

$$\lesssim \frac{1}{\varphi_{t-1}^{1+\delta/2}} \prod_{k=j+1}^{t-1}\exp(-(2+\delta)\varphi_k\lambda_m) \overset{(B.39)}{\lesssim} \exp((2+\delta)d_t).$$

As discussed in the proof of **(a)**, $d_t \to -\infty$ as $t \to \infty$. Thus, $a_t' \to 0$ as $t \to \infty$. For $(b_t')$, we have the following recursive relation:

$$b_t' = \frac{1}{\varphi_{t-1}^{1+\delta/2}} \sum_{i=j+1}^{t-1} \varphi_i^{2+\delta} \prod_{k=i+1}^{t-1}(1-\varphi_k\lambda_m)^{2+\delta} = \frac{1}{\varphi_{t-1}^{1+\delta/2}}\left[\varphi_{t-1}^{2+\delta} + \sum_{i=j+1}^{t-2}\varphi_i^{2+\delta}\prod_{k=i+1}^{t-2}(1-\varphi_k\lambda_m)^{2+\delta}\cdot(1-\varphi_{t-1}\lambda_m)^{2+\delta}\right],$$

$$\implies \varphi_{t-1}^{1+\delta/2}b_t' = (1-\varphi_{t-1}\lambda_m)^{2+\delta}\varphi_{t-2}^{1+\delta/2}b_{t-1}' + \varphi_{t-1}^{2+\delta}.$$

By Leluc & Portier (2023, Lemma 3), we directly obtain

$$\limsup_{t\to\infty} \frac{\varphi_{t-1}^{1+\delta/2}b_t'}{(1/t^\varphi)^{1+\delta/2}} \le \begin{cases} \frac{1}{(2+\delta)C_\varphi\lambda_m}\limsup_{t\to\infty}(\frac{1}{t^\varphi})^{\delta/2}, & \varphi\in(0.5,1), \\ \frac{1}{(2+\delta)C_\varphi\lambda_m-(1+\delta/2)}\limsup_{t\to\infty}(\frac{1}{t^\varphi})^{\delta/2}, & \varphi=1, \end{cases} \implies \lim_{t\to\infty}b_t' = 0.$$

Combining the above four displays, we can conclude that $S_t \to 0$ and hence $\sigma_t^2 \overset{a.s.}{\to} 0$ as $t \to \infty$ by (B.43). Therefore, all the conditions of Lemma B.5 are satisfied, and we conclude that

$$\mathcal{I}_{2,t} = \frac{1}{\sqrt{\varphi_{t-1}}} \sum_{i=0}^{t-1} \varphi_i \Pi_{t,i+1} \boldsymbol{\theta}_i \mathbf{1}_{\mathcal{A}_{i-1}} \overset{d}{\to} \mathcal{N}(\mathbf{0}, \Sigma^*),$$

where $\Sigma^*$ satisfies $(\mathcal{R} - \zeta I)\Sigma^* + \Sigma^*(\mathcal{R} - \zeta I) = \Gamma^\star$.

• **Proof of (c).** By arguments analogous to (B.24), it suffices to show that

$$\mathbb{E}[\|\mathcal{I}_{3,t}\|\mathbf{1}_{\{\tau_{k,r}>t\}}] = \frac{1}{\sqrt{\varphi_t}} \mathbb{E}\left[\left\|\sum_{i=0}^{t-1} \varphi_i \Pi_{t,i+1} \boldsymbol{\delta}_i\right\| \mathbf{1}_{\{\tau_{k,r}>t\}}\right] \to 0 \quad \text{as } t \to \infty.$$

Here, the stopping time $\tau_{k,r}$ is defined as in Section B.3, with the same choice of $\eta$ and the corresponding parameters $r$ and $k$. For simplicity, let

$$e_t := \mathbb{E}\left[\left\|\sum_{i=0}^{t-1} \varphi_i \Pi_{t,i+1} \boldsymbol{\delta}_i\right\| \mathbf{1}_{\{\tau_{k,r}>t\}}\right].$$

For all sufficiently large $t$ (in particular, for $t \geq k$), the sequence $\{e_t\}$ satisfies the following recursion:

$$\begin{aligned}
e_t &\leq \mathbb{E}\left[\|\varphi_{t-1}\boldsymbol{\delta}_{t-1}\|\mathbf{1}_{\{\tau_{k,r}>t\}}\right] + \mathbb{E}\left[\left\|(I - \varphi_{t-1}\mathcal{R})\left(\sum_{i=0}^{t-2} \varphi_i \Pi_{t-1,i+1} \boldsymbol{\delta}_i\right)\right\| \mathbf{1}_{\{\tau_{k,r}>t\}}\right] \\
&\leq \varphi_{t-1}\mathbb{E}\left[\|\boldsymbol{\delta}_{t-1}\|\mathbf{1}_{\{\tau_{k,r}>t-1\}}\right] + (1 - \varphi_{t-1}\lambda_m)\mathbb{E}\left[\left\|\sum_{i=0}^{t-2} \varphi_i \Pi_{t-1,i+1} \boldsymbol{\delta}_i\right\| \mathbf{1}_{\{\tau_{k,r}>t-1\}}\right] \\
&\leq (1 - \varphi_{t-1}\lambda_m)e_{t-1} + \varphi_{t-1}\mathbb{E}\left[\|\boldsymbol{\delta}_{t-1}\|\mathbf{1}_{\{\tau_{k,r}>t-1\}}\right].
\end{aligned} \tag{B.44}$$

To characterize the asymptotic behavior of $e_t$, it remains to bound $\mathbb{E}\left[\|\boldsymbol{\delta}_t\|\mathbf{1}_{\{\tau_{k,r}>t\}}\right]$. By (B.22) and the Cauchy-Schwarz inequality, we have

$$\begin{aligned}
\mathbb{E}[\|\boldsymbol{\delta}_t\|\mathbf{1}_{\{\tau_{k,r}>t\}}] &\lesssim \mathbb{E}\left[\|B_t - B^\star\|^2\mathbf{1}_{\{\tau_{k,r}>t\}}\right] + \mathbb{E}\left[\|\Delta_t\|^2\mathbf{1}_{\{\tau_{k,r}>t\}}\right] \\
&\overset{\substack{(\text{B.25}) \\ (\text{B.33})}}{\lesssim} \mathbb{E}\left[\left\|\frac{1}{t}\sum_{i=0}^{t-1}(H_i - \nabla^2 f_i)\right\|^2\right] + \frac{1}{t^2}\left(\sum_{i=0}^{t-1}\left(\mathbb{E}\left[\|\boldsymbol{x}_i - \boldsymbol{x}^\star\|^2\mathbf{1}_{\{\tau_{k,r}>i\}}\right]\right)^{1/2}\right)^2 + \varphi_t \\
&\overset{\substack{(\text{B.27}) \\ (\text{B.34})}}{\lesssim} \frac{1}{t} + \varphi_t + \varphi_t \lesssim \varphi_t.
\end{aligned}$$

Therefore, the recursive relation in (B.44) can be written as

$$e_t \leq (1 - \varphi_{t-1}\lambda_m)e_{t-1} + \varphi_{t-1}\mathbb{E}[\|\boldsymbol{\delta}_t\|\mathbf{1}_{\{\tau_{k,r}>t\}}] \leq (1 - \varphi_{t-1}\lambda_m)e_{t-1} + O(\varphi_{t-1}^2),$$

which implies

$$\frac{e_t}{\sqrt{\varphi_{t-1}}} \leq (1 - \varphi_{t-1}\lambda_m)\frac{e_{t-1}}{\sqrt{\varphi_{t-2}}}\sqrt{\frac{\varphi_{t-2}}{\varphi_{t-1}}} + \frac{O(\varphi_{t-1}^2)}{\sqrt{\varphi_{t-1}}} \leq (1 - \varphi_{t-1}\lambda_m)\left(1 + \frac{\varphi}{2t} + O(1/t^2)\right)\frac{e_{t-1}}{\sqrt{\varphi_{t-2}}} + O(\varphi_{t-1}^{3/2}).$$

•• **Case 1:** $\varphi \in (0.5, 1)$. Given that $1/t = o(\varphi_{t-1})$, there exists $\eta_1 \in (0, 1)$ such that

$$(1 - \varphi_{t-1}\lambda_m)\left(1 + \frac{\varphi}{2t} + O(1/t^2)\right) \leq 1 - \varphi_{t-1}\lambda_m(1 - \eta_1)$$

for all $t$ large enough. Therefore, we have

$$\frac{e_t}{\sqrt{\varphi_{t-1}}} \leq (1 - \varphi_{t-1}\lambda_m(1 - \eta_1))\frac{e_{t-1}}{\sqrt{\varphi_{t-2}}} + O(\varphi_{t-1}^{3/2}).$$

By Leluc & Portier (2023, Lemma 3), we obtain

$$\limsup_{t\to\infty} \frac{e_t}{\sqrt{\varphi_{t-1}}} \lesssim \frac{1}{C_\varphi \lambda_m (1 - \eta_1)} \cdot \limsup_{t\to\infty} \sqrt{\varphi_{t-1}}, \implies \lim_{t\to\infty} \frac{e_t}{\sqrt{\varphi_{t-1}}} = 0.$$

•• **Case 2:** $\varphi = 1$. Given $C_\varphi > 1/(2\lambda_m)$, we have

$$(1 - \varphi_{t-1}\lambda_m)\left(1 + \frac{1}{2t} + O(1/t^2)\right) = 1 - \frac{1}{t}\left(C_\varphi \lambda_m - \frac{1}{2}\right) + O(1/t^2).$$

There exists $\eta_2 \in (0, 1)$ such that

$$(1 - \varphi_{t-1}\lambda_m)\left(1 + \frac{\varphi}{2t} + O(1/t^2)\right) \leq 1 - \frac{1}{t}\left(C_\varphi \lambda_m - \frac{1}{2}\right)(1 - \eta_2)$$

for all $t$ large enough. Therefore, we have the following recursion:

$$\frac{e_t}{\sqrt{\varphi_{t-1}}} \leq \left(1 - \frac{1}{t}\left(C_\varphi \lambda_m - \frac{1}{2}\right)(1 - \eta_2)\right)\frac{e_{t-1}}{\sqrt{\varphi_{t-2}}} + O(\varphi_{t-1}^{3/2}).$$

Again by Leluc & Portier (2023, Lemma 3), we obtain

$$\limsup_{t\to\infty} \frac{e_t}{\sqrt{\varphi_{t-1}}} \lesssim \frac{1}{\left(C_\varphi \lambda_m - \frac{1}{2}\right)(1 - \eta_2)} \cdot \limsup_{t\to\infty} \sqrt{\varphi_{t-1}}, \implies \lim_{t\to\infty} \frac{e_t}{\sqrt{\varphi_{t-1}}} = 0.$$

Combining **Case 1** and **Case 2**, we conclude that $\mathcal{I}_{3,t} \xrightarrow{p} \mathbf{0}$ as $t \to \infty$.

• **Proof of (d).** By Lemma B.6, we know for each run of the algorithm, there exists an (potentially random) index $J < \infty$ such that $\mathbf{1}_{\mathcal{A}_{i-1}^c} = 0$ for all $i > J$. Thus, for such a realization, we have for $t > J$ that

$$\left\|\mathcal{I}_{4,t}\right\| \leq \frac{1}{\sqrt{\varphi_{t-1}}} \sum_{i=0}^{J} \varphi_i \|\Pi_{t,i+1}\| \|\boldsymbol{\theta}_i\| \lesssim \left(\sum_{i=0}^{J} \frac{1}{\sqrt{\varphi_{t-1}}} \|\Pi_{t,i+1}\|\right)\left(\max_{0 \leq i \leq J} \|\boldsymbol{\theta}_i\|\right).$$

As in the proof of **(a)**, for any $0 \leq i \leq J$, we have $\|\Pi_{t,i+1}\|/\sqrt{\varphi_{t-1}} \to 0$ as $t \to \infty$. This concludes that $\mathcal{I}_{4,t} \xrightarrow{a.s.} \mathbf{0}$ as $t \to \infty$.

Now, we have proved **(a)-(d)** and, by (B.38), we complete the proof of Theorem 4.3.

### B.7. Proof of Lemma B.6

Recall from (B.20a) that $\boldsymbol{\theta}_t = -(I - K_t)B_t^{-1}(g_t - \nabla f_t) + (\boldsymbol{z}_{t,\tau} - (I - K_t)\Delta\boldsymbol{x}_t)$ forms a martingale difference sequence. We now investigate the limiting covariance of $\mathbb{E}[\boldsymbol{\theta}_t\boldsymbol{\theta}_t^\top \mid \mathcal{F}_{t-1}]$. Note that

$$\begin{aligned}
\mathbb{E}[\boldsymbol{\theta}_t\boldsymbol{\theta}_t^\top \mid \mathcal{F}_{t-1}] &= \mathbb{E}[\{(I - K_t)B_t^{-1}(g_t - \nabla f_t) - (\boldsymbol{z}_{t,\tau} - (I - K_t)\Delta\boldsymbol{x}_t)\} \\
&\quad \{(I - K_t)B_t^{-1}(g_t - \nabla f_t) - (\boldsymbol{z}_{t,\tau} - (I - K_t)\Delta\boldsymbol{x}_t)\}^\top \mid \mathcal{F}_{t-1}] \\
&= (I - K_t)B_t^{-1}\mathbb{E}[(g_t - \nabla f_t)(g_t - \nabla f_t)^\top \mid \mathcal{F}_{t-1}]B_t^{-1}(I - K_t) \\
&\quad + \mathbb{E}[(\boldsymbol{z}_{t,\tau} - (I - K_t)\Delta\boldsymbol{x}_t)(\boldsymbol{z}_{t,\tau} - (I - K_t)\Delta\boldsymbol{x}_t)^\top \mid \mathcal{F}_{t-1}] := \mathcal{J}_{1,t} + \mathcal{J}_{2,t},
\end{aligned} \tag{B.45}$$

where the second equality is due to the tower property and the fact that $\boldsymbol{z}_{t,\tau} - (I - K_t)\Delta\boldsymbol{x}_t$ forms a martingale difference sequence given in Lemma 3.5. Specifically,

$$\begin{aligned}
\mathbb{E}[(g_t - \nabla f_t)(\boldsymbol{z}_{t,\tau} - (I - K_t)\Delta\boldsymbol{x}_t)^\top \mid \mathcal{F}_{t-1}] &= \mathbb{E}[\mathbb{E}[(g_t - \nabla f_t)(\boldsymbol{z}_{t,\tau} - (I - K_t)\Delta\boldsymbol{x}_t)^\top \mid \mathcal{F}_{t-0.5}] \mid \mathcal{F}_{t-1}] \\
&= \mathbb{E}[(g_t - \nabla f_t)\mathbb{E}[(\boldsymbol{z}_{t,\tau} - (I - K_t)\Delta\boldsymbol{x}_t)^\top \mid \mathcal{F}_{t-0.5}] \mid \mathcal{F}_{t-1}] = \mathbf{0}.
\end{aligned}$$

• **Convergence of the $\mathcal{J}_{1,t}$.** For the term $\mathcal{J}_{1,t}$, we have $\mathbb{E}[(g_t - \nabla f_t)(g_t - \nabla f_t)^\top \mid \mathcal{F}_{t-1}] = \mathbb{E}[g_t g_t^\top \mid \mathcal{F}_{t-1}] - \nabla f_t \nabla f_t^\top$. Also note that

$$\begin{aligned}
\left\|\mathbb{E}[g_t g_t^\top \mid \mathcal{F}_{t-1}] - \mathbb{E}[\nabla F(\boldsymbol{x}^\star; \xi)(\nabla F(\boldsymbol{x}^\star; \xi))^\top]\right\| &= \left\|\mathbb{E}[g_t g_t^\top - \nabla F(\boldsymbol{x}^\star; \xi_t)(\nabla F(\boldsymbol{x}^\star; \xi_t))^\top \mid \mathcal{F}_{t-1}]\right\| \\
&\leq 2\mathbb{E}[\|g_t - \nabla F(\boldsymbol{x}^\star; \xi_t)\| \cdot \|g_t\| \mid \mathcal{F}_{t-1}] + \mathbb{E}[\|g_t - \nabla F(\boldsymbol{x}^\star; \xi_t)\|^2 \mid \mathcal{F}_{t-1}] \\
&\leq 2\sqrt{\mathbb{E}[\|g_t - \nabla F(\boldsymbol{x}^\star; \xi_t)\|^2 \mid \mathcal{F}_{t-1}]}\sqrt{\mathbb{E}[\|g_t\|^2 \mid \mathcal{F}_{t-1}]} + \mathbb{E}[\|g_t - \nabla F(\boldsymbol{x}^\star; \xi_t)\|^2 \mid \mathcal{F}_{t-1}],
\end{aligned} \tag{B.46}$$

where the first equality holds since $\xi_t$ is independent of $\mathcal{F}_{t-1}$, the second inequality uses $\|aa^\top - bb^\top\| \le 2\|a\|\|a-b\| + \|a-b\|^2$ and the last inequality follows from the Cauchy–Schwarz inequality. For the first term in (B.46), by the $\Upsilon_H$-Lipschitz continuity of $\nabla f(\boldsymbol{x})$ and Assumption 3.2 with $q_g > 2$, we have

$$\mathbb{E}[\|g_t - \nabla F(\boldsymbol{x}^\star; \xi_t)\|^2 \mid \mathcal{F}_{t-1}] \le \mathbb{E}[\sup_{\boldsymbol{x}} \|\nabla^2 F(\boldsymbol{x}; \xi)\|^2] \cdot \|\boldsymbol{x}_t - \boldsymbol{x}^\star\|^2 \le \Upsilon_H^2 \|\boldsymbol{x}_t - \boldsymbol{x}^\star\|^2, \tag{B.47}$$

and

$$\mathbb{E}[\|g_t\|^2 \mid \mathcal{F}_{t-1}] \le 2\|\nabla f_t\|^2 + 2\mathbb{E}[\|g_t - \nabla f_t\|^2 \mid \mathcal{F}_{t-1}] \le 2\Upsilon_H^2 \|\boldsymbol{x}_t - \boldsymbol{x}^\star\|^2 + 2C_{g,1}^{2/q_g} \|\boldsymbol{x}_t - \boldsymbol{x}^\star\|^2 + 2C_{q_g,2}^{2/q_g}. \tag{B.48}$$

Here the second inequality in (B.48) also uses $2/q_g \in (0,1)$ and hence $(a+b)^{2/q_g} \le a^{2/q_g} + b^{2/q_g}$ for $a, b \ge 0$. Plugging (B.47) and (B.48) into (B.46) gives us

$$\left\| \mathbb{E}[g_t g_t^\top \mid \mathcal{F}_{t-1}] - \mathbb{E}[\nabla F(\boldsymbol{x}^\star; \xi_t)(\nabla F(\boldsymbol{x}^\star; \xi_t))^\top] \right\| \overset{a.s.}{\to} 0,$$

which further implies

$$\mathbb{E}[(g_t - \nabla f_t)(g_t - \nabla f_t)^\top \mid \mathcal{F}_{t-1}] \overset{a.s.}{\to} \mathbb{E}[\nabla F(\boldsymbol{x}^\star; \xi)(\nabla F(\boldsymbol{x}^\star; \xi))^\top]. \tag{B.49}$$

Since $K_t \overset{a.s.}{\to} K^\star$ and $B_t \overset{a.s.}{\to} B^\star$, which are shown in Lemma 4.1, we then have

$$\mathcal{J}_{1,t} \overset{a.s.}{\to} (I - K^\star)(B^\star)^{-1} \mathbb{E}[\nabla F(\boldsymbol{x}^\star; \xi)(\nabla F(\boldsymbol{x}^\star; \xi))^\top](B^\star)^{-1}(I - K^\star) \overset{(4.1)}{=} (I - K^\star)\Omega^\star(I - K^\star). \tag{B.50}$$

- **Convergence of the $\mathcal{J}_{2,t}$.** For the term $\mathcal{J}_{2,t}$, we have

$$\begin{aligned}
\mathcal{J}_{2,t} &= \mathbb{E}[(\boldsymbol{z}_{t,\tau} - (I - K_t)\Delta\boldsymbol{x}_t)(\boldsymbol{z}_{t,\tau} - (I - K_t)\Delta\boldsymbol{x}_t)^\top \mid \mathcal{F}_{t-1}] \\
&= \mathbb{E}[(\widetilde{K}_t - K_t)\Delta\boldsymbol{x}_t \Delta\boldsymbol{x}_t^\top (\widetilde{K}_t - K_t)^\top \mid \mathcal{F}_{t-1}] \\
&= \mathbb{E}[(\widetilde{K}_t - K_t)B_t^{-1} g_t g_t^\top B_t^{-1}(\widetilde{K}_t - K_t)^\top \mid \mathcal{F}_{t-1}] \\
&= \mathbb{E}[(\widetilde{K}_t - K_t)B_t^{-1}(g_t - \nabla f_t)(g_t - \nabla f_t)^\top B_t^{-1}(\widetilde{K}_t - K_t)^\top \mid \mathcal{F}_{t-1}] \\
&\quad + \mathbb{E}[(\widetilde{K}_t - K_t)B_t^{-1} \nabla f_t \nabla f_t^\top B_t^{-1}(\widetilde{K}_t - K_t)^\top \mid \mathcal{F}_{t-1}] \\
&\quad + \mathbb{E}[(\widetilde{K}_t - K_t)B_t^{-1}(g_t - \nabla f_t)\nabla f_t^\top B_t^{-1}(\widetilde{K}_t - K_t)^\top \mid \mathcal{F}_{t-1}] \\
&\quad + \mathbb{E}[(\widetilde{K}_t - K_t)B_t^{-1} \nabla f_t(g_t - \nabla f_t)^\top B_t^{-1}(\widetilde{K}_t - K_t)^\top \mid \mathcal{F}_{t-1}].
\end{aligned} \tag{B.51}$$

For the last two terms in (B.51), we can apply the tower property. In particular, for the last term, we have

$$\begin{aligned}
&\mathbb{E}[(\widetilde{K}_t - K_t)B_t^{-1} \nabla f_t(g_t - \nabla f_t)^\top B_t^{-1}(\widetilde{K}_t - K_t)^\top \mid \mathcal{F}_{t-1}] \\
&= \mathbb{E}[(\widetilde{K}_t - K_t)B_t^{-1} \nabla f_t \mathbb{E}[(g_t - \nabla f_t)^\top \mid \mathcal{F}_{t-1} \cup \sigma(\{S_{t,j}\}_j)]B_t^{-1}(\widetilde{K}_t - K_t)^\top \mid \mathcal{F}_{t-1}] \\
&= \mathbb{E}[(\widetilde{K}_t - K_t)B_t^{-1} \nabla f_t \mathbb{E}[(g_t - \nabla f_t)^\top \mid \mathcal{F}_{t-1}]B_t^{-1}(\widetilde{K}_t - K_t)^\top \mid \mathcal{F}_{t-1}] = \boldsymbol{0},
\end{aligned} \tag{B.52}$$

where the last equality holds since $\mathbb{E}[g_t - \nabla f_t \mid \mathcal{F}_{t-1}] = 0$. Similarly, for the third term in (B.51), we have

$$\begin{aligned}
&\mathbb{E}[(\widetilde{K}_t - K_t)B_t^{-1}(g_t - \nabla f_t)\nabla f_t^\top B_t^{-1}(\widetilde{K}_t - K_t)^\top \mid \mathcal{F}_{t-1}] \\
&= \mathbb{E}[(\widetilde{K}_t - K_t)B_t^{-1}\mathbb{E}[(g_t - \nabla f_t) \mid \mathcal{F}_{t-1} \cup \sigma(\{S_{t,j}\}_j)]\nabla f_t^\top B_t^{-1}(\widetilde{K}_t - K_t)^\top \mid \mathcal{F}_{t-1}] \\
&= \mathbb{E}[(\widetilde{K}_t - K_t)B_t^{-1}\mathbb{E}[(g_t - \nabla f_t) \mid \mathcal{F}_{t-1}]\nabla f_t^\top B_t^{-1}(\widetilde{K}_t - K_t)^\top \mid \mathcal{F}_{t-1}] = \boldsymbol{0}.
\end{aligned} \tag{B.53}$$

For the second term in (B.51), we have

$$\begin{aligned}
&\left\| \mathbb{E}[(\widetilde{K}_t - K_t)B_t^{-1} \nabla f_t \nabla f_t^\top B_t^{-1}(\widetilde{K}_t - K_t)^\top \mid \mathcal{F}_{t-1}] \right\| \le \mathbb{E}[\|(\widetilde{K}_t - K_t)B_t^{-1} \nabla f_t \nabla f_t^\top B_t^{-1}(\widetilde{K}_t - K_t)^\top\| \mid \mathcal{F}_{t-1}] \\
&\le \mathbb{E}[\|(\widetilde{K}_t - K_t)\|^2 \|B_t^{-1}\|^2 \|\nabla f_t\|^2 \mid \mathcal{F}_{t-1}] \le (C_K + 0.5)^2 \cdot 1/\gamma_H^2 \cdot \mathbb{E}[\|\nabla f_t\|^2 \mid \mathcal{F}_{t-1}] \\
&= (C_K + 0.5)^2 \cdot 1/\gamma_H^2 \cdot \|\nabla f_t\|^2 \overset{a.s.}{\to} 0.
\end{aligned} \tag{B.54}$$

Here, the third inequality follows from the uniform bounds $\|\widetilde{K}_t\| \le C_K$ in (A.28), $\|K_t\| < 0.5$ by Lemma 3.7 together with the condition on $\tau$ that $\tau(1 - \sqrt{\mu_t/\nu_t})^{\tau-2} \le \gamma_H/(4\Upsilon_H)$, and $\|B_t^{-1}\| \le 1/\gamma_H$ in (A.26). For the first term in (B.51), we need the following technical lemma to proceed.

**Lemma B.7.** *Let* $\Omega_t := B_t^{-1}\mathbb{E}[(g_t - \nabla f_t)(g_t - \nabla f_t)^\top \mid \mathcal{F}_{t-1}]B_t^{-1}$. *Under the conditions of Theorem 4.3, we have the following convergence as* $t \to \infty$:

$$\mathbb{E}[(\widetilde{K}_t - K_t)\Omega_t(\widetilde{K}_t - K_t)^\top \mid \mathcal{F}_{t-1}] \overset{a.s.}{\to} \mathbb{E}[(\widetilde{K}^\star - K^\star)\Omega^\star(\widetilde{K}^\star - K^\star)^\top].$$

By Lemma B.7, we have

$$\begin{aligned}
&\mathbb{E}[(\widetilde{K}_t - K_t)B_t^{-1}(g_t - \nabla f_t)(g_t - \nabla f_t)^\top B_t^{-1}(\widetilde{K}_t - K_t)^\top \mid \mathcal{F}_{t-1}] \\
&= \mathbb{E}[(\widetilde{K}_t - K_t)B_t^{-1}\mathbb{E}[(g_t - \nabla f_t)(g_t - \nabla f_t)^\top \mid \mathcal{F}_{t-1} \cup \sigma(\{S_{t,j}\}_j)]B_t^{-1}(\widetilde{K}_t - K_t)^\top \mid \mathcal{F}_{t-1}] \\
&= \mathbb{E}[(\widetilde{K}_t - K_t)B_t^{-1}\mathbb{E}[(g_t - \nabla f_t)(g_t - \nabla f_t)^\top \mid \mathcal{F}_{t-1}]B_t^{-1}(\widetilde{K}_t - K_t)^\top) \mid \mathcal{F}_{t-1}] \\
&\overset{a.s.}{\to} \mathbb{E}[(\widetilde{K}^\star - K^\star)\Omega^\star(\widetilde{K}^\star - K^\star)^\top] = \mathbb{E}[\widetilde{K}^\star\Omega^\star(\widetilde{K}^\star)^\top] - K^\star\Omega^\star K^\star. && \text{(B.55)}
\end{aligned}$$

Here, $\widetilde{K}^\star$ is defined analogously to $\widetilde{K}_t$ in (3.2), but evaluated at $(\alpha^\star, \beta^\star, \gamma^\star)$ and $B^\star$; see Section 4.1 for more details. As a result, substituting (B.52), (B.53), (B.54) and (B.55) into (B.51) yields

$$\mathcal{J}_{2,t} \overset{a.s.}{\to} \mathbb{E}[\widetilde{K}^\star\Omega^\star(\widetilde{K}^\star)^\top] - K^\star\Omega^\star K^\star. \tag{B.56}$$

Combining (B.50) and (B.56) with (B.45), we obtain

$$\mathbb{E}[\boldsymbol{\theta}_t\boldsymbol{\theta}_t^\top \mid \mathcal{F}_{t-1}] \overset{a.s.}{\to} \mathbb{E}[(I - \widetilde{K}^\star)\Omega^\star(I - \widetilde{K}^\star)^\top],$$

where we use the definition $K^\star := \mathbb{E}[\widetilde{K}^\star]$. This completes the proof.

## B.8. Proof of Lemma B.7

By (B.49) and Lemma 4.1, we know $\Omega_t \to \Omega^\star$ almost surely. Therefore, we have

$$\begin{aligned}
&\left\| \mathbb{E}[(\widetilde{K}_t - K_t)\Omega_t(\widetilde{K}_t - K_t)^\top \mid \mathcal{F}_{t-1}] - \mathbb{E}[(\widetilde{K}^\star - K^\star)\Omega^\star(\widetilde{K}^\star - K^\star)^\top] \right\| \\
&\leq \left\| \mathbb{E}[(\widetilde{K}^\star - K^\star)\Omega_t(\widetilde{K}^\star - K^\star)^\top \mid \mathcal{F}_{t-1}] - \mathbb{E}[(\widetilde{K}_t - K_t)\Omega^\star(\widetilde{K}_t - K_t)^\top \mid \mathcal{F}_{t-1}] \right\| \\
&\quad + \left\| \mathbb{E}[(\widetilde{K}_t - K_t)\Omega^\star(\widetilde{K}_t - K_t)^\top \mid \mathcal{F}_{t-1}] - \mathbb{E}[(\widetilde{K}^\star - K^\star)\Omega^\star(\widetilde{K}^\star - K^\star)^\top] \right\|. && \text{(B.57)}
\end{aligned}$$

For the first term on the right-hand side of (B.57), we have

$$\begin{aligned}
&\left\| \mathbb{E}[(\widetilde{K}_t - K_t)\Omega_t(\widetilde{K}_t - K_t)^\top \mid \mathcal{F}_{t-1}] - \mathbb{E}[(\widetilde{K}_t - K_t)\Omega^\star(\widetilde{K}_t - K_t)^\top \mid \mathcal{F}_{t-1}] \right\| \\
&= \left\| \mathbb{E}[(\widetilde{K}_t - K_t)(\Omega_t - \Omega^\star)(\widetilde{K}_t - K_t)^\top \mid \mathcal{F}_{t-1}] \right\| \\
&\leq \mathbb{E}\left[ \|(\widetilde{K}_t - K_t)(\Omega_t - \Omega^\star)(\widetilde{K}_t - K_t)^\top\| \mid \mathcal{F}_{t-1} \right] \\
&\leq (C_K + 0.5)^2 \cdot \mathbb{E}\left[\|\Omega_t - \Omega^\star\| \mid \mathcal{F}_{t-1}\right] = (C_K + 0.5)^2 \|\Omega_t - \Omega^\star\| \overset{a.s.}{\to} 0. && \text{(B.58)}
\end{aligned}$$

Here, the third inequality follows from $\|\widetilde{K}_t\| \leq C_K$ and $\|K_t\| < 0.5$ (under $\tau(1 - \sqrt{\mu_t/\nu_t})^{\tau-2} \leq \gamma_H/(4\Upsilon_H)$). For the second term on the right-hand side of (B.57), we have

$$\begin{aligned}
&\left\| \mathbb{E}[(\widetilde{K}_t - K_t)\Omega^\star(\widetilde{K}_t - K_t)^\top \mid \mathcal{F}_{t-1}] - \mathbb{E}[(\widetilde{K}^\star - K^\star)\Omega^\star(\widetilde{K}^\star - K^\star)^\top] \right\| \\
&\leq \left\| \mathbb{E}[(\widetilde{K}_t - K_t)\Omega^\star(\widetilde{K}_t - K_t)^\top \mid \mathcal{F}_{t-1}] - \mathbb{E}[(\widetilde{K}^\star - K^\star)\Omega^\star(\widetilde{K}_t - K_t)^\top \mid \mathcal{F}_{t-1}] \right\| \\
&\quad + \left\| \mathbb{E}[(\widetilde{K}^\star - K^\star)\Omega^\star(\widetilde{K}_t - K_t)^\top \mid \mathcal{F}_{t-1}] - \mathbb{E}[(\widetilde{K}^\star - K^\star)\Omega^\star(\widetilde{K}^\star - K^\star)^\top] \right\| \\
&\leq \|\Omega^\star\| \cdot (C_K + 0.5) \cdot (\mathbb{E}[\|\widetilde{K}_t - \widetilde{K}^\star\| \mid \mathcal{F}_{t-1}] + \|K_t - K^\star\|) \\
&\quad + \|\Omega^\star\| \cdot (C_K + 0.5) \cdot (\mathbb{E}[\|\widetilde{K}_t - \widetilde{K}^\star\| \mid \mathcal{F}_{t-1}] + \|K_t - K^\star\|).
\end{aligned}$$

By Lemma 4.1, we know $K_t \overset{a.s.}{\to} K^\star$ as $t \to \infty$. Also, in the same spirit as the proof of Lemma 4.1 (see also (B.99)), we obtain $\mathbb{E}[\|\widetilde{K}_t - \widetilde{K}^\star\| \mid \mathcal{F}_{t-1}] \lesssim \|B_t - B^\star\| \overset{a.s.}{\to} 0$. Thus, the above display leads to

$$\left\| \mathbb{E}[(\widetilde{K}_t - K_t)\Omega^\star(\widetilde{K}_t - K_t)^\top \mid \mathcal{F}_{t-1}] - \mathbb{E}[(\widetilde{K}^\star - K^\star)\Omega^\star(\widetilde{K}^\star - K^\star)^\top] \right\| \overset{a.s.}{\to} 0. \tag{B.59}$$

Substituting (B.58) and (B.59) into (B.57) yields

$$\mathbb{E}[(\widetilde{K}_t - K_t)\Omega_t(\widetilde{K}_t - K_t)^\top \mid \mathcal{F}_{t-1}] \overset{a.s.}{\to} \mathbb{E}[(\widetilde{K}^\star - K^\star)\Omega^\star(\widetilde{K}^\star - K^\star)^\top] \quad \text{as } t \to \infty.$$

This completes the proof.

## B.9. Proof of Proposition 4.4

• **Proof of (a).** We plug $\widetilde{K}^\star = K^\star = \mathbf{0}$ into (4.2) and have $[(1 - \zeta)I]\Sigma^\star + \Sigma^\star[(1 - \zeta)I] = \Omega^\star$. Thus, for $\varphi \in (0.5, 1)$, $\zeta = 0$, then $\Sigma^\star = 0.5\Omega^\star$. For $\varphi = 1$, $\zeta = 1/(2C_\varphi)$, then $\Sigma^\star = \frac{C_\varphi \Omega^\star}{2C_\varphi - 1}$. Furthermore, when $\varphi = C_\varphi = 1$, we have $\Sigma^\star = \Omega^\star$. Then, we apply the result into Theorem 4.3, and get that

$$\sqrt{t} \cdot (\boldsymbol{x}_t - \boldsymbol{x}^\star) \overset{d}{\to} \mathcal{N}(\mathbf{0}, \Omega^\star).$$

We complete the proof of (a).

• **Proof of (b).** When $\gamma^\star = 1$, we follow the similar steps in the proof of Lemma 3.5, and get that $\widetilde{K}^\star = \prod_{j=0}^{\tau-1}(I - B^\star S_j(S_j^\top (B^\star)^2 S_j)^\dagger S_j^\top B^\star) = \widetilde{\mathcal{C}}^\star$, and $K^\star := \mathbb{E}[\widetilde{K}^\star] = \mathbb{E}[\widetilde{\mathcal{C}}^\star]$. We apply these results to (4.2), and finally get

$$[(I - \mathcal{C}^\star) - \zeta I]\Sigma^\star + \Sigma^\star[(I - \mathcal{C}^\star) - \zeta I] = \mathcal{G}^\star.$$

We complete the proof of (b).

## B.10. Proof of Lemma 4.5

The proof is composed of two parts.

• **Part 1: Bound of $\mathbb{E}[\|\boldsymbol{x}_t - \boldsymbol{x}^\star\|^4]$.** For (A.24), taking square on both sides and then the conditional expectation given $\mathcal{F}_{t-1}$ yields

$$\mathbb{E}[(f_{t+1} - f^\star)^2 \mid \mathcal{F}_{t-1}] \le (f_t - f^\star)^2 + 2\varphi_t \mathbb{E}[(f_t - f^\star)\nabla f_t^\top \boldsymbol{z}_{t,\tau} \mid \mathcal{F}_{t-1}] + \varphi_t^2 \Upsilon_H \mathbb{E}[(f_t - f^\star)\|\boldsymbol{z}_{t,\tau}\|^2 \mid \mathcal{F}_{t-1}]$$
$$+ \varphi_t^2 \mathbb{E}[(\nabla f_t^\top \boldsymbol{z}_{t,\tau})^2 \mid \mathcal{F}_{t-1}] + \varphi_t^3 \Upsilon_H \mathbb{E}[\nabla f_t^\top \boldsymbol{z}_{t,\tau}\|\boldsymbol{z}_{t,\tau}\|^2 \mid \mathcal{F}_{t-1}] + \frac{\Upsilon_H^2}{4}\varphi_t^4 \mathbb{E}[\|\boldsymbol{z}_{t,\tau}\|^4 \mid \mathcal{F}_{t-1}]. \tag{B.60}$$

We rearrange the terms on the right-hand side by the order of $\varphi_t$ and then analyze each of them.

•• **Term 1:** $2\varphi_t \mathbb{E}[(f_t - f^\star)\nabla f_t^\top \boldsymbol{z}_{t,\tau} \mid \mathcal{F}_{t-1}]$. For Term 1, we have

$$2\varphi_t \mathbb{E}[(f_t - f^\star)\nabla f_t^\top \boldsymbol{z}_{t,\tau} \mid \mathcal{F}_{t-1}] \overset{(A.27)}{\le} -\frac{1}{\Upsilon_H}\varphi_t(f_t - f^\star)\|\nabla f_t\|^2 \overset{(A.22)}{\le} -\frac{2\gamma_H}{\Upsilon_H}\varphi_t(f_t - f^\star)^2. \tag{B.61}$$

•• **Term 2:** $\varphi_t^2 \Upsilon_H \mathbb{E}[(f_t - f^\star)\|\boldsymbol{z}_{t,\tau}\|^2 \mid \mathcal{F}_{t-1}] + \varphi_t^2 \mathbb{E}[(\nabla f_t^\top \boldsymbol{z}_{t,\tau})^2 \mid \mathcal{F}_{t-1}]$. For Term 2, we have

$$\varphi_t^2 \Upsilon_H \mathbb{E}[(f_t - f^\star)\|\boldsymbol{z}_{t,\tau}\|^2 \mid \mathcal{F}_{t-1}] + \varphi_t^2 \mathbb{E}[(\nabla f_t^\top \boldsymbol{z}_{t,\tau})^2 \mid \mathcal{F}_{t-1}] \le \varphi_t^2 \mathbb{E}[(\Upsilon_H(f_t - f^\star) + \|\nabla f_t\|^2)\|\boldsymbol{z}_{t,\tau}\|^2 \mid \mathcal{F}_{t-1}]$$
$$\overset{(A.23)}{\le} 3\Upsilon_H \varphi_t^2(f_t - f^\star)\mathbb{E}[\|\boldsymbol{z}_{t,\tau}\|^2 \mid \mathcal{F}_{t-1}] \le \frac{\gamma_H}{\Upsilon_H}\varphi_t(f_t - f^\star)^2 + \frac{9\Upsilon_H^3}{4\gamma_H}\varphi_t^3 \mathbb{E}[\|\boldsymbol{z}_{t,\tau}\|^4 \mid \mathcal{F}_{t-1}], \tag{B.62}$$

where the last inequality is due to Young's inequality.

•• **Term 3:** $\varphi_t^3 \Upsilon_H \mathbb{E}[\nabla f_t^\top z_{t,\tau} \|z_{t,\tau}\|^2 \mid \mathcal{F}_{t-1}]$. By Young's inequality and (A.23), we obtain

$$
\begin{aligned}
\varphi_t^3 \Upsilon_H \mathbb{E}[\nabla f_t^\top z_{t,\tau} \|z_{t,\tau}\|^2 \mid \mathcal{F}_{t-1}] &\leq \varphi_t^3 \Upsilon_H \mathbb{E}[\|\nabla f_t\| \|z_{t,\tau}\|^3 \mid \mathcal{F}_{t-1}] \\
&\leq \frac{\varphi_t^3 \Upsilon_H}{2} \left( \mathbb{E}[\|\nabla f_t\|^2 \|z_{t,\tau}\|^2 \mid \mathcal{F}_{t-1}] + \mathbb{E}[\|z_{t,\tau}\|^4 \mid \mathcal{F}_{t-1}] \right) \\
&\leq \frac{\varphi_t^3 \Upsilon_H}{2} \left( \frac{1}{2} \mathbb{E}[\|\nabla f_t\|^4 \mid \mathcal{F}_{t-1}] + \frac{1}{2} \mathbb{E}[\|z_{t,\tau}\|^4 \mid \mathcal{F}_{t-1}] + \mathbb{E}[\|z_{t,\tau}\|^4 \mid \mathcal{F}_{t-1}] \right) \\
&= \frac{\varphi_t^3 \Upsilon_H}{4} \mathbb{E}[\|\nabla f_t\|^4 \mid \mathcal{F}_{t-1}] + \frac{3\varphi_t^3 \Upsilon_H}{4} \mathbb{E}[\|z_{t,\tau}\|^4 \mid \mathcal{F}_{t-1}] \\
&\overset{(A.23)}{\leq} \varphi_t^3 \Upsilon_H^3 (f_t - f^\star)^2 + \frac{3\Upsilon_H}{4} \varphi_t^3 \mathbb{E}[\|z_{t,\tau}\|^4 \mid \mathcal{F}_{t-1}].
\end{aligned}
\tag{B.63}
$$

•• **Term 4:** $\mathbb{E}[\|z_{t,\tau}\|^4 \mid \mathcal{F}_{t-1}]$. Following the analysis in (A.29), we apply Assumption 3.2 with $q_g = 4$ and obtain

$$
\begin{aligned}
\mathbb{E}[\|z_{t,\tau}\|^4 \mid \mathcal{F}_{t-1}] &\leq \frac{(1+C_K)^4}{\gamma_H^4} \mathbb{E}[\|g_t\|^4 \mid \mathcal{F}_{t-1}] \leq \frac{8(1+C_K)^4}{\gamma_H^4} (\|\nabla f_t\|^4 + \mathbb{E}[\|g_t - \nabla f_t\|^4 \mid \mathcal{F}_{t-1}]) \\
&\overset{(A.23)}{\leq} \frac{8(1+C_K)^4}{\gamma_H^4} (4\Upsilon_H^2 (f_t - f^\star)^2 + C_{g,1} \|x_t - x^\star\|^4 + C_{g,2}) \\
&\overset{(A.22)}{\leq} \frac{8(1+C_K)^4}{\gamma_H^4} \left( 4\Upsilon_H^2 (f_t - f^\star)^2 + \frac{4C_{g,1}}{\gamma_H^2} (f_t - f^\star)^2 + C_{g,2} \right) \\
&\leq \frac{32(1+C_K)^4 \left[ \Upsilon_H^2 + C_{g,1}/\gamma_H^2 \right]}{\gamma_H^4} (f_t - f^\star)^2 + \frac{8(1+C_K)^4 C_{g,2}}{\gamma_H^4} \\
&=: C_1 (f_t - f^\star)^2 + C_2,
\end{aligned}
\tag{B.64}
$$

where

$$
C_1 := \frac{32(1+C_K)^4 \left[ \Upsilon_H^2 + C_{g,1}/\gamma_H^2 \right]}{\gamma_H^4}, \qquad C_2 := \frac{8(1+C_K)^4 C_{g,2}}{\gamma_H^4}.
$$

Substituting (B.61), (B.62), (B.63) and (B.64) into (B.60), we obtain

$$
\begin{aligned}
\mathbb{E}[(f_{t+1} - f^\star)^2 \mid \mathcal{F}_{t-1}] &\leq (f_t - f^\star)^2 - \left( \frac{\gamma_H}{\Upsilon_H} \varphi_t - \frac{9\Upsilon_H^3 C_1 + 4\Upsilon_H^3 \gamma_H + 3C_1 \Upsilon_H \gamma_H}{4\gamma_H} \varphi_t^3 - \frac{\Upsilon_H^2 C_1}{4} \varphi_t^4 \right) (f_t - f^\star)^2 \\
&\quad + \frac{9\Upsilon_H^3 C_2 + 3C_2 \Upsilon_H \gamma_H}{4\gamma_H} \varphi_t^3 + \frac{\Upsilon_H^2 C_2}{4} \varphi_t^4.
\end{aligned}
\tag{B.65}
$$

Since $\varphi_t = C_\varphi/(t+1)^\varphi$ for $\varphi \in (0.5, 1)$, for $t$ large enough we have

$$
\frac{9\Upsilon_H^3 C_1 + 4\Upsilon_H^3 \gamma_H + 3C_1 \Upsilon_H \gamma_H}{4\gamma_H} \varphi_t^2 + \frac{\Upsilon_H^2 C_1}{4} \varphi_t^3 < \frac{\gamma_H}{2\Upsilon_H}.
\tag{B.66}
$$

Substituting (B.66) into (B.65) and then taking the expectation on both sides gives us

$$
\mathbb{E}\left[ (f_{t+1} - f^\star)^2 \right] \leq \left( 1 - \frac{\gamma_H}{2\Upsilon_H} \varphi_t \right) \mathbb{E}[(f_t - f^\star)^2] + \frac{9\Upsilon_H^3 C_2 + 3\gamma_H \Upsilon_H C_2 + \gamma_H \Upsilon_H^2 C_2 C_\varphi}{4\gamma_H} \varphi_t^3.
$$

Again by Leluc & Portier (2023, Lemma 3), we obtain

$$
\limsup_{t \to \infty} \frac{\mathbb{E}[(f_t - f^\star)^2]}{\varphi_t^2} \lesssim 1 \implies \mathbb{E}[(f_t - f^\star)^2] \lesssim \varphi_t^2.
$$

By (A.22), we further have

$$
\mathbb{E}\left[ \|x_t - x^\star\|^4 \right] \lesssim \mathbb{E}[(f_t - f^\star)^2] \lesssim \varphi_t^2.
\tag{B.67}
$$

• **Part 2: Bound of $\mathbb{E}[\|B_t - B^\star\|^4]$.** By the construction of $B_t$ in (2.1), we have

$$
\mathbb{E}\left[ \|B_t - B^\star\|^4 \right] \lesssim \mathbb{E}\left[ \left\| \frac{1}{t} \sum_{i=0}^{t-1} (H_i - \nabla^2 F_i) \right\|^4 \right] + \mathbb{E}\left[ \left\| \frac{1}{t} \sum_{i=0}^{t-1} (\nabla^2 F_i - \nabla^2 F^\star) \right\|^4 \right].
\tag{B.68}
$$

•• **Term 1:** $\mathbb{E}[\|\frac{1}{t}\sum_{i=0}^{t-1}(H_i - \nabla^2 F_i)\|^4]$. For Term 1, note that $H_i - \nabla^2 F_i$ is a martingale difference sequence and Assumption 3.3 with $q_H = 4$ implies

$$\mathbb{E}\left[\|H_i - \nabla^2 F_i\|_F^4\right] \lesssim \mathbb{E}\left[\|H_i - \nabla^2 F_i\|^4\right] \leq C_{H,1}\mathbb{E}\left[\|\boldsymbol{x}_i - \boldsymbol{x}^\star\|^4\right] + C_{H,2}.$$

Therefore, same as in Chen et al. (2020, (63)), we apply Rio (2009, Theorem 2.1) and obtain

$$\mathbb{E}\left[\left\|\frac{1}{t}\sum_{i=0}^{t-1}(H_i - \nabla^2 F_i)\right\|^4\right] \leq \mathbb{E}\left[\left\|\frac{1}{t}\sum_{i=0}^{t-1}(H_i - \nabla^2 F_i)\right\|_F^4\right] \lesssim \frac{1}{t^4}\left[\sum_{i=0}^{t-1}\left(\mathbb{E}\left[\|H_i - \nabla^2 F_i\|_F^4\right]\right)^{1/2}\right]^2$$

$$\lesssim \frac{1}{t^4}\left(\sum_{i=0}^{t-1}C_{H,1}^{1/2}\left(\mathbb{E}\left[\|\boldsymbol{x}_i - \boldsymbol{x}^\star\|^4\right]\right)^{1/2}\right)^2 + \frac{1}{t^4}\left(\sum_{i=0}^{t-1}C_{H,2}^{1/2}\right)^2$$

$$\overset{(B.67)}{\lesssim} \frac{1}{t^4}\left(\sum_{i=0}^{t-1}\varphi_i\right)^2 + \frac{1}{t^2} \lesssim \frac{1}{t^4}\cdot t^{2(1-\varphi)} + \frac{1}{t^2} \lesssim \frac{1}{t^2}, \tag{B.69}$$

where the second last inequality follows from the fact that $\sum_{i=0}^{t-1}\varphi_i \lesssim \int_0^t 1/i^\varphi\,di \lesssim t^{1-\varphi}$.

•• **Term 2:** $\mathbb{E}[\|\frac{1}{t}\sum_{i=0}^{t-1}(\nabla^2 F_i - \nabla^2 F^\star)\|^4]$. For Term 2, by the $\Upsilon_L$-Lipschitz continuity of $\nabla^2 F(\boldsymbol{x})$, we have

$$\mathbb{E}\left[\left\|\frac{1}{t}\sum_{i=0}^{t-1}(\nabla^2 F_i - \nabla^2 F^\star)\right\|^4\right] \leq \mathbb{E}\left[\left(\frac{1}{t}\sum_{i=0}^{t-1}\|\nabla^2 F_i - \nabla^2 F^\star\|\right)^4\right] \leq \frac{\Upsilon_L^4}{t^4}\mathbb{E}\left[\left(\sum_{i=0}^{t-1}\|\boldsymbol{x}_i - \boldsymbol{x}^\star\|\right)^4\right]$$

$$\leq \frac{\Upsilon_L^4}{t^4}\left(\sum_{i=0}^{t-1}\left(\mathbb{E}\left[\|\boldsymbol{x}_i - \boldsymbol{x}^\star\|^4\right]\right)^{1/4}\right)^4 \quad \text{(by Hölder's inequality)}$$

$$\overset{(B.67)}{\lesssim} \frac{1}{t^4}\left(\sum_{i=0}^{t-1}\sqrt{\varphi_i}\right)^4 \lesssim \frac{1}{t^4}t^{4(1-\varphi/2)} \lesssim \varphi_t^2. \tag{B.70}$$

Plugging (B.69) and (B.70) into (B.68) gives us

$$\mathbb{E}\left[\|B_t - B^\star\|^4\right] \lesssim \frac{1}{t^2} + \varphi_t^2 \lesssim \varphi_t^2.$$

This, together with (B.67), completes the proof for both parts.

### B.11. Proof of Theorem 4.6

Without loss of generality, we suppose $\varphi_t \leq 1, \forall t \geq 0$. The weighted sample covariance matrix $\widehat{\Sigma}_t$ can be decomposed as

$$\widehat{\Sigma}_t = \frac{1}{t}\sum_{i=1}^{t}\frac{1}{\varphi_{i-1}}(\boldsymbol{x}_i - \boldsymbol{x}^\star)(\boldsymbol{x}_i - \boldsymbol{x}^\star)^\top + \frac{1}{t}\sum_{i=1}^{t}\frac{1}{\varphi_{i-1}}(\bar{\boldsymbol{x}}_t - \boldsymbol{x}^\star)(\bar{\boldsymbol{x}}_t - \boldsymbol{x}^\star)^\top$$

$$-\frac{1}{t}\sum_{i=1}^{t}\frac{1}{\varphi_{i-1}}(\boldsymbol{x}_i - \boldsymbol{x}^\star)(\bar{\boldsymbol{x}}_t - \boldsymbol{x}^\star)^\top - \frac{1}{t}\sum_{i=1}^{t}\frac{1}{\varphi_{i-1}}(\bar{\boldsymbol{x}}_t - \boldsymbol{x}^\star)(\boldsymbol{x}_i - \boldsymbol{x}^\star)^\top. \tag{B.71}$$

By (B.38), we have

$$\frac{1}{\sqrt{\varphi_{t-1}}}(\boldsymbol{x}_t - \boldsymbol{x}^\star) = \mathcal{L}_{1,t} + \mathcal{L}_{2,t},$$

where

$$\mathcal{L}_{1,t} = \mathcal{I}_{2,t} + \mathcal{I}_{4,t} = \frac{1}{\sqrt{\varphi_{t-1}}}\sum_{i=0}^{t-1}\varphi_i\Pi_{t,i+1}\boldsymbol{\theta}_i, \qquad \mathcal{L}_{2,t} = \mathcal{I}_{1,t} + \mathcal{I}_{3,t} = \frac{1}{\sqrt{\varphi_{t-1}}}\left(\Pi_{t,0}\Delta_0 + \sum_{i=0}^{t-1}\varphi_i\Pi_{t,i+1}\boldsymbol{\delta}_i\right). \tag{B.72}$$

Therefore, we can further decompose the first term on the right-hand side of (B.71) as follows:

$$\frac{1}{t}\sum_{i=1}^{t}\frac{1}{\varphi_{i-1}}(\boldsymbol{x}_i - \boldsymbol{x}^\star)(\boldsymbol{x}_i - \boldsymbol{x}^\star)^\top = \sum_{k=1}^{2}\sum_{l=1}^{2}\frac{1}{t}\sum_{i=1}^{t}\mathcal{L}_{k,i}\mathcal{L}_{l,i}^\top. \tag{B.73}$$

The following lemma shows that the dominant term $\frac{1}{t}\sum_{i=1}^{t}\mathcal{L}_{1,i}\mathcal{L}_{1,i}^\top$ converges to the limiting covariance matrix $\Sigma^\star$ and establishes the convergence rate.

**Lemma B.8.** *Under the conditions of Theorem 4.6, we have*

$$\mathbb{E}\left[\left\|\frac{1}{t}\sum_{i=1}^{t}\mathcal{L}_{1,i}\mathcal{L}_{1,i}^\top - \Sigma^\star\right\|\right] \lesssim \frac{1}{\sqrt{t\varphi_t}}.$$

With Lemma B.8, the next two lemmas show that $\frac{1}{t}\sum_{i=1}^{t}\frac{1}{\varphi_{i-1}}(\boldsymbol{x}_i - \boldsymbol{x}^\star)(\boldsymbol{x}_i - \boldsymbol{x}^\star)^T$ converges to $\Sigma^\star$, and the remaining terms are negligible as $\bar{\boldsymbol{x}}_t$ converges to $\boldsymbol{x}^\star$ fast.

**Lemma B.9.** *Under the conditions of Theorem 4.6, we have*

$$\mathbb{E}\left[\left\|\frac{1}{t}\sum_{i=1}^{t}\frac{1}{\varphi_{i-1}}(\boldsymbol{x}_i - \boldsymbol{x}^\star)(\boldsymbol{x}_i - \boldsymbol{x}^\star)^\top - \Sigma^\star\right\|\right] \lesssim \frac{1}{\sqrt{t\varphi_t}} \tag{B.74}$$

*and*

$$\frac{1}{t}\sum_{i=1}^{t}\frac{1}{\varphi_{i-1}}\mathbb{E}\left[\|\boldsymbol{x}_i - \boldsymbol{x}^\star\|^2\right] \lesssim 1. \tag{B.75}$$

**Lemma B.10.** *Under the conditions of Theorem 4.6, we have*

$$\mathbb{E}\left[\left\|\frac{1}{t}\sum_{i=1}^{t}\frac{1}{\varphi_{i-1}}(\bar{\boldsymbol{x}}_t - \boldsymbol{x}^\star)(\bar{\boldsymbol{x}}_t - \boldsymbol{x}^\star)^\top\right\|\right] \leq \frac{1}{t}\sum_{i=1}^{t}\frac{1}{\varphi_{i-1}}\mathbb{E}\left[\|\bar{\boldsymbol{x}}_t - \boldsymbol{x}^\star\|^2\right] \lesssim \frac{1}{t\varphi_t}. \tag{B.76}$$

By the decomposition (B.71) of $\widehat{\Sigma}_t$, we apply the Cauchy–Schwarz inequality and obtain

$$\mathbb{E}[\|\widehat{\Sigma}_t - \Sigma^\star\|] \leq \mathbb{E}\left[\left\|\frac{1}{t}\sum_{i=1}^{t}\frac{1}{\varphi_{i-1}}(\boldsymbol{x}_i - \boldsymbol{x}^\star)(\boldsymbol{x}_i - \boldsymbol{x}^\star)^\top - \Sigma^\star\right\|\right] + \mathbb{E}\left[\left\|\frac{1}{t}\sum_{i=1}^{t}\frac{1}{\varphi_{i-1}}(\bar{\boldsymbol{x}}_t - \boldsymbol{x}^\star)(\bar{\boldsymbol{x}}_t - \boldsymbol{x}^\star)^\top\right\|\right]$$

$$+ 2\sqrt{\frac{1}{t}\sum_{i=1}^{t}\frac{1}{\varphi_{i-1}}\mathbb{E}\left[\|\boldsymbol{x}_i - \boldsymbol{x}^\star\|^2\right]}\sqrt{\frac{1}{t}\sum_{i=1}^{t}\frac{1}{\varphi_{i-1}}\mathbb{E}\left[\|\bar{\boldsymbol{x}}_t - \boldsymbol{x}^\star\|^2\right]}.$$

Plugging (B.74), (B.75), (B.76) into the above display, we obtain

$$\mathbb{E}[\|\widehat{\Sigma}_t - \Sigma^\star\|] \lesssim \frac{1}{\sqrt{t\varphi_t}}.$$

This completes the proof.

### B.12. Proof of Lemma B.8

We define

$$\widetilde{C}_k^\star := \prod_{j=0}^{\tau-1}\begin{pmatrix} (1-\alpha^\star)(I - Z_{k,j}^\star) & \alpha^\star(I - Z_{k,j}^\star) \\ (1-\alpha^\star)(1-\beta^\star)I - (1-\alpha^\star)\gamma^\star Z_{k,j}^\star & (\alpha^\star + \beta^\star - \alpha^\star\beta^\star)I - \alpha^\star\gamma^\star Z_{k,j}^\star \end{pmatrix} \in \mathbb{R}^{2d\times 2d}, \tag{B.77}$$

where $Z_{k,j}^\star = B^\star S_{k,j}(S_{k,j}^\top(B^\star)^2 S_{k,j})^\dagger S_{k,j}^\top B^\star \in \mathbb{R}^{d\times d}$ and $S_{k,j}$ is the same sketching matrix in $\widetilde{Z}_{k,j}$ in (3.1). Then, we let

$$\widetilde{K}_k^\star := \begin{pmatrix} I & \boldsymbol{0} \end{pmatrix}\widetilde{C}_k^\star\begin{pmatrix} I \\ I \end{pmatrix} \in \mathbb{R}^{d\times d}. \tag{B.78}$$

Due to the independence of $\{S_{k,j}\}_j$, it is obvious that $\mathbb{E}[\widetilde{C}_k^\star] = C^\star$ and $\mathbb{E}[\widetilde{K}_k^\star] = K^\star$, where $K^\star$ is defined in Section 4.1 and is constructed from $C^\star$ in the same manner as above. Then, we can define

$$\widetilde{\boldsymbol{\theta}}_k := -(I - \widetilde{K}_k^\star)(B^\star)^{-1}\nabla F(\boldsymbol{x}^\star; \xi_k) \quad \text{and} \quad \widehat{\boldsymbol{\theta}}_k := \boldsymbol{\theta}_k - \widetilde{\boldsymbol{\theta}}_k. \tag{B.79}$$

Note that $\widetilde{\boldsymbol{\theta}}_k$ and $\boldsymbol{\theta}_k$ share the same randomness but $\widetilde{\boldsymbol{\theta}}_k$ is constructed at $\boldsymbol{x}^\star$ instead of $\boldsymbol{x}_k$.

In the following, we decompose $\mathcal{L}_{1,i}$ as

$$\mathcal{L}_{1,i} = \frac{1}{\sqrt{\varphi_{i-1}}}\sum_{k=0}^{i-1}\varphi_k\Pi_{i,k+1}\widetilde{\boldsymbol{\theta}}_k + \frac{1}{\sqrt{\varphi_{i-1}}}\sum_{k=0}^{i-1}\varphi_k\Pi_{i,k+1}\widehat{\boldsymbol{\theta}}_k =: \widetilde{\mathcal{L}}_{1,i} + \widehat{\mathcal{L}}_{1,i}. \tag{B.80}$$

Intuitively, as $\boldsymbol{x}_i \to \boldsymbol{x}^\star$, $\widetilde{\mathcal{L}}_{1,i}$ should serve as a good approximation to $\mathcal{L}_{1,i}$ and $\widehat{\mathcal{L}}_{1,i}$ could be negligible. The next two lemmas provide bounds for $\widetilde{\mathcal{L}}_{1,i}$ and $\widehat{\mathcal{L}}_{1,i}$, respectively.

**Lemma B.11.** *Under the conditions of Theorem 4.6, we have*

$$\mathbb{E}\left[\left\|\frac{1}{t}\sum_{i=1}^{t}\widetilde{\mathcal{L}}_{1,i}\widetilde{\mathcal{L}}_{1,i}^\top - \Sigma^\star\right\|\right] \lesssim \frac{1}{\sqrt{t\varphi_t}}.$$

**Lemma B.12.** *Under the conditions of Theorem 4.6, we have*

$$\mathbb{E}\left[\left\|\frac{1}{t}\sum_{i=1}^{t}\widehat{\mathcal{L}}_{1,i}\widehat{\mathcal{L}}_{1,i}^\top\right\|\right] \leq \frac{1}{t}\sum_{i=1}^{t}\mathbb{E}\left[\|\widehat{\mathcal{L}}_{1,i}\|^2\right] \lesssim \varphi_t.$$

By the decomposition (B.80), we have

$$\mathbb{E}\left[\left\|\frac{1}{t}\sum_{i=1}^{t}\mathcal{L}_{1,i}\mathcal{L}_{1,i}^\top - \Sigma^\star\right\|\right] \leq \mathbb{E}\left[\left\|\frac{1}{t}\sum_{i=1}^{t}\widetilde{\mathcal{L}}_{1,i}\widetilde{\mathcal{L}}_{1,i}^\top - \Sigma^\star\right\|\right] + \mathbb{E}\left[\left\|\frac{1}{t}\sum_{i=1}^{t}\widehat{\mathcal{L}}_{1,i}\widehat{\mathcal{L}}_{1,i}^\top\right\|\right] + 2\mathbb{E}\left[\left\|\frac{1}{t}\sum_{i=1}^{t}\widetilde{\mathcal{L}}_{1,i}\widehat{\mathcal{L}}_{1,i}^\top\right\|\right]. \tag{B.81}$$

Applying the Cauchy-Schwarz inequality to the last term, we obtain

$$\mathbb{E}\left[\left\|\frac{1}{t}\sum_{i=1}^{t}\widetilde{\mathcal{L}}_{1,i}\widehat{\mathcal{L}}_{1,i}^\top\right\|\right] \leq \mathbb{E}\left[\frac{1}{t}\sum_{i=1}^{t}\|\widetilde{\mathcal{L}}_{1,i}\|\|\widehat{\mathcal{L}}_{1,i}\|\right] \leq \mathbb{E}\left[\sqrt{\frac{1}{t}\sum_{i=1}^{t}\|\widetilde{\mathcal{L}}_{1,i}\|^2}\sqrt{\frac{1}{t}\sum_{i=1}^{t}\|\widehat{\mathcal{L}}_{1,i}\|^2}\right]$$

$$\leq \sqrt{\frac{1}{t}\sum_{i=1}^{t}\mathbb{E}\left[\|\widetilde{\mathcal{L}}_{1,i}\|^2\right]}\sqrt{\frac{1}{t}\sum_{i=1}^{t}\mathbb{E}\left[\|\widehat{\mathcal{L}}_{1,i}\|^2\right]}. \tag{B.82}$$

Given Lemmas B.11 and B.12, we only need to bound $\frac{1}{t}\sum_{i=1}^{t}\mathbb{E}[\|\widetilde{\mathcal{L}}_{1,i}\|^2]$. We have

$$\mathbb{E}[\|\widetilde{\mathcal{L}}_{1,i}\|^2] \overset{(\text{B.80})}{=} \frac{1}{\varphi_{i-1}}\sum_{k_1,k_2=0}^{i-1}\varphi_{k_1}\varphi_{k_2}\mathbb{E}\left[\widetilde{\boldsymbol{\theta}}_{k_1}^\top\Pi_{i,k_1+1}^\top\Pi_{i,k_2+1}\widetilde{\boldsymbol{\theta}}_{k_2}\right]$$

$$= \frac{1}{\varphi_{i-1}}\sum_{k=0}^{i-1}\varphi_k^2\mathbb{E}\left[\left\|\Pi_{i,k+1}\widetilde{\boldsymbol{\theta}}_k\right\|^2\right] \leq \frac{1}{\varphi_{i-1}}\sum_{k=0}^{i-1}\varphi_k^2\|\Pi_{i,k+1}\|^2\mathbb{E}[\|\widetilde{\boldsymbol{\theta}}_k\|^2]$$

$$\leq \frac{1}{\varphi_{i-1}}\sum_{k=0}^{i-1}\varphi_k^2\prod_{l=k+1}^{i-1}(1 - \varphi_l\lambda_m)^2\sqrt{\mathbb{E}[\|\widetilde{\boldsymbol{\theta}}_k\|^4]}, \tag{B.83}$$

where the second equality uses the fact that $\{\widetilde{\boldsymbol{\theta}}_k\}$ are mean zero and independent, and the last inequality is the direct application of Jensen's inequality. Next, we will bound the fourth-order moment of $\|\widetilde{\boldsymbol{\theta}}_k\|$. We have

$$\mathbb{E}\left[\|\nabla F(\boldsymbol{x}^\star; \xi_k)\|^4\right] \lesssim \mathbb{E}\left[\|\nabla F(\boldsymbol{x}^\star; \xi_k) - \nabla F(\boldsymbol{x}_k; \xi_k)\|^4\right] + \mathbb{E}\left[\|\nabla F(\boldsymbol{x}_k; \xi_k) - \nabla F_k\|^4\right] + \mathbb{E}\left[\|\nabla F_k - \nabla F^\star\|^4\right]$$

$$\lesssim \Upsilon_H^4\mathbb{E}\left[\|\boldsymbol{x}_k - \boldsymbol{x}^\star\|^4\right] + \mathbb{E}\left[\|\boldsymbol{x}_k - \boldsymbol{x}^\star\|^4\right] + 1 + \mathbb{E}\left[\|\boldsymbol{x}_k - \boldsymbol{x}^\star\|^4\right] \quad \text{(by Assumptions 3.1 and 3.2 with } q_g = 4)$$

$$\lesssim 1 + \varphi_k^2 \quad \text{(by Lemma 4.5)}$$

$$\lesssim 1. \tag{B.84}$$

Then, by (3.2) and (A.28), we have

$$\mathbb{E}[\|\widetilde{\boldsymbol{\theta}}_k\|^4] \overset{(B.79)}{\leq} \mathbb{E}\left[\|I - \widetilde{K}_k^\star\|^4 \|(B^\star)^{-1}\|^4 \|\nabla F(\boldsymbol{x}^\star; \xi_k)\|^4\right] \leq \frac{(1+C_K)^4}{\gamma_H^4} \mathbb{E}\left[\|\nabla F(\boldsymbol{x}^\star; \xi_k)\|^4\right] \overset{(B.84)}{\lesssim} 1. \qquad (B.85)$$

Plugging (B.85) into (B.83) gives us

$$\frac{1}{t}\sum_{i=1}^{t}\mathbb{E}[\|\widetilde{\mathcal{L}}_{1,i}\|^2] \lesssim \frac{1}{t}\sum_{i=0}^{t-1}\frac{1}{\varphi_i}\sum_{k=0}^{i}\varphi_k^2\prod_{l=k+1}^{i}(1-\varphi_l\lambda_m)^2 \lesssim \frac{1}{t}\sum_{i=0}^{t-1}\frac{1}{\varphi_i}\cdot\varphi_i = 1, \qquad (B.86)$$

where the second inequality is the direct application of Na & Mahoney (2025, Lemma B.3). Combining (B.82), (B.86) and Lemma B.12, we obtain

$$\mathbb{E}\left[\left\|\frac{1}{t}\sum_{i=1}^{t}\frac{1}{\varphi_i}\widetilde{\mathcal{L}}_{1,i}\widehat{\mathcal{L}}_{1,i}^\top\right\|\right] \lesssim \sqrt{\varphi_t}.$$

Combining the above display with Lemmas B.11, B.12, and (B.81), we obtain

$$\mathbb{E}\left[\left\|\frac{1}{t}\sum_{i=1}^{t}\mathcal{L}_{1,i}\mathcal{L}_{1,i}^\top - \Sigma^\star\right\|\right] \lesssim \frac{1}{\sqrt{t\varphi_t}}.$$

This completes the proof.

## B.13. Proof of Lemma B.11

Consider the eigenvalue decomposition of $\mathcal{R} = I - K^\star = U\Lambda U^\top$ with $\Lambda = \mathrm{diag}(\lambda_1, \ldots, \lambda_d)$. We have

$$\widetilde{\mathcal{L}}_{1,i} = \frac{1}{\sqrt{\varphi_{i-1}}}\sum_{k=0}^{i-1}\varphi_k\prod_{l=k+1}^{i-1}(I - \varphi_l\mathcal{R})\widetilde{\boldsymbol{\theta}}_k = \frac{1}{\sqrt{\varphi_{i-1}}}U\sum_{k=0}^{i-1}\varphi_k\prod_{l=k+1}^{i-1}(I - \varphi_l\Lambda)U^\top\widetilde{\boldsymbol{\theta}}_k. \qquad (B.87)$$

Let $\widetilde{\mathcal{Q}}_i = U^\top\widetilde{\mathcal{L}}_{1,i}$ and $\Xi^\star = U^\top\Gamma^\star U$ with $\Gamma^\star = \mathbb{E}[(I - \widetilde{K}^\star)\Omega^\star(I - \widetilde{K}^\star)^\top]$ defined in (4.1). It is known by Na & Mahoney (2025, (5.10)) that the limiting covariance $\Sigma^\star$ in (4.2) can be explicitly expressed as

$$\Sigma^\star = U\left(\Theta \circ \Xi^\star\right)U^\top \quad \text{with} \quad \Theta_{k,l} = \frac{1}{\lambda_k + \lambda_l - \mathbf{1}_{\{\varphi=1\}}/C_\varphi},$$

where $\circ$ denotes the matrix Hadamard product. Therefore, we obtain

$$\mathbb{E}\left[\left\|\frac{1}{t}\sum_{i=1}^{t}\widetilde{\mathcal{L}}_{1,i}\widetilde{\mathcal{L}}_{1,i}^\top - \Sigma^\star\right\|\right] = \mathbb{E}\left[\left\|\frac{1}{t}\sum_{i=1}^{t}\widetilde{\mathcal{Q}}_i\widetilde{\mathcal{Q}}_i^\top - \Theta\circ\Xi^\star\right\|\right]$$

$$\leq \mathbb{E}\left[\left\|\frac{1}{t}\sum_{i=1}^{t}\widetilde{\mathcal{Q}}_i\widetilde{\mathcal{Q}}_i^\top - \Theta\circ\Xi^\star\right\|_F\right] \leq \sqrt{\mathbb{E}\left[\left\|\frac{1}{t}\sum_{i=1}^{t}\widetilde{\mathcal{Q}}_i\widetilde{\mathcal{Q}}_i^\top - \Theta\circ\Xi^\star\right\|_F^2\right]}.$$

For the term above, we perform bias-variance decomposition as follows:

$$\mathbb{E}\left[\left\|\frac{1}{t}\sum_{i=1}^{t}\widetilde{\mathcal{Q}}_i\widetilde{\mathcal{Q}}_i^\top - \Theta\circ\Xi^\star\right\|_F^2\right] = \sum_{p,q=1}^{d}\mathbb{E}\left[\left(\frac{1}{t}\sum_{i=1}^{t}\widetilde{\mathcal{Q}}_{i,p}\widetilde{\mathcal{Q}}_{i,q} - \Theta_{p,q}\Xi_{p,q}^\star\right)^2\right] = I + II, \qquad (B.88)$$

where

$$I := \sum_{p,q=1}^{d}\left\{\mathbb{E}\left[\left(\frac{1}{t}\sum_{i=1}^{t}\widetilde{\mathcal{Q}}_{i,p}\widetilde{\mathcal{Q}}_{i,q}\right)^2\right] - \left(\mathbb{E}\left[\frac{1}{t}\sum_{i=1}^{t}\widetilde{\mathcal{Q}}_{i,p}\widetilde{\mathcal{Q}}_{i,q}\right]\right)^2\right\} \quad \text{(variance)}, \qquad (B.89a)$$

$$II := \sum_{p,q=1}^{d}\left(\mathbb{E}\left[\frac{1}{t}\sum_{i=1}^{t}\widetilde{\mathcal{Q}}_{i,p}\widetilde{\mathcal{Q}}_{i,q}\right] - \Theta_{p,q}\Xi_{p,q}^\star\right)^2 \quad \text{(bias}^2), \qquad (B.89b)$$

where $\widetilde{\mathcal{Q}}_{i,p}$ represents the $p$-th element in $\widetilde{\mathcal{Q}}_i$.

We start with the analysis for the Term $II$. By (B.87), we have

$$
\mathbb{E}\left[\frac{1}{t}\sum_{i=1}^{t}\widetilde{\mathcal{Q}}_{i,p}\widetilde{\mathcal{Q}}_{i,q}\right] = \frac{1}{t}\sum_{i=1}^{t}\frac{1}{\varphi_{i-1}}\sum_{k_1=0}^{i-1}\sum_{k_2=0}^{i-1}\prod_{l_1=k_1+1}^{i-1}(1-\varphi_{l_1}\lambda_p)\prod_{l_2=k_2+1}^{i-1}(1-\varphi_{l_2}\lambda_q)\varphi_{k_1}\varphi_{k_2}\mathbb{E}\left[\left(U^\top\widetilde{\boldsymbol{\theta}}_{k_1}\widetilde{\boldsymbol{\theta}}_{k_2}^\top U\right)_{p,q}\right]
$$

$$
= \frac{1}{t}\sum_{i=1}^{t}\frac{1}{\varphi_{i-1}}\sum_{k=0}^{i-1}\prod_{l=k+1}^{i-1}(1-\varphi_l\lambda_p)(1-\varphi_l\lambda_q)\varphi_k^2\Xi^\star_{p,q}, \tag{B.90}
$$

where the second equality is due to the definition of $\widetilde{\boldsymbol{\theta}}_k$ in (B.79) and the independence between the random sample $\xi_k$ and the sketching matrices $\{S_{k,j}\}_j$. In particular, we have

$$
\mathbb{E}[U^\top\widetilde{\boldsymbol{\theta}}_{k_1}\widetilde{\boldsymbol{\theta}}_{k_2}^\top U] = 0 \ \text{ for } \ k_1 \neq k_2 \qquad \text{and} \qquad \mathbb{E}[U^\top\widetilde{\boldsymbol{\theta}}_{k}\widetilde{\boldsymbol{\theta}}_{k}^\top U] = \Xi^\star \ \text{ for } \ k_1 = k_2 = k.
$$

We give the following technical lemma which will be used for upper bounding the Term $II$.

**Lemma B.13** (Kuang et al. (2025), Lemma B.3). *Suppose $\{\phi_i\}_i$ and $\{\sigma_i\}_i$ are two positive sequences, and $\{\phi_i\}_i$ satisfies $\lim_{i\to\infty} i(1-\phi_{i-1}/\phi_i) = \phi < 0$ for a constant $\phi$. Let $\varphi_i = c_\varphi/(i+1)^\varphi + o(1/(i+1)^\varphi)$ for constants $c_\varphi > 0$ and $\varphi \in (0,1)$. For any $l \geq 1$, we have*

$$
\left|\frac{1}{\phi_t}\sum_{i=0}^{t}\prod_{j=i+1}^{t}\prod_{k=1}^{l}(1-\varphi_j\sigma_k)\varphi_i\phi_i - \frac{1}{\sum_{k=1}^{l}\sigma_k}\right| \lesssim \begin{cases} \varphi_t, & \varphi \in (0,0.5), \\ \left(0.5 - \dfrac{\phi/c_\varphi^2}{(\sum_{k=1}^{l}\sigma_k)^2}\right)\varphi_t, & \varphi = 0.5, \\ -\dfrac{\phi}{(\sum_{k=1}^{l}\sigma_k)^2}\cdot\dfrac{1}{t\varphi_t}, & \varphi \in (0.5,1). \end{cases}
$$

With this lemma, we plug (B.90) into the definition of term $II$ in (B.89b), and have

$$
II \leq \sum_{p,q=1}^{d}\left(\frac{1}{t}\sum_{i=1}^{t}\left|\frac{1}{\varphi_{i-1}}\sum_{k=0}^{i-1}\prod_{l=k+1}^{i-1}(1-\varphi_l\lambda_p)(1-\varphi_l\lambda_q)\varphi_k^2 - \Theta_{p,q}\right|\right)^2(\Xi^\star_{p,q})^2
$$

$$
= \sum_{p,q=1}^{d}\left(\frac{1}{t}\sum_{i=0}^{t-1}\left|\frac{1}{\varphi_i}\sum_{k=0}^{i}\prod_{l=k+1}^{i}(1-\varphi_l\lambda_p)(1-\varphi_l\lambda_q)\varphi_k^2 - \Theta_{p,q}\right|\right)^2(\Xi^\star_{p,q})^2
$$

$$
\lesssim \sum_{p,q=1}^{d}\left(\frac{\varphi}{(\lambda_p+\lambda_q)^2}\cdot\frac{1}{t}\sum_{i=0}^{t-1}\frac{1}{i\varphi_i}\right)^2(\Xi^\star_{p,q})^2 \quad \text{(by Lemma B.13)}
$$

$$
\lesssim \left(\frac{1}{t}\sum_{i=0}^{t-1}\frac{1}{i\varphi_i}\right)^2 \lesssim \frac{1}{t^2\varphi_t^2}, \tag{B.91}
$$

where the third inequality also uses the fact that $\lim_{i\to\infty} i(1-\varphi_{i-1}/\varphi_i) = -\varphi$, which can be easily verified; and the last inequality is due to the fact that $\sum_{i=0}^{t-1} 1/(i\varphi_i) \lesssim \int_0^t 1/i^{1-\varphi}di \lesssim 1/\varphi_t$.

Now, we deal with the Term $I$ in (B.89a). By (B.87), we expand $I$ as

$$
I = \sum_{p,q=1}^{d}\frac{1}{t^2}\sum_{i_1,i_2=1}^{t}\frac{1}{\varphi_{i_1-1}}\frac{1}{\varphi_{i_2-1}}\sum_{k_1,k_1'=0}^{i_1-1}\sum_{k_2,k_2'=0}^{i_2-1}\prod_{l_1=k_1+1}^{i_1-1}(1-\varphi_{l_1}\lambda_p)\prod_{l_1'=k_1'+1}^{i_1-1}(1-\varphi_{l_1'}\lambda_q)\prod_{l_2=k_2+1}^{i_2-1}(1-\varphi_{l_2}\lambda_p)\prod_{l_2'=k_2'+1}^{i_2-1}(1-\varphi_{l_2'}\lambda_q)
$$

$$
\varphi_{k_1}\varphi_{k_1'}\varphi_{k_2}\varphi_{k_2'}\left\{\mathbb{E}\left[\left(U^\top\widetilde{\boldsymbol{\theta}}_{k_1}\widetilde{\boldsymbol{\theta}}_{k_1'}^\top U\right)_{p,q}\left(U^\top\widetilde{\boldsymbol{\theta}}_{k_2}\widetilde{\boldsymbol{\theta}}_{k_2'}^\top U\right)_{p,q}\right] - \mathbb{E}\left[\left(U^\top\widetilde{\boldsymbol{\theta}}_{k_1}\widetilde{\boldsymbol{\theta}}_{k_1'}^\top U\right)_{p,q}\right]\mathbb{E}\left[\left(U^\top\widetilde{\boldsymbol{\theta}}_{k_2}\widetilde{\boldsymbol{\theta}}_{k_2'}^\top U\right)_{p,q}\right]\right\}
$$

$$
= \sum_{p,q=1}^{d}\frac{1}{t^2}\sum_{i_1,i_2=0}^{t-1}\frac{1}{\varphi_{i_1}}\frac{1}{\varphi_{i_2}}\sum_{k_1,k_1'=0}^{i_1}\sum_{k_2,k_2'=0}^{i_2}\prod_{l_1=k_1+1}^{i_1}(1-\varphi_{l_1}\lambda_p)\prod_{l_1'=k_1'+1}^{i_1}(1-\varphi_{l_1'}\lambda_q)\prod_{l_2=k_2+1}^{i_2}(1-\varphi_{l_2}\lambda_p)\prod_{l_2'=k_2'+1}^{i_2}(1-\varphi_{l_2'}\lambda_q)
$$

$$\varphi_{k_1}\varphi_{k_1'}\varphi_{k_2}\varphi_{k_2'}\left\{\mathbb{E}\left[\left(U^\top\widetilde{\boldsymbol{\theta}}_{k_1}\widetilde{\boldsymbol{\theta}}_{k_1'}^\top U\right)_{p,q}\left(U^\top\widetilde{\boldsymbol{\theta}}_{k_2}\widetilde{\boldsymbol{\theta}}_{k_2'}^\top U\right)_{p,q}\right]-\mathbb{E}\left[\left(U^\top\widetilde{\boldsymbol{\theta}}_{k_1}\widetilde{\boldsymbol{\theta}}_{k_1'}^\top U\right)_{p,q}\right]\mathbb{E}\left[\left(U^\top\widetilde{\boldsymbol{\theta}}_{k_2}\widetilde{\boldsymbol{\theta}}_{k_2'}^\top U\right)_{p,q}\right]\right\}.$$

Since $\{\widetilde{\boldsymbol{\theta}}_k\}_k$ are independent across $k$ and $\mathbb{E}[\widetilde{\boldsymbol{\theta}}_k]=\mathbf{0}$, any configuration for which $\{k_1,k_1',k_2,k_2'\}$ contains a singleton index (appearing only once) yields zero, and disjoint pairs factorize to zero. Therefore, the term in braces can be nonzero only when there are no singletons and the two pairs are not disjoint. In particular, we will consider the following three cases: (1) $k_1=k_1'=k_2=k_2'$; (2) $k_1=k_2$, $k_1'=k_2'$, $k_1\neq k_1'$; (3) $k_1=k_2'$, $k_1'=k_2$, $k_1\neq k_1'$.

- **Case 1:** $k_1=k_1'=k_2=k_2'=k$. In this case, since $k_1=k_1'$, we have

$$\left(\mathbb{E}\left[\left(U^\top\widetilde{\boldsymbol{\theta}}_k\widetilde{\boldsymbol{\theta}}_k^\top U\right)_{p,q}\right]\right)^2\geq 0\qquad\text{and}\qquad\mathbb{E}\left[\sum_{p,q=1}^d\left(U^\top\widetilde{\boldsymbol{\theta}}_k\widetilde{\boldsymbol{\theta}}_k^\top U\right)_{p,q}^2\right]=\mathbb{E}[\|\widetilde{\boldsymbol{\theta}}_k\|^4]\overset{(\text{B.85})}{\lesssim}1.$$

Summing over the indices in **Case 1** and applying the above display, we obtain

$$\begin{aligned}I_1&\leq\sum_{p,q=1}^d\frac{1}{t^2}\sum_{i_1=0}^{t-1}\sum_{i_2=0}^{t-1}\frac{1}{\varphi_{i_1}}\frac{1}{\varphi_{i_2}}\sum_{k=0}^{i_1\wedge i_2}\prod_{l_1=k+1}^{i_1}(1-\varphi_{l_1}\lambda_p)(1-\varphi_{l_1}\lambda_q)\prod_{l_2=k+1}^{i_2}(1-\varphi_{l_2}\lambda_p)(1-\varphi_{l_2}\lambda_q)\varphi_k^4\mathbb{E}\left[\left(U^\top\widetilde{\boldsymbol{\theta}}_k\widetilde{\boldsymbol{\theta}}_k^\top U\right)_{p,q}^2\right]\\&\lesssim\frac{1}{t^2}\sum_{i_1=0}^{t-1}\sum_{i_2=0}^{t-1}\frac{1}{\varphi_{i_1}\varphi_{i_2}}\sum_{k=0}^{i_1\wedge i_2}\prod_{l_1=k+1}^{i_1}(1-\varphi_{l_1}\lambda_m)^2\prod_{l_2=k+1}^{i_2}(1-\varphi_{l_2}\lambda_m)^2\varphi_k^4\\&\lesssim\frac{1}{t^2}\sum_{i_1=0}^{t-1}\frac{1}{\varphi_{i_1}}\sum_{i_2=0}^{i_1}\frac{1}{\varphi_{i_2}}\prod_{l=i_2+1}^{i_1}(1-\varphi_l\lambda_m)^2\sum_{k=0}^{i_2}\prod_{l=k+1}^{i_2}(1-\varphi_l\lambda_m)^4\varphi_k^4\\&\lesssim\frac{1}{t^2}\sum_{i_1=0}^{t-1}\frac{1}{\varphi_{i_1}}\sum_{i_2=0}^{i_1}\frac{1}{\varphi_{i_2}}\prod_{l=i_2+1}^{i_1}(1-\varphi_l\lambda_m)^2\varphi_{i_2}^3\lesssim\frac{1}{t^2}\sum_{i_1=0}^{t-1}\frac{1}{\varphi_{i_1}}\varphi_{i_1}=\frac{1}{t},\end{aligned}\tag{B.92}$$

where the two inequalities in the last line are both direct applications of Na & Mahoney (2025, Lemma B.3).

- **Case 2:** $k_1=k_2$, $k_1'=k_2'$, $k_1\neq k_1'$. In this case, since $k_1\neq k_1'$, we have

$$\mathbb{E}\left[\left(U^\top\widetilde{\boldsymbol{\theta}}_{k_1}\widetilde{\boldsymbol{\theta}}_{k_1'}^\top U\right)_{p,q}\right]\mathbb{E}\left[\left(U^\top\widetilde{\boldsymbol{\theta}}_{k_2}\widetilde{\boldsymbol{\theta}}_{k_2'}^\top U\right)_{p,q}\right]=0\ \text{ and }\ \mathbb{E}\left[\sum_{p,q=1}^d\left(U^\top\widetilde{\boldsymbol{\theta}}_{k_1}\right)_p^2\left(U^\top\widetilde{\boldsymbol{\theta}}_{k_1'}\right)_q^2\right]=\mathbb{E}[\|\widetilde{\boldsymbol{\theta}}_{k_1}\|^2]\mathbb{E}[\|\widetilde{\boldsymbol{\theta}}_{k_1'}\|^2]\overset{(\text{B.85})}{\lesssim}1.$$

Summing over the indices in **Case 2** and applying the above display yield

$$\begin{aligned}I_2&=\sum_{p,q=1}^d\frac{1}{t^2}\sum_{i_1=0}^{t-1}\sum_{i_2=0}^{t-1}\frac{1}{\varphi_{i_1}}\frac{1}{\varphi_{i_2}}\sum_{k_1=0}^{i_1\wedge i_2}\prod_{l_1=k_1+1}^{i_1}(1-\varphi_{l_1}\lambda_p)\prod_{l_2=k_1+1}^{i_2}(1-\varphi_{l_2}\lambda_p)\varphi_{k_1}^2\\&\quad\times\sum_{k_1'=0,k_1'\neq k_1}^{i_1\wedge i_2}\prod_{l_1'=k_1'+1}^{i_1}(1-\varphi_{l_1'}\lambda_q)\prod_{l_2'=k_1'+1}^{i_2}(1-\varphi_{l_2'}\lambda_q)\varphi_{k_1'}^2\mathbb{E}\left[\left(U^\top\widetilde{\boldsymbol{\theta}}_{k_1}\right)_p^2\left(U^\top\widetilde{\boldsymbol{\theta}}_{k_1'}\right)_q^2\right]\\&\lesssim\frac{1}{t^2}\sum_{i_1,i_2=0}^{t-1}\frac{1}{\varphi_{i_1}\varphi_{i_2}}\sum_{k_1=0}^{i_1\wedge i_2}\prod_{l_1=k_1+1}^{i_1}(1-\varphi_{l_1}\lambda_m)\prod_{l_2=k_1+1}^{i_2}(1-\varphi_{l_2}\lambda_m)\varphi_{k_1}^2\sum_{k_1'=0,k_1'\neq k_1}^{i_1\wedge i_2}\prod_{l_1'=k_1'+1}^{i_1}(1-\varphi_{l_1'}\lambda_m)\prod_{l_2'=k_1'+1}^{i_2}(1-\varphi_{l_2'}\lambda_m)\varphi_{k_1'}^2\\&\lesssim\frac{1}{t^2}\sum_{i_1=0}^{t-1}\frac{1}{\varphi_{i_1}}\sum_{i_2=0}^{i_1}\frac{1}{\varphi_{i_2}}\prod_{l=i_2+1}^{i_1}(1-\varphi_l\lambda_m)^2\left\{\sum_{k=0}^{i_2}\prod_{l=k+1}^{i_2}(1-\varphi_l\lambda_m)^2\varphi_k^2\right\}^2\lesssim\frac{1}{t^2}\sum_{i_1=0}^{t-1}\frac{1}{\varphi_{i_1}}\sum_{i_2=0}^{i_1}\frac{1}{\varphi_{i_2}}\prod_{l=i_2+1}^{i_1}(1-\varphi_l\lambda_m)^2\varphi_{i_2}^2\\&\lesssim\frac{1}{t^2}\sum_{i_1=0}^{t-1}\frac{1}{\varphi_{i_1}}\lesssim\frac{1}{t^2}\cdot t^{1+\varphi}\lesssim\frac{1}{t\varphi_t},\end{aligned}\tag{B.93}$$

where the fourth and fifth inequalities are both due to Na & Mahoney (2025, Lemma B.3).

• **Case 3:** $k_1 = k_2'$, $k_1' = k_2$, $k_1 \neq k_1'$. In this case, we have

$$\mathbb{E}\left[\left(U^\top \widetilde{\boldsymbol{\theta}}_{k_1} \widetilde{\boldsymbol{\theta}}_{k_1'}^\top U\right)_{p,q}\right] \mathbb{E}\left[\left(U^\top \widetilde{\boldsymbol{\theta}}_{k_2} \widetilde{\boldsymbol{\theta}}_{k_2'}^\top U\right)_{p,q}\right] = 0 \quad \text{and} \quad \mathbb{E}\left[\sum_{p,q=1}^d \left(U^\top \widetilde{\boldsymbol{\theta}}_{k_1}\right)_p^2 \left(U^\top \widetilde{\boldsymbol{\theta}}_{k_1'}\right)_q^2\right] = \|\Xi^\star\|_F^2 = \|\Gamma^\star\|_F^2 \lesssim 1.$$

Therefore, the analysis of $I_3$ is identical to that of $I_2$ and we can conclude that

$$I_3 \lesssim \frac{1}{t\varphi_t}. \tag{B.94}$$

Combining (B.92), (B.93) and (B.94), we obtain

$$I \lesssim \frac{1}{t} + \frac{1}{t\varphi_t} + \frac{1}{t\varphi_t} \lesssim \frac{1}{t\varphi_t}. \tag{B.95}$$

Plugging (B.91) and (B.95) into (B.88), we complete the proof.

## B.14. Proof of Lemma B.12

We present the following lemma to bound $\widehat{\boldsymbol{\theta}}^k$ defined in (B.79).

**Lemma B.14.** *Under the conditions of Theorem 4.6, we have*

$$\mathbb{E}[\|\widehat{\boldsymbol{\theta}}_k\|^2] \lesssim \mathbb{E}[\|\boldsymbol{x}_k - \boldsymbol{x}^\star\|^2] + \mathbb{E}[\|B_k - B^\star\|^2].$$

This lemma indicates that the difference between $\boldsymbol{\theta}_k$ and its approximation $\widetilde{\boldsymbol{\theta}}_k$ vanishes. Combining Lemma B.14 with Lemma 4.5, we get $\mathbb{E}[\|\widehat{\boldsymbol{\theta}}_k\|^2] \lesssim \varphi_k$. Furthermore, we recall the expression of $\widehat{\mathcal{L}}_{1,i}$ in (B.80). Since $(\widehat{\boldsymbol{\theta}}_k)$ is a martingale difference sequence, we follow the analysis in (B.83) and (B.86) and obtain

$$\mathbb{E}\left[\left\|\frac{1}{t}\sum_{i=1}^t \widehat{\mathcal{L}}_{1,i}\widehat{\mathcal{L}}_{1,i}^\top\right\|\right] \leq \frac{1}{t}\sum_{i=1}^t \mathbb{E}[\|\widehat{\mathcal{L}}_{1,i}\|^2] \leq \frac{1}{t}\sum_{i=0}^{t-1}\frac{1}{\varphi_i}\sum_{k=0}^i \varphi_k^2 \prod_{l=k+1}^i (1-\varphi_l\lambda_m)^2 \mathbb{E}[\|\widehat{\boldsymbol{\theta}}_k\|^2]$$

$$\lesssim \frac{1}{t}\sum_{i=0}^{t-1}\frac{1}{\varphi_i}\sum_{k=0}^i \varphi_k^3 \prod_{l=k+1}^i (1-\varphi_l\lambda_m)^2 \lesssim \frac{1}{t}\sum_{i=0}^{t-1}\frac{1}{\varphi_i}\varphi_i^2 \lesssim \frac{1}{t}t^{1-\varphi} \lesssim \varphi_t. \tag{B.96}$$

This completes the proof.

## B.15. Proof of Lemma B.14

We expand $\widehat{\boldsymbol{\theta}}_k$ based on its definition in (B.79) as

$$\widehat{\boldsymbol{\theta}}_k = \boldsymbol{\theta}_k - \widetilde{\boldsymbol{\theta}}_k = (I-K_k)B_k^{-1}\nabla f_k - (I-K_k)B_k^{-1}\nabla F(\boldsymbol{x}_k;\xi_k) + (I-\widetilde{K}_k^\star)(B^\star)^{-1}\nabla F(\boldsymbol{x}^\star;\xi_k) + [\boldsymbol{z}_{k,\tau} - (I-K_k)\Delta\boldsymbol{x}_k]$$

$$= (I-K_k)B_k^{-1}\nabla f_k - (I-\widetilde{K}_k)B_k^{-1}\left[\nabla F(\boldsymbol{x}_k;\xi_k) - \nabla F(\boldsymbol{x}^\star;\xi_k)\right] - \left[(I-\widetilde{K}_k)B_k^{-1} - (I-\widetilde{K}_k^\star)(B^\star)^{-1}\right]\nabla F(\boldsymbol{x}^\star;\xi_k).$$

Then, we can bound $\|\widehat{\boldsymbol{\theta}}^k\|^2$ as

$$\|\widehat{\boldsymbol{\theta}}_k\|^2 \lesssim \|I-K_k\|^2\|B_k^{-1}\|^2\|\nabla f_k\|^2 + \|I-\widetilde{K}_k\|^2\|B_k^{-1}\|^2\|\nabla F(\boldsymbol{x}_k;\xi_k) - \nabla F(\boldsymbol{x}^\star;\xi_k)\|^2$$

$$+ \left\|(I-\widetilde{K}_k)B_k^{-1} - (I-\widetilde{K}_k^\star)(B^\star)^{-1}\right\|^2 \|\nabla F(\boldsymbol{x}^\star;\xi_k)\|^2 =: I' + II' + III'.$$

For the first two terms, by Assumption 3.1, Lemma 3.7, (A.23), (A.26), (A.28) and the condition on $\tau$, we obtain

$$\mathbb{E}\left[I'\right] \lesssim \mathbb{E}\left[\|\boldsymbol{x}_k - \boldsymbol{x}^\star\|^2\right] \quad \text{and} \quad \mathbb{E}\left[II'\right] \lesssim \mathbb{E}\left[\|\boldsymbol{x}_k - \boldsymbol{x}^\star\|^2\right]. \tag{B.97}$$

Regarding the term $III'$, by (A.26) and (A.28), we have

$$\left\|(I-\widetilde{K}_k)B_k^{-1} - (I-\widetilde{K}_k^\star)(B^\star)^{-1}\right\|^2 \leq \left(\|I-\widetilde{K}_k\|\left\|B_k^{-1} - (B^\star)^{-1}\right\| + \|(B^\star)^{-1}\|\left\|\widetilde{K}_k - \widetilde{K}_k^\star\right\|\right)^2$$

$$\lesssim \|I-\widetilde{K}_k\|^2\left\|B_k^{-1} - (B^\star)^{-1}\right\|^2 + \|(B^\star)^{-1}\|^2\left\|\widetilde{K}_k - \widetilde{K}_k^\star\right\|^2 \quad \text{(by Young's inequality)}$$

$$\leq \|I-\widetilde{K}_k\|^2\left\|B_k^{-1}\right\|^2\|(B^\star)^{-1}\|^2\|B_k - B^\star\|^2 + \|(B^\star)^{-1}\|^2\left\|\widetilde{K}_k - \widetilde{K}_k^\star\right\|^2 \lesssim \|B_k - B^\star\|^2 + \|\widetilde{K}_k - \widetilde{K}_k^\star\|^2.$$

Furthermore, applying the tower property gives us

$$
\mathbb{E}\left[III'\right] \lesssim \mathbb{E}\left[\mathbb{E}\left[\left(\|B_k - B^\star\|^2 + \|\widetilde{K}_k - \widetilde{K}_k^\star\|^2\right)\|\nabla F(\boldsymbol{x}^\star; \xi_k)\|^2 \Big| \mathcal{F}_{k-1}\right]\right]
$$

$$
= \mathbb{E}\left[\|B_k - B^\star\|^2 \mathbb{E}\left[\|\nabla F(\boldsymbol{x}^\star; \xi_k)\|^2 \big| \mathcal{F}_{k-1}\right]\right] + \mathbb{E}\left[\mathbb{E}\left[\|\widetilde{K}_k - \widetilde{K}_k^\star\|^2 \big| \mathcal{F}_{k-1}\right]\mathbb{E}\left[\|\nabla F(\boldsymbol{x}^\star; \xi_k)\|^2 \big| \mathcal{F}_{k-1}\right]\right]
$$

$$
\overset{(\text{B.84})}{\lesssim} \mathbb{E}\left[\|B_k - B^\star\|^2\right] + \mathbb{E}[\|\widetilde{K}_k - \widetilde{K}_k^\star\|^2]. \tag{B.98}
$$

Here, the second equality is due to $\|B_k - B^\star\|$ being $\mathcal{F}_{k-1}$-measurable as well as the independence between $\xi_k$ and $\{S_{k,j}\}_{j=0}^\tau$. Similar to the proof of Lemma 4.1, by the definitions of $\widetilde{C}_k$ in Section 3.2 and $\widetilde{C}_k^\star$ in (B.77), we obtain

$$
\|\widetilde{C}_k - \widetilde{C}_k^\star\| \lesssim \sum_{j=0}^{\tau-1}(2 + 1/\sqrt{\gamma_S})^j \cdot (|\alpha_k - \alpha^\star| + |\beta_k - \beta^\star| + |\gamma_k - \gamma^\star| + \|Z_{k,j} - \widetilde{Z}_{k,j}^\star\|) \cdot (2 + 1/\sqrt{\gamma_S})^{\tau-j-1}
$$

$$
\lesssim \sum_{j=0}^{\tau-1}(2 + 1/\sqrt{\gamma_S})^{\tau-1} \cdot \left(\mathbb{E}[\|\widetilde{Z}_k - \widetilde{Z}^\star\| \mid \mathcal{F}_{k-1}] + \|Z_{k,j} - \widetilde{Z}_{k,j}^\star\|\right)
$$

$$
\lesssim \sum_{j=0}^{\tau-1}\left(\frac{2\|B_k - B^\star\|}{\gamma_H} \cdot \mathbb{E}\left[\|S\|\|S^\dagger\|\right] + \|Z_{k,j} - \widetilde{Z}_{k,j}^\star\|\right) \quad \text{(by Lemma B.1)}
$$

$$
\lesssim \|B_k - B^\star\| + \sum_{j=0}^{\tau-1}\|B_k S_{k,j}(S_{k,j}^\top B_k^2 S_{k,j})^\dagger S_{k,j}^\top B_k - B^\star S_{k,j}(S_{k,j}^\top (B^\star)^2 S_{k,j})^\dagger S_{k,j}^\top B^\star\|
$$

$$
\lesssim \|B_k - B^\star\| + \|B_k - B^\star\| \sum_{j=0}^{\tau-1}\|S_{k,j}\|\|S_{k,j}^\dagger\| \quad \text{(by Lemma B.1)}, \tag{B.99}
$$

where the first inequality also uses the fact that

$$
\prod_{i=1}^n A_i - \prod_{i=1}^n B_i = \sum_{k=1}^n\left(\prod_{i=1}^{k-1} A_i\right)(A_k - B_k)\left(\prod_{j=k+1}^n B_j\right).
$$

Given the definitions of $\widetilde{K}_k$ in (3.2) and $\widetilde{K}_k^\star$ in (B.78), squaring both sides of (B.99) and taking conditional expectations give us

$$
\mathbb{E}\left[\|\widetilde{K}_k - \widetilde{K}_k^\star\|^2 \mid \mathcal{F}_{k-1}\right] = \mathbb{E}\left[\left\|(I \quad \mathbf{0})(\widetilde{C}_k - \widetilde{C}_k^\star)\begin{pmatrix}I \\ I\end{pmatrix}\right\|^2 \Big| \mathcal{F}_{k-1}\right]
$$

$$
\overset{(\text{B.99})}{\lesssim} \|B_k - B^\star\|^2 + \|B_k - B^\star\|^2 \cdot \mathbb{E}\left[\left(\sum_{j=0}^{\tau-1}\|S_{k,j}\|\|S_{k,j}^\dagger\|\right)^2\right] \lesssim \|B_k - B^\star\|^2, \tag{B.100}
$$

where the last inequality is due to Assumption 3.4 with $q_S = 2$. Plugging (B.100) into (B.98) yields

$$
\mathbb{E}\left[III'\right] \lesssim \mathbb{E}\left[\|B_k - B^\star\|^2\right]. \tag{B.101}
$$

Combining (B.97) and (B.101), we complete the proof.

### B.16. Proof of Lemma B.9

Recalling the decomposition (B.73), we have shown the consistency of the dominant term $\frac{1}{t}\sum_{i=1}^t \mathcal{L}_{1,i}\mathcal{L}_{1,i}^\top$ in Lemma B.8. The next lemma suggests that the terms involving $\{\mathcal{L}_{2,i}\}_i$ are higher order errors.

**Lemma B.15.** *Under the conditions of Theorem 4.6, we have*

$$
\mathbb{E}\left[\left\|\frac{1}{t}\sum_{i=1}^t \mathcal{L}_{2,i}\mathcal{L}_{2,i}^\top\right\|\right] \leq \frac{1}{t}\sum_{i=1}^t \mathbb{E}\left[\|\mathcal{L}_{2,i}\|^2\right] \lesssim \varphi_t.
$$

With the lemma given above, we are now ready to prove the lemma.

• **Proof of** (B.74). By the decomposition (B.73), we follow similar procedures given in (B.81) and (B.82) and have

$$
\mathbb{E}\left[\left\|\frac{1}{t}\sum_{i=1}^{t}\frac{1}{\varphi_{i-1}}(\boldsymbol{x}_i - \boldsymbol{x}^\star)(\boldsymbol{x}_i - \boldsymbol{x}^\star)^\top - \Sigma^\star\right\|\right]
$$
$$
\leq \mathbb{E}\left[\left\|\frac{1}{t}\sum_{i=1}^{t}\mathcal{L}_{1,i}\mathcal{L}_{1,i}^\top - \Sigma^\star\right\|\right] + \mathbb{E}\left[\left\|\frac{1}{t}\sum_{i=1}^{t}\mathcal{L}_{2,i}\mathcal{L}_{2,i}^\top\right\|\right] + 2\sqrt{\frac{1}{t}\sum_{i=1}^{t}\mathbb{E}\left[\|\mathcal{L}_{1,i}\|^2\right]}\sqrt{\frac{1}{t}\sum_{i=1}^{t}\mathbb{E}\left[\|\mathcal{L}_{2,i}\|^2\right]}. \quad \text{(B.102)}
$$

Given Lemmas B.8 and B.15, the remaining is to establish the bound for $\frac{1}{t}\sum_{i=1}^{t}\mathbb{E}\left[\|\mathcal{L}_{1,i}\|^2\right]$. We first bound the moment for $\|\boldsymbol{\theta}_k\|$. Based on its definition (B.20a) and Lemma 3.5, we have

$$
\boldsymbol{\theta}_k = -(I - \widetilde{K}_k)B_k^{-1}(g_k - \nabla f_k) + (\widetilde{K}_k - K_k)B_k^{-1}\nabla f_k.
$$

Furthermore, by Assumption 3.2 with $q_g = 2$, Lemma 3.7, (A.23), (A.28) and the condition on $\tau$, we get

$$
\mathbb{E}\left[\|\boldsymbol{\theta}_k\|^2\right] \lesssim \frac{1}{\gamma_H^2}\mathbb{E}\left[\|g_k - \nabla f_k\|^2\right] + \frac{1}{\gamma_H^2}\mathbb{E}\left[\|\nabla f_k\|^2\right] \leq \mathbb{E}\left[\|\boldsymbol{x}_k - \boldsymbol{x}^\star\|^2\right] + 1 + \mathbb{E}\left[\|\boldsymbol{x}_k - \boldsymbol{x}^\star\|^2\right]
$$
$$
\lesssim 1 \quad (\text{ by Lemma 4.5}). \quad \text{(B.103)}
$$

Since $(\boldsymbol{\theta}_k)$ is a martingale difference sequence, we follow (B.96) and get

$$
\frac{1}{t}\sum_{i=1}^{t}\mathbb{E}\left[\|\mathcal{L}_{1,i}\|^2\right] \leq \frac{1}{t}\sum_{i=0}^{t-1}\frac{1}{\varphi_i}\sum_{k=0}^{i}\varphi_k^2\prod_{l=k+1}^{i}(1 - \varphi_l\lambda_m)^2\mathbb{E}\left[\|\boldsymbol{\theta}_k\|^2\right] \overset{\text{(B.103)}}{\lesssim} \frac{1}{t}\sum_{i=0}^{t-1}\frac{1}{\varphi_i}\cdot\varphi_i \lesssim 1, \quad \text{(B.104)}
$$

where the second inequality also uses Na & Mahoney (2025, Lemma B.3). Combining the above display, Lemmas B.8 and B.15, and plugging them into (B.102), we obtain

$$
\mathbb{E}\left[\left\|\frac{1}{t}\sum_{i=1}^{t}\frac{1}{\varphi_{i-1}}(\boldsymbol{x}_i - \boldsymbol{x}^\star)(\boldsymbol{x}_i - \boldsymbol{x}^\star)^\top - \Sigma^\star\right\|\right] \lesssim \frac{1}{\sqrt{t\varphi_t}} + \sqrt{\varphi_t} \lesssim \frac{1}{\sqrt{t\varphi_t}}.
$$

• **Proof of** (B.75). By the decomposition (B.73), we have

$$
\frac{1}{t}\sum_{i=1}^{t}\frac{1}{\varphi_{i-1}}\mathbb{E}\left[\|\boldsymbol{x}_i - \boldsymbol{x}^\star\|^2\right] \lesssim \sum_{k=1}^{2}\frac{1}{t}\sum_{i=0}^{t-1}\mathbb{E}\left[\|\mathcal{L}_{k,i}\|^2\right] \lesssim 1,
$$

where the last inequality follows from (B.104) and Lemma B.15.

This completes the proof.

### B.17. Proof of Lemma B.15

For the term $\mathcal{L}_{2,i}$ defined in (B.72), by the Cauchy–Schwarz inequality, we have

$$
\frac{1}{t}\sum_{i=1}^{t}\mathbb{E}\left[\|\mathcal{L}_{2,i}\|^2\right] \lesssim \frac{1}{t}\sum_{i=0}^{t-1}\frac{1}{\varphi_i}\prod_{k=0}^{i}(1 - \varphi_k\lambda_m)^2\|\boldsymbol{x}_0 - \boldsymbol{x}^\star\|^2 + \frac{1}{t}\sum_{i=0}^{t-1}\frac{1}{\varphi_i}\left(\sum_{k=0}^{i}\prod_{l=k+1}^{i}(1 - \varphi_l\lambda_m)\varphi_k\sqrt{\mathbb{E}\left[\|\boldsymbol{\delta}_k\|^2\right]}\right)^2. \quad \text{(B.105)}
$$

Then, we focus on the rate of $\mathbb{E}\left[\|\boldsymbol{\delta}_k\|^2\right]$. By the definition of $\boldsymbol{\delta}_k$ in (B.20b), we have

$$
\|\boldsymbol{\delta}_k\|^2 \lesssim \|K_k - K^\star\|^2\|\boldsymbol{x}_k - \boldsymbol{x}^\star\|^2 + \|(B^\star)^{-1}\|^2\|\nabla f_k - B^\star(\boldsymbol{x}_k - \boldsymbol{x}^\star)\|^2 + \|B_k^{-1}\|^2\|(B^\star)^{-1}\|^2\|B_k - B^\star\|^2\|\nabla f_k\|^2
$$
$$
\lesssim \|B_k - B^\star\|^2\|\boldsymbol{x}_k - \boldsymbol{x}^\star\|^2 + \|\boldsymbol{x}_k - \boldsymbol{x}^\star\|^4 + \|B_k - B^\star\|^2\|\boldsymbol{x}_k - \boldsymbol{x}^\star\|^2,
$$

where the second inequality is due to Assumption 3.1, (A.23), (A.26) and (B.14). Taking the expectation on both sides yields

$$\mathbb{E}\left[\|\boldsymbol{\delta}_k\|^2\right] \lesssim \mathbb{E}\left[\|B_k - B^\star\|^2\|\boldsymbol{x}_k - \boldsymbol{x}_k^\star\|^2\right] + \mathbb{E}\left[\|\boldsymbol{x}_k - \boldsymbol{x}^\star\|^4\right]$$
$$\lesssim \sqrt{\mathbb{E}\left[\|B_k - B^\star\|^4\right]}\sqrt{\mathbb{E}\left[\|\boldsymbol{x}_k - \boldsymbol{x}^\star\|^4\right]} + \mathbb{E}\left[\|\boldsymbol{x}_k - \boldsymbol{x}^\star\|^4\right] \lesssim \varphi_k^2, \quad \text{(B.106)}$$

where the last inequality follows from Lemma 4.5. Plugging (B.106) into (B.105) gives us

$$\frac{1}{t}\sum_{i=1}^t \mathbb{E}\left[\|\mathcal{L}_{2,i}\|^2\right] \lesssim \frac{1}{t}\sum_{i=0}^{t-1}\frac{1}{\varphi_i}\prod_{k=0}^i (1-\varphi_k\lambda_m)^2\|\boldsymbol{x}_0 - \boldsymbol{x}^\star\|^2 + \frac{1}{t}\sum_{i=0}^{t-1}\frac{1}{\varphi_i}\left(\sum_{k=0}^i\prod_{l=k+1}^i (1-\varphi_l\lambda_m)\varphi_k^2\right)^2$$
$$\lesssim \frac{1}{t}\sum_{i=0}^{t-1}\frac{1}{\varphi_i}o(\varphi_i^2) + \frac{1}{t}\sum_{i=0}^{t-1}\frac{1}{\varphi_i}\varphi_i^2 \lesssim \varphi_t,$$

where the last inequality is the direct application of Na & Mahoney (2025, Lemma B.3). We complete the proof.

### B.18. Proof of Lemma B.10

The first inequality is trivial. We only show the second inequality based on the decomposition (B.38). In particular, we decompose $\bar{\boldsymbol{x}}_t - \boldsymbol{x}^\star$ as

$$\bar{\boldsymbol{x}}_t - \boldsymbol{x}^\star = \frac{1}{t}\sum_{i=0}^{t-1}\sum_{k=0}^i\prod_{l=k+1}^i (I - \varphi_l\mathcal{R})\varphi_k\boldsymbol{\theta}_k$$
$$+ \left[\frac{1}{t}\sum_{i=0}^{t-1}\prod_{k=0}^i (I - \varphi_k\mathcal{R})(\boldsymbol{x}_0 - \boldsymbol{x}^\star) + \frac{1}{t}\sum_{i=0}^{t-1}\sum_{k=0}^i\prod_{l=k+1}^i (I - \varphi_l\mathcal{R})\varphi_k\boldsymbol{\delta}_k\right] =: \mathcal{Z}_{1,t} + \mathcal{Z}_{2,t}. \quad \text{(B.107)}$$

For the term $\mathcal{Z}_{1,t}$, exchanging the indices gives us

$$\mathcal{Z}_{1,t} = \frac{1}{t}\sum_{i=0}^{t-1}\sum_{k=0}^i\prod_{l=k+1}^i (I - \varphi_l\mathcal{R})\varphi_k\boldsymbol{\theta}_k = \frac{1}{t}\sum_{k=0}^{t-1}\sum_{i=k}^{t-1}\prod_{l=k+1}^i (I - \varphi_l\mathcal{R})\varphi_k\boldsymbol{\theta}_k.$$

Since $(\boldsymbol{\theta}_k)$ is a martingale difference sequence, the interaction terms in $\mathbb{E}\left[\|\mathcal{Z}_{1,t}\|^2\right]$ vanish. Therefore, we have

$$\mathbb{E}\left[\|\mathcal{Z}_{1,t}\|^2\right] = \frac{1}{t^2}\sum_{k=0}^{t-1}\varphi_k^2\mathbb{E}\left[\left\|\sum_{i=k}^{t-1}\prod_{l=k+1}^i (I - \varphi_l\mathcal{R})\boldsymbol{\theta}_k\right\|^2\right] \le \frac{1}{t^2}\sum_{k=0}^{t-1}\left[\sum_{i=k}^{t-1}\prod_{l=k+1}^i (1 - \varphi_l\lambda_m)\right]^2\varphi_k^2\mathbb{E}\left[\|\boldsymbol{\theta}_k\|^2\right] =: (\#),$$

which can be rewritten by exchanging the indices as

$$(\#) = \frac{1}{t^2}\sum_{k=0}^{t-1}\sum_{i_1=k}^{t-1}\sum_{i_2=k}^{t-1}\prod_{l_1=k+1}^{i_1}(1-\varphi_{l_1}\lambda_m)\prod_{l_2=k+1}^{i_2}(1-\varphi_{l_2}\lambda_m)\varphi_k^2\mathbb{E}\left[\|\boldsymbol{\theta}_k\|^2\right]$$
$$= \frac{1}{t^2}\sum_{i_1=0}^{t-1}\sum_{i_2=0}^{t-1}\sum_{k=0}^{i_1\wedge i_2}\prod_{l_1=k+1}^{i_1}(1-\varphi_{l_1}\lambda_m)\prod_{l_2=k+1}^{i_2}(1-\varphi_{l_2}\lambda_m)\varphi_k^2\mathbb{E}\left[\|\boldsymbol{\theta}_k\|^2\right]$$
$$\le \frac{2}{t^2}\sum_{i_1=0}^{t-1}\sum_{i_2=0}^{i_1}\prod_{l_1=i_2+1}^{i_1}(1-\varphi_{l_1}\lambda_m)\sum_{k=0}^{i_2}\prod_{l_2=k+1}^{i_2}(1-\varphi_{l_2}\lambda_m)^2\varphi_k^2\mathbb{E}\left[\|\boldsymbol{\theta}_k\|^2\right],$$

where the last inequality is due to the symmetry between the indices $i_1$ and $i_2$. We plug in (B.103) and get

$$\mathbb{E}\left[\|\mathcal{Z}_{1,t}\|^2\right] \lesssim \frac{1}{t^2}\sum_{i_1=0}^{t-1}\sum_{i_2=0}^{i_1}\prod_{l_1=i_2+1}^{i_1}(1-\varphi_{l_1}\lambda_m)\sum_{k=0}^{i_2}\prod_{l_2=k+1}^{i_2}(1-\varphi_{l_2}\lambda_m)^2\varphi_k^2$$
$$\lesssim \frac{1}{t^2}\sum_{i_1=0}^{t-1}\sum_{i_2=0}^{i_1}\prod_{l_1=i_2+1}^{i_1}(1-\varphi_{l_1}\lambda_m)\varphi_{i_2} \lesssim \frac{1}{t^2}\sum_{i_1=0}^{t-1}1 = \frac{1}{t}, \quad \text{(B.108)}$$

where the last two inequalities are due to Na & Mahoney (2025, Lemma B.3). For the term $\mathcal{Z}_{2,t}$, similarly to the analysis in (B.105), we have

$$\mathbb{E}\left[\|\mathcal{Z}_{2,t}\|^2\right] \lesssim \left(\frac{1}{t}\sum_{i=0}^{t-1}\prod_{k=0}^{i}(1-\varphi_k\lambda_m)\right)^2 \|\boldsymbol{x}_0-\boldsymbol{x}^\star\|^2 + \left(\frac{1}{t}\sum_{i=0}^{t-1}\sum_{k=0}^{i}\prod_{l=k+1}^{i}(1-\varphi_l\lambda_m)\varphi_k\sqrt{\mathbb{E}\left[\|\boldsymbol{\delta}_k\|^2\right]}\right)^2$$

$$\overset{(B.106)}{\lesssim}\left(\frac{1}{t}\sum_{i=0}^{t-1}\prod_{k=0}^{i}(1-\varphi_k\lambda_m)\right)^2 \|\boldsymbol{x}_0-\boldsymbol{x}^\star\|^2 + \left(\frac{1}{t}\sum_{i=0}^{t-1}\sum_{k=0}^{i}\prod_{l=k+1}^{i}(1-\varphi_l\lambda_m)\varphi_k^2\right)^2$$

$$\lesssim \left(\frac{1}{t}\sum_{i=0}^{t-1}o(\varphi_i)\right) + \left(\frac{1}{t}\sum_{i=0}^{t-1}\varphi_i\right)^2 \lesssim \varphi_t^2, \tag{B.109}$$

where the third inequality is again due to Na & Mahoney (2025, Lemma B.3). Combining (B.107), (B.108) and (B.109) together yields

$$\frac{1}{t}\sum_{i=0}^{t-1}\frac{1}{\varphi_i}\mathbb{E}\left[\|\bar{\boldsymbol{x}}_t-\boldsymbol{x}^\star\|^2\right] \lesssim \frac{1}{t^2}\sum_{i=0}^{t-1}\frac{1}{\varphi_i} \lesssim \frac{1}{t\varphi_t},$$

where the last inequality is due to the fact that $\sum_{i=0}^{t-1}1/\varphi_i \lesssim \int_0^t i^\varphi di \lesssim t/\varphi_t$. This completes the proof.

# C. Additional Experimental Results

In this section, we follow the experiments in Section 5 and introduce detailed experimental setups along with additional experimental results.

## C.1. Detailed experimental setup

For the linear regression problem, we consider the model $\xi_b = \xi_a^\top \boldsymbol{x}^\star + \varepsilon$, where $\varepsilon \sim \mathcal{N}(0,\sigma^2)$ is Gaussian noise. For this model, we use the squared loss $F(\boldsymbol{x};\xi) = 0.5(\xi_b - \xi_a^\top\boldsymbol{x})^2$.

For the logistic regression problem, we consider the model $P(\xi_b|\xi_a) = \frac{\exp(\xi_b\cdot\xi_a^\top\boldsymbol{x}^\star)}{1+\exp(\xi_b\cdot\xi_a^\top\boldsymbol{x}^\star)}$ with $\xi_b \in \{-1,1\}$. For this model, we use the log loss $F(\boldsymbol{x};\xi) = \log(1+\exp(-\xi_b\cdot\xi_a^\top\boldsymbol{x}))$.

• **Model parameters setup.** For both regression tasks, we consider dimensions $d \in \{20,40\}$ and set the ground-truth model parameter $\boldsymbol{x}^\star \in \mathbb{R}^d$ to be linearly spaced between 0 and 1. For the linear model, we fix the noise variance at $\sigma^2 = 1$. For each dimension $d$, we generate the covariate $\xi_a \sim \mathcal{N}(\boldsymbol{0},\Sigma_a)$ with three different types of $\Sigma_a$. (i) Identity matrix: $\Sigma_a = I$; (ii) Equi-correlation matrix: $[\Sigma_a]_{i,i} = 1$ and $[\Sigma_a]_{i,j} = r$ for $i \neq j$; and (iii) Toeplitz matrix: $[\Sigma_a]_{i,j} = r^{|i-j|}$. For the latter two covariance matrices, we set $r = 0.4$.

• **Algorithm parameters setup.** We vary the number of sketching steps $\tau \in \{5,10,\infty\}$, where $\tau = \infty$ corresponds to solving (2.2) exactly. For the sketching solver, we apply both randomized Kaczmarz sampling $S \sim \text{Uniform}(\{\boldsymbol{e}_i\}_{i=1}^d)$ (Strohmer & Vershynin, 2008) and Gaussian sampling $S \sim \mathcal{N}(0,I_d)$. We set $(\alpha_t,\beta_t,\gamma_t)$ according to the formulas in (2.4), with $(\mu_t,\nu_t)$ approximated by empirical averages based on $B_t$ (see (2.5)). To examine cheaper parameter updates, we recompute these approximations every $N \in \{1,500,1000\}$ iterations and keep them fixed between refreshes; $N = 1$ recovers the standard non-periodic setting. In the QQ-plot experiments comparing accelerated and unaccelerated sketching, we also include the degenerate specification $\mu_t\nu_t = \gamma_t = 1$. For the online Newton method, we use a diminishing stepsize $\varphi_t = 1/(t+1)^{0.501}$. We set the nominal confidence level to 95%, and perform inference on the coordinate-wise mean of $\boldsymbol{x}^\star$, i.e. $\sum_{i=1}^d \boldsymbol{x}_i^\star/d$. For each parameter choice, we conduct 200 independent runs.

## C.2. Additional results and discussions

In this section, we present and discuss additional experimental results. Table 2 reports numerical experiments for $d = 20$ on both linear and logistic regression problems, including the standard SGD baseline and periodic refreshes $N \in \{1,500,1000\}$ for the accelerated parameters. For the Newton-type methods, the empirical coverage rates consistently remain close to the nominal level of 95% across most parameter configurations and refresh periods. The similar coverage behavior across different values of $N$ further suggests that recomputing $(\mu_t,\nu_t)$ at every iteration is not necessary for maintaining valid inference in these examples.

| $d$ | $\Sigma_a$ | $\tau$ | Linear Regression (Kaczmarz) | | | Linear Regression (Gaussian) | | | Logistic Regression (Kaczmarz) | | | Logistic Regression (Gaussian) | | |
|---|---|---|---|---|---|---|---|---|---|---|---|---|---|---|
| | | | MAE $(10^{-2})$ | Ave Cov (%) | Ave Len $(10^{-2})$ | MAE $(10^{-2})$ | Ave Cov (%) | Ave Len $(10^{-2})$ | MAE $(10^{-2})$ | Ave Cov (%) | Ave Len $(10^{-2})$ | MAE $(10^{-2})$ | Ave Cov (%) | Ave Len $(10^{-2})$ |
| 20 | Identity | SGD | 17.52 | 64.50 | 0.64 | — | — | — | 6.11 | 81.50 | 0.63 | — | — | — |
| | | $\infty$ | 18.07 | 96.50 | 3.53 | 17.81 | 97.50 | 3.53 | 2.92 | 94.50 | 0.42 | 2.93 | 95.00 | 0.42 |
| | | 10 | 17.60 | 95.00 | 3.52 | 18.60 | 97.00 | 3.74 | 2.95 | 94.00 | 0.52 | 3.11 | 93.50 | 0.54 |
| | | | 17.81 | 92.00 | 3.53 | 18.64 | 95.50 | 3.72 | 2.98 | 93.50 | 0.52 | 3.16 | 97.00 | 0.55 |
| | | | 17.77 | 93.50 | 3.53 | 19.09 | 94.50 | 3.74 | 2.98 | 93.50 | 0.52 | 3.10 | 97.50 | 0.55 |
| | | 5 | 17.87 | 96.50 | 3.48 | 18.40 | 90.50 | 3.59 | 3.01 | 94.50 | 0.54 | 3.10 | 95.50 | 0.54 |
| | | | 17.74 | 94.00 | 3.51 | 18.28 | 95.00 | 3.52 | 2.99 | 96.00 | 0.55 | 3.10 | 91.50 | 0.55 |
| | | | 18.00 | 94.50 | 3.50 | 18.33 | 98.00 | 3.55 | 2.99 | 95.00 | 0.54 | 3.13 | 94.00 | 0.55 |
| | Toeplitz $r = 0.4$ | SGD | 17.85 | 70.50 | 0.65 | — | — | — | 5.19 | 77.00 | 0.63 | — | — | — |
| | | $\infty$ | 20.62 | 94.50 | 2.38 | 20.84 | 96.50 | 2.38 | 2.48 | 92.50 | 0.47 | 4.54 | 94.50 | 0.56 |
| | | 10 | 16.25 | 94.50 | 3.04 | 17.47 | 93.50 | 3.05 | 2.53 | 92.50 | 0.48 | 4.81 | 94.00 | 0.80 |
| | | | 16.15 | 94.00 | 3.03 | 17.67 | 96.00 | 3.04 | 2.48 | 93.00 | 0.48 | 4.75 | 93.00 | 0.80 |
| | | | 16.56 | 96.50 | 3.02 | 17.14 | 93.50 | 3.03 | 2.53 | 94.00 | 0.49 | 4.76 | 97.50 | 0.80 |
| | | 5 | 15.69 | 96.00 | 3.04 | 16.97 | 95.00 | 3.02 | 2.52 | 94.00 | 0.48 | 4.57 | 95.50 | 0.81 |
| | | | 15.83 | 94.50 | 3.02 | 16.46 | 93.50 | 3.03 | 2.53 | 95.50 | 0.49 | 4.63 | 95.50 | 0.80 |
| | | | 15.79 | 95.00 | 3.04 | 16.79 | 96.00 | 3.05 | 2.53 | 92.00 | 0.48 | 4.56 | 92.50 | 0.80 |
| | Equi-Corr $r = 0.4$ | SGD | 18.05 | 79.00 | 0.62 | — | — | — | 5.10 | 80.50 | 0.63 | — | — | — |
| | | $\infty$ | 13.56 | 94.50 | 1.14 | 13.77 | 92.50 | 1.14 | 2.53 | 95.50 | 0.41 | 2.48 | 97.00 | 0.41 |
| | | 10 | 10.82 | 95.50 | 1.69 | 13.05 | 97.00 | 1.68 | 2.44 | 93.50 | 0.46 | 2.60 | 94.50 | 0.48 |
| | | | 10.75 | 95.50 | 1.67 | 13.77 | 92.00 | 1.68 | 2.51 | 96.00 | 0.46 | 2.62 | 94.00 | 0.49 |
| | | | 10.79 | 98.00 | 1.68 | 13.36 | 92.00 | 1.69 | 2.48 | 95.00 | 0.46 | 2.65 | 98.00 | 0.48 |
| | | 5 | 10.33 | 95.00 | 1.74 | 12.89 | 95.00 | 1.77 | 2.49 | 96.50 | 0.47 | 2.52 | 94.50 | 0.48 |
| | | | 10.23 | 97.00 | 1.75 | 12.76 | 96.00 | 1.78 | 2.48 | 94.00 | 0.47 | 2.62 | 97.50 | 0.48 |
| | | | 10.44 | 94.50 | 1.75 | 12.81 | 96.00 | 1.78 | 2.49 | 94.00 | 0.47 | 2.59 | 96.00 | 0.48 |

*Table 2. A subset of evaluation results for the inference procedure under different parameter choices with $d = 20$. "MAE" denotes the mean absolute iterate error over 200 independent runs, "Ave Cov" denotes the empirical coverage rate, and "Ave Len" denotes the average confidence-interval length. The row labeled "SGD" reports the standard SGD baseline. Since SGD does not involve sketching, there is no distinction between the Kaczmarz and Gaussian variants; accordingly, the SGD results are reported only once for each regression model, and redundant entries are marked by "—". For the sketching-based methods with $\tau \in \{5, 10\}$, each three-row block, from top to bottom, corresponds to a different refresh period $N \in \{1, 500, 1000\}$ for the acceleration parameters $(\mu_t, \nu_t)$. Here, $N = 1$ recovers the standard non-periodic setting.*

One exception is the linear regression case under Gaussian sketching. When $\Sigma_a$ is the identity and the number of sketching steps $\tau = 5$, the empirical coverage rate drops to $90.5\%$, which is slightly below the nominal level. This mild undercoverage may be attributed to the estimation error in the weighted sample covariance estimator $\widehat{\Sigma}_t$ and the approximation error introduced by the sketching solver when $\tau$ is small. These numerical effects may lead to narrower confidence intervals in finite samples. Nevertheless, the observed coverage rates of Newton-type methods remain within an acceptable range for practical inference. In contrast, the SGD baseline exhibits more pronounced undercoverage in several settings, consistent with prior findings on first-order online inference. Overall, the close alignment of the Newton-type methods with the target coverage level provides numerical validations for the asymptotic normality established in Theorem 4.3 and confirms the consistency of the proposed weighted sample covariance estimator in Theorem 4.6. Notably, even with a limited number of sketching steps ($\tau = 5$), the coverage rates remain largely comparable to those achieved by the exact Newton method ($\tau = \infty$) in almost all cases, demonstrating the robustness of the accelerated sketching approach for statistical inference.

In terms of the mean absolute error, our proposed Newton method achieves comparable iteration error with the exact Newton method even with a small sketching steps $\tau$. Similar to the case with $d = 40$, in some particular settings, e.g., $\Sigma_a$ is Toeplitz or Equi-correlation in linear regression problems, the iteration error of the inexact Newton method is even smaller than that of the exact Newton method. Furthermore, the average confidence interval lengths exhibit only modest increases when the sketching solver is applied. Thus, these results indicate that our proposed inexact Newton method preserves statistical efficiency.

Finally, we present additional experiments on QQ plots. Figures 2 and 3 report quantile–quantile comparisons between the Newton method with accelerated sketching and the Newton method with unaccelerated sketching for logistic regression, under two sketching distributions: randomized Gaussian sketching (Figure 2) and Kaczmarz sketching (Figure 3). Across all configurations, the scatter points closely align with the 45-degree reference line, indicating that the quantiles of the accelerated

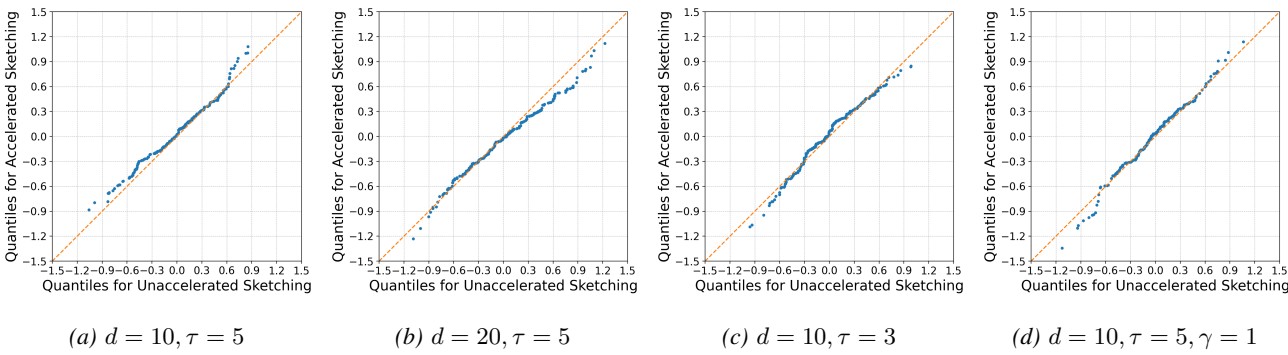

*(a)* $d = 10, \tau = 5$     *(b)* $d = 20, \tau = 5$     *(c)* $d = 10, \tau = 3$     *(d)* $d = 10, \tau = 5, \gamma = 1$

*Figure 2. QQ plots for logistic regression with Gaussian accelerated v.s. unaccelerated sketching. The yellow line refers to $y = x$.*

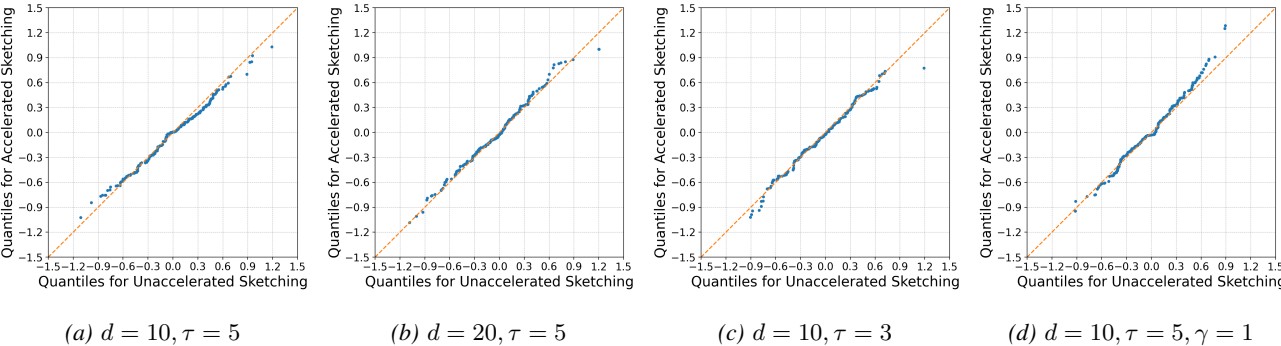

*(a)* $d = 10, \tau = 5$     *(b)* $d = 20, \tau = 5$     *(c)* $d = 10, \tau = 3$     *(d)* $d = 10, \tau = 5, \gamma = 1$

*Figure 3. QQ plots for logistic regression with Kaczmarz accelerated v.s. unaccelerated sketching. The yellow line refers to $y = x$.*

sketching scheme closely match those of the unaccelerated sketching scheme. This close agreement is observed consistently for both sketching distributions (Kaczmarz and Gaussian), across different problem dimensions ($d = 10, 20$), and for varying numbers of sketching steps ($\tau = 3, 5$). The alignment persists even in the degenerate case $\gamma = \mu\nu = 1$, which is consistent with Proposition 4.4(b). These results lead to two conclusions for logistic regression problems. First, the proposed Newton method with accelerated sketching satisfies the asymptotic normality established in Theorem 4.3. Second, Nesterov's acceleration does not degrade statistical efficiency, as the limiting covariance of the accelerated sketching scheme remains comparable to that of the unaccelerated sketching scheme.

