# OpenReview forum: "Inference of Online Newton Methods with Nesterov's Accelerated Sketching"
_ICML.cc/2026/Conference — ICML 2026 regular_

### Official Review · Reviewer_EmcX · 2026-03-12

**Soundness:** 3
**Presentation:** 3
**Significance:** 3
**Originality:** 3
**Overall Recommendation:** 4
**Confidence:** 3

**Summary:**

This paper investigates online Newton methods that incorporate Hessian averaging. The Newton step is approximated using a sketch-and-project linear solver, which is further accelerated by Nesterov's method (Algorithm 1, p.4; outer update, Eq. (2.8), p.4). The paper's main contribution is a comprehensive inference framework for this second-order streaming method. This framework encompasses: (i) a proof of global almost-sure convergence (Theorem 3.8, p.6); (ii) a characterization of its last-iterate asymptotic normality, where the limiting covariance is defined by a Lyapunov equation dependent on the solver operator (Theorem 4.3, p.7); and (iii) a fully online covariance estimator that comes with a non-asymptotic convergence guarantee.

**Compliance With Llm Reviewing Policy:**

Affirmed.

**Key Questions For Authors:**

1. Are there regimes where acceleration improves solver speed but worsens statistical efficiency (via changed covariance), or do you expect monotone improvement?

2. Can you clarify the decomposition of uncertainty: data vs computational (sketching)? Does the proposed estimator cover both?

3. What is the minimal set of assumptions needed for the inference results (asymptotic normality + Lyapunov equation)?

**Limitations:**

I suggest including a compact "theorem map" early in the paper, perhaps at the end of the Introduction. This map would outline the logical flow: global convergence → asymptotic normality → covariance estimation, and clearly indicate the minimal additional assumptions required at each stage. While Sections 6–7 already follow this structure, an upfront roadmap would significantly reduce the reader's cognitive load and help them navigate the theoretical contributions more easily.

**Strengths And Weaknesses:**

Strengths

1. The Lyapunov characterization provides a clean conceptual handle. The covariance formula makes the "computational–statistical trade-off" explicit through the limiting solver operator   (see discussion around Eq. (1.4) on p.2 and Eq. (4.2) on p.7). Furthermore, the paper connects special cases back to exact Newton and unaccelerated sketching (Proposition 4.4, p.7).

2. The online covariance estimator is practically oriented. The estimator (\hat\Sigma_t) is constructed from iterates and supports confidence intervals in a streaming fashion (p.7 Eq. (4.4) and Theorem 4.6). The empirical coverage results are aligned with the theory (Table 1 p.8; Table 2 p.50).

3. Interesting and nonstandard angle: most work on sketching for Newton focuses on optimization error; connecting accelerated sketching with statistical inference is distinctive.

Weaknesses

1. In the core algorithm, $((\alpha_t,\beta_t,\gamma_t))$ are defined via ($(\mu_t,\nu_t)$) (p.4 Eq. (2.4)–(2.5)), but $(\mu_t,\nu_t)$ are generally intractable.

2. A potential limitation is the reliance on fairly strong assumptions—namely, smoothness, convexity, moment conditions, and constraints on the sketch distribution. Consequently, the applicability of the results may not readily extend beyond relatively benign settings such as generalized linear models.

---

> ### Author Rebuttal · Authors · 2026-03-27
>
> **Thank you for your careful reading and positive opinion of the paper. All your questions are reasonable and can be answered, so we hope you can support the paper more affirmatively based on our responses.**
> ### 1, Response to W1
> Indeed. We explain in the response to Reviewer hYL1. We have also complemented experiments on two other periodic parameter updates to resolve this weakness. See new extended [Table 1](https://anonymous.4open.science/r/Testttttttt-92B485L7/table1.png) and [Table 2](https://anonymous.4open.science/r/Testttttttt-92B485L7/table2.png) (URLs and the destination are anonymous).
> ### 2, Response to W2&Q3
> Indeed, we focus on smooth and strongly convex stochastic objectives, with gradient and Hessian estimates satisfying moment growth conditions, and proper sketching distributions. That said, we would like to justify the following points.
>
> **(1)** All assumptions are standard (if not weaker) in the literature (Chen et al., 2020; Zhu et al., 2021; Na and Mahoney, 2025; Kuang et al., 2025). In particular, sketching conditions are satisfied for common Gaussian and Kaczmarz sketches. Gradient and Hessian moment conditions are verified for linear and logistic regressions (Chen et al., 2020), **and beyond** (see response to Reviewer hYL1). Smoothness is a natural assumption for studying second-order methods. Strong convexity is important for global convergence; a nonconvex setting generally only ensures convergence to a stationary point.
>
> **(2)** One way to relax strong convexity is to convexify the Hessian matrix at each step to ensure a descent direction. However, this only ensures global convergence to a stationary point, as done in (Na and Mahoney, 2025, Section 3). For online inference, local strong convexity is still required and quite essential in all existing literature. Indeed, the minimax optimal analyses in Duchi and Ruan (2021); Davis et al. (2024), when reduced to the unconstrained case, even require local convexity.
>
> For our analysis, if we assume the global convergence of the sketched Newton iterates (as assumed in some work for other methods, e.g., Davis et al. (2024)), **the minimal set of assumptions needed for normality and Lyapunov equation are**: local strong convexity & $q_g>2$ moment for gradient & $q_H=2$ moment for Hessian & bounded expected sketching condition number ($q_S=1$). For covariance estimation, we only need: local strong convexity & $q_g=q_H=4$ & $q_S=2$. Both precisely match the aforementioned literature.
>
> **(3)** To our knowledge, no study allows uniformly weaker conditions. In addition to relaxing to local strong convexity while weakening global convergence to convergence to a stationary point, Davis et al. (2024); Jiang et al. (2025) relax the smoothness condition; however, their normality requires $q_g = 4$, and covariance estimation requires sub-exponential gradient noise.
> ### 3, Response to Q1
> We answer from two aspects. See responses to Reviewer QAUF for more details.
>
> **ASN v.s. EN.** It is always the case that ASN gains computational efficiency but sacrifices statistical efficiency, since $\Sigma_{asn}^\star\succeq \Omega^\star$.
>
> **ASN v.s. USN.** ASN gains computational efficiency as shown in Gower et al. (2018); Derezinski et al. (2025). Our analysis shows ASN does not degrade statistical efficiency; rather, the efficiency gap to optimality $\|\|\Sigma_{asn}^\star-\Omega^\star\|\| = O(\rho_{asn}^\tau)$ improves upon $\|\|\Sigma_{usn}^\star-\Omega^\star\|\| = O(\rho_{usn}^\tau)$, where $\rho_{asn}\leq\rho_{usn}$ is the accelerated rate. This will be clarified clearly in the final version.
> ### 4, Response to Q2
> For sketched Newton methods, data uncertainty enters through the stochastic gradient $g_t = \nabla F(x_t;\xi_t)$, while computational uncertainty enters through random sketching matrices in *NASketch* solver. Our covariance estimator $\hat{\Sigma}\_t$ is a consistent estimator of $\Sigma_{asn}^\star$ (Theorem 4.6); thus, it indeed captures both.
>
> In particular, $\Sigma_{asn}^\star$ is given by the Lyapunov equation (4.2), depending on $K^\star$, $\Gamma^\star$, $\Omega^\star$ in (4.1). Here, $\Omega^\star$ is the optimal covariance achieved by SGD and EN, with expectation taken over $\xi$, thereby capturing data uncertainty. $K^\star$ and $\Gamma^\star$ are sketching-related projection matrices, with expectations taken over sketching distributions, thereby capturing computational uncertainty. Thus, the Lyapunov (4.2) equation explicitly accounts for both sources of uncertainty.
> ### 5, Improvement
> We will add a theorem roadmap in Line 164 by suggestion.

---

> > ### Author Rebuttal · Reviewer_EmcX · 2026-04-04
> >
> > Thank you for the detailed rebuttal. My main questions were addressed in a satisfactory way.
> >
> > In particular, the clarification on the uncertainty decomposition was helpful. The rebuttal makes clear that the limiting covariance incorporates both sources of randomness: data uncertainty through the stochastic gradient term, and computational uncertainty through the randomized sketching operator. It also clarifies that the proposed online covariance estimator is intended to estimate the full limiting covariance, rather than only the data-driven part. This directly answers one of my main questions.
> >
> > The discussion of acceleration is also clearer after rebuttal. The authors’ response draws an important distinction between comparison to exact Newton and comparison to unaccelerated sketching: relative to exact Newton, there is a computational gain together with some statistical cost, whereas relative to unaccelerated sketching, acceleration does not appear to worsen statistical efficiency and may improve the gap to optimality. I think this is a useful and nontrivial point, and it should be stated more explicitly in the final version.
> >
> >
> > On the assumptions, the rebuttal clarifies both why the paper focuses on the smooth strongly convex regime and what the smaller assumption set is for the local inference results once convergence is granted. This addresses my question about the minimal assumptions. At the same time, the scope limitation remains real: the theory is still most directly applicable to relatively benign settings such as regression/GLM-type problems.
> >
> > Overall, the rebuttal strengthens my positive assessment. The paper remains technically solid, and the combination of global convergence, last-iterate asymptotic normality via a Lyapunov characterization, and a fully online covariance estimator is a meaningful contribution. I also appreciate the commitment to add a clearer theorem roadmap in the final version. I am maintaining my Weak Accept recommendation, with slightly increased confidence.

---

> > > ### Author Response · Authors · 2026-04-04
> > >
> > > Thank you for your recommendation. We agree with all your summaries. We sincerely appreciate your careful review and insightful suggestions, which have helped us improve both the clarity and positioning of the paper.
> > >
> > > Studying inference for more complex loss landscapes (e.g., inference at saddle points) is certainly a very promising yet challenging direction.

---

### Official Review · Reviewer_hYL1 · 2026-03-13

**Soundness:** 4
**Presentation:** 3
**Significance:** 3
**Originality:** 3
**Overall Recommendation:** 5
**Confidence:** 3

**Summary:**

This paper studies uncertainty quantification and statistical inference for an online
Newton method in which the Newton system is solved approximately via Nesterov's
accelerated sketch-and-project solver. The method inherits the $O(d^2)$ per-iteration
complexity of first-order methods while exploiting curvature information. Under standard
smoothness and moment conditions on a strongly convex stochastic objective, the authors
establish: (i) global almost-sure convergence of the outer Newton iterates $x_t$;
(ii) asymptotic normality of the last iterate, specifically
$\varphi_t^{-1/2}(x_t - x^\star) \xrightarrow{d} \mathcal{N}(0, \Sigma^\star)$,
where $\Sigma^\star$ is characterized by a Lyapunov equation that depends explicitly on
the accelerated sketching operator; and (iii) a fully online, consistent covariance
estimator $\hat{\Sigma}_t$ enabling practical confidence interval construction. The
paper connects these results to exact Newton (Leluc & Portier, 2023) and unaccelerated
sketched Newton (Na & Mahoney, 2025; Kuang et al., 2025) as degenerate cases.

**Compliance With Llm Reviewing Policy:**

Affirmed.

**Key Questions For Authors:**

1. **On the computational cost of acceleration parameters:** Computing $(\mu_t, \nu_t)$
   at each outer iteration requires evaluating $\lambda_{\min}(Z_t)$ and the
   second-moment quantity in (2.5), which may cost $O(d^2)$–$O(d^3)$ and undermine
   the claimed $O(d^2)$ per-iteration complexity. Do the authors compute these exactly
   in the experiments, or do they use fixed limiting parameters $(\alpha^\star,
   \beta^\star, \gamma^\star)$ as suggested by Lemma 4.1? If the latter, can this be
   theoretically justified and explicitly stated?

2. **On the quasi-Newton/finite-difference claim:** Assumption 3.3 requires unbiased
   Hessian estimates, which quasi-Newton and finite-difference approximations do not
   generally provide. Can the authors either (a) provide a concrete example of a
   setting where their analysis formally extends to approximate Hessian oracles, or
   (b) rephrase this as a computational remark restricted to the exact Hessian case?

3. **On the Hessian moment exponent $q_H$:** The paper carefully specifies $q_g = 2$,
   $> 2, 4$  for global convergence, asymptotic normality, and inference respectively,
   but provides no analogous discussion for $q_H$. What values of $q_H$ are required
   in Theorems 3.8, 4.3, and 4.6? Are these verifiable for regression problems beyond
   those in Chen et al. (2020)?

4. **On experiments:** The experiments are restricted to linear and logistic regression.
   Can the authors provide results on a problem with a less well-conditioned Hessian
   or a non-standard sketching distribution, where the benefit of Nesterov's
   acceleration over unaccelerated sketching in the inference task would be more
   apparent?

5. typo: L199 "middle point" --> "midpoint"

**Strengths And Weaknesses:**

## Soundness

This is overall a strong paper with technically careful and well-organized proofs.
The main analytical challenges are genuine and well-handled. The Cayley–Hamilton
approach to bounding the spectral radius of the asymmetric $2\times 2$ block operator
$G_t(z)$ in Lemma 3.6 — decomposing the accelerated sketching dynamics along
eigendirections of $Z_t$ and reducing to a scalar recursion — is the technical heart
of the paper, and the three-case analysis (distinct real, repeated, complex eigenvalues)
is thorough. The connections to existing results in Proposition 4.4 and the fully
online covariance estimator in Theorem 4.6 round out the contribution cleanly.

However, there are some weaknesses that the authors are encouraged to address during the discussion period:

**W1: On the computational feasibility of the acceleration parameters.** The acceleration
parameters $(\alpha_t, \beta_t, \gamma_t)$ are set via (2.4) as functions of
$(\mu_t, \nu_t)$, defined in (2.5) as the minimum eigenvalue and a second-moment
condition-number quantity of the expected projection matrix
$Z_t$ (see eq. (2.5)), where expectation is taken over the sketching randomness. Computing $\mu_t$ and $\nu_t$
exactly at each outer iteration requires evaluating this expectation in closed form
and then computing $\lambda_{\min}(Z_t)$, which costs $O(d^2)$–$O(d^3)$ per step
depending on the method. This potentially erodes the $O(d^2)$ per-iteration complexity
advantage the paper claims over exact Newton, and represents a genuine added
computational cost of accelerated over unaccelerated sketching that is not discussed
anywhere in the paper.

A natural practical remedy suggested by Lemma 4.1 — which shows
$(\alpha_t, \beta_t, \gamma_t) \xrightarrow{a.s.} (\alpha^\star, \beta^\star,
\gamma^\star)$ as $B_t \to B^\star$ — would be to use fixed limiting parameters
pre-computed once, rather than updating them every iteration. This would eliminate the
per-iteration overhead entirely. However, the paper neither recommends this nor
clarifies whether it is what is actually done in the experiments, leaving a gap between
the theoretical formulation and what is computationally viable in practice.


**W2: On the claim about quasi-Newton and finite-difference extensions.** The paper states
(page 3): *"we suppose the Hessian estimate $H_t$ is accessible; otherwise, it can be
approximated by quasi-Newton or finite-difference schemes, and our analysis extends to
those settings with proper adjustments."* This claim is not substantiated anywhere in
the paper. The analysis in its current form critically relies on the unbiasedness of the
Hessian estimates (Assumption 3.3), which is not guaranteed by quasi-Newton or
finite-difference approximations. This should be toned down to a computational remark
— noting that the method is described for the exact Hessian oracle and that extensions
to approximate schemes would require separate analysis — rather than presented as a
justified claim about the scope of the current results.

**W3: On Assumption 3.3 (Hessian moment condition).** The authors discuss the gradient
moment exponent $q_g$ carefully in relation to each theorem (convergence, normality,
inference require $q_g = 2, > 2, = 4$ respectively), but provide no analogous
discussion for the Hessian exponent $q_H$. The specific values of $q_H$ required by
each theorem are left implicit, even though Assumption 3.3 is comparably strong and
arguably harder to verify in practice than Assumption 3.2. The reference to
"Chen et al. (2020, Appendix A)" for verification covers regression models but does
not discuss what values of $q_H$ are needed or sufficient.


## Presentation

**Introduction clarity.** The introduction can be improved on two fronts. First, the
strong convexity of $f$ is only revealed as an assumption much later (Assumption 3.1),
yet the introduction's discussion of unique minimizers, exponential convergence of
sketching solvers, and statistically efficient estimation all implicitly require it.
Stating strong convexity early in the introduction would make these discussions
significantly easier to follow. Second, when asymptotic normality is first mentioned,
it would be clearer to immediately specify the normalization: the result concerns the
normalized fluctuation $\varphi_t^{-1/2}(x_t - x^\star)$, where $\varphi_t$ is the
vanishing stepsize. Without this, a reader naturally wonders whether the result applies
to a constant stepsize regime or involves a different centering, which creates
unnecessary confusion early in the paper.

Otherwise the paper is well written with a clear structure. The problem decomposition
through filtrations $\mathcal{F}_{t-0.5}$ and $\mathcal{F}_t$ is carefully set up and
consistently maintained.

**Contribution framing.** The paper's main contribution is to extend the last-iterate
asymptotic normality result of Na & Mahoney (2025) and the covariance estimation
framework of Kuang et al. (2025) from the unaccelerated sketch-and-project Newton
solver to the Nesterov-accelerated variant of Derezinski & Rebrova (2024). Algorithmically,
there is no novel method proposed — Algorithm 1 is the existing accelerated
sketch-and-project solver embedded in an online Newton framework. The novelty is
entirely in the analysis, and the technical challenges (asymmetric block operator,
conditionally random acceleration parameters, 2d-dimensional state-co-state dynamics)
are clearly non-trivial and well-articulated. The paper should be read and evaluated
accordingly.

## Significance

The paper makes a meaningful contribution to the growing literature on online inference
for second-order methods. The main finding — that Nesterov's acceleration does not
degrade statistical efficiency relative to unaccelerated sketching, as demonstrated
both theoretically (Proposition 4.4) and empirically (Figure 1) — is of independent
interest. The explicit characterization of $\Sigma^\star$ via a Lyapunov equation that
encodes the sketching operator is new and provides a foundation for understanding
computational-statistical trade-offs in accelerated randomized solvers.


## Originality

The work is a technically involved extension of Na & Mahoney (2025) and Kuang et al.
(2025). The algorithmic setup is not new, but the analysis requires genuinely new tools
(Cayley–Hamilton for asymmetric operators, convergence rates for randomly varying
acceleration parameters $(\alpha_t, \beta_t, \gamma_t)$, fourth-moment bounds for the
online covariance estimator under accelerated sketching). The paper is appropriately
honest about what is and is not novel.

---

> ### Author Rebuttal · Authors · 2026-03-26
>
> **We are pleased the reviewer fully recognizes the complete story of the paper. We hope our responses can further convince the reviewer to support our work.**
> ### 1, Response to W1&Q1
> The reviewer is correct. Let's further clarify theory and practice.
>
> **Theory.** We explain at two levels. **(1)** To achieve an accelerated rate, one has to specify $(\mu_t,\nu_t)$ in (2.5), which may indeed cost $O(d^2)-O(d^3)$. However, as the reviewer said, this is not our fault, nor do we claim the accelerated rate as our contribution. This is the SOTA analysis in the literature. We also think this requirement is quite reasonable, just like many accelerated methods have to correctly specify problem parameters (e.g., Lipschitz constant) to ensure acceleration. Designing a parameter-free solver that preserves acceleration without (2.5) is beyond our scope. **(2)** To establish normality and inference **(our contribution)**, we do not have to specify $(\mu_t,\nu_t)$ as in (2.5). It suffices to have a convergent $(\mu_t,\nu_t)$, which in turn implies convergence of $(\alpha_t,\beta_t,\gamma_t)$; their limits appear in the Lyapunov equation. We proved the optimal setup (2.5) enjoys such convergence, while alternative schemes (e.g., update once after a period or just fix) are also valid and directly applicable to our theory (just that the solver may not accelerate at each step). We will clarify this in Lines 182 and 296 in the final version.
>
> **Practice.** In our experiments (App. C), we tried two ways of setting $(\mu_t,\nu_t)$. (1) Set as theory values in (2.5) with Monte Carlo approximation to the expectation, ensuring the accelerated rate of sketching solver. (2) Set $\mu_t \nu_t=c$ for constant $c=1$, borrowing from Gower et al., 2018, Section 5.1.
>
> As suggested, we have now also tried two other periodic parameter updates: compute (2.5) every 500 or 1000 iterations. The results are reported via an anonymous link in response to **Reviewer EmcX**. The coverage rates of ASN all remain close to 95%, matching our theory.
> ### 2, Response to W2&Q2
> Indeed, we will tone this down by revising Line 138 and rephrasing them as a computational remark. We do not intend to burden the reader with rigorous extensions.
>
> That said, asymptotic normality for finite-difference gradient and Hessian estimates has been established in Spall (2000); Na (2025). While these estimators are biased, the bias is controllable via mesh size. We expect our sketching solver analysis to extend to their exact Newton steps. However, this extension is not trivial, so we will take a conservative stance on this aspect.
> ### 3, Response to W3&Q3
> **(1)** We actually have specified $q_H$: $q_H = 0$ for Theorem 3.8, $q_H = 2$ for Theorem 4.3 (conditions from Lemma 4.2), and $q_H = 4$ for Theorem 4.6 (conditions from Lemma 4.5). We will summarize them in Line 247, as we did for $q_g$.
>
> **(2)** The $q_H$ condition is standard (Na and Mahoney, 2025; Kuang et al., 2025): $q_H = 0$ for global convergence as we only require a descent direction in this phase; $q_H = 2$ for normality as we rely on square-integrable martingale difference arguments to ensure the convergence of Hessian averaging; $q_H = 4$ for covariance estimation as Hessian transforms $g_t$ to $B_t^{-1} g_t$, requiring the same moments as the gradient counterpart.
>
> **(3)** Chen et al., 2020, Lemma 3.1 verifies gradient conditions for linear and logistic regressions. Their analysis in App. A directly extends to Hessian conditions, as the assumptions in their Lemma 3.1 ensure the necessary moment bounds. In particular, $q_H=4$ holds when the design covariate has bounded 8th moment.
>
> **(4)** Beyond Chen et al., 2020, our conditions can be verified for broader model classes, provided the design data have enough moments. For example, in single-index models with link functions having bounded curvature, the same argument as in Chen et al., 2020 applies. In portfolio problems, such as mean-variance or global minimum-variance models (*High-dimensional portfolio selection with cardinality constraints*, Sec. 4.3.1; Du et al., 2023, JASA), as well as PCA-type objectives, Hessian moment conditions also hold.
> ### 4, Response to Q4
> We have prioritized Reviewer QAUF’s request to include SGD, as well as your and Reviewer EmcX’s points on periodic parameter updates to resolve the computation of (2.5), and we will add USN results as desired by Reviewer n436. Our experiments are arguably already as extensive as existing published work (Chen et al., 2020; Zhu et al., 2021; Na and Mahoney, 2025). The advantage of the accelerated over the unaccelerated solver was observed for Gaussian and Kaczmarz sketching in Gower et al., 2018 for regression problems. We further show that the accelerated solver is as statistically efficient as the unaccelerated solver. We think this message is clear from the paper (figures).
> ### 5, Improvement
> We will follow your suggestion and add clarifications in Lines 9 and 42 to clear things up front.

---

### Official Review · Reviewer_n436 · 2026-03-14

**Soundness:** 2
**Presentation:** 3
**Significance:** 2
**Originality:** 3
**Overall Recommendation:** 4
**Confidence:** 4

**Summary:**

The paper studies inference of Newtons method when the newton step is estimated by accelerated sketching. Consider the Newton method, where $x_t = x_{t-1} + \psi_t \Delta x_t,$ where $\delta x_t \in R^d$ solves $B \Delta x_t = g_t$ where $g$ and $B$ are the hessian and the gradient of the loss function. Since solving the linear equation can be expensive O(d^3), recent research has proposed using an sketching based inner loop to approximately solve the linear system in O(d^2) cost. Na and Mahonry (2025) have  established the the asymptotic property of the unaccelerated sketching. This paper extends the result to the case of accelerated sketching. The paper compares the asymptotic properties of (1) exact newton, (2) unaccelerated sketched newton and (3) sketching with nestrov acceleation for newton step estimation (the inner-loop). The paper shows that the accelerated sketching is consistent and derive its asymptotic variance and also provide an cheap empirical estimator for it, making the proposal practical.

**Compliance With Llm Reviewing Policy:**

Affirmed.

**Final Justification:**

The authors helped me understand the paper contribution better, hence the updated rating.

**Key Questions For Authors:**

See my comments 1-4 in the previous section.

**Limitations:**

yes

**Strengths And Weaknesses:**

The paper studies an interesting and relevant problem: inference for Newton-type methods when the Newton step is computed approximately via accelerated sketching. Extending prior asymptotic inference results from unaccelerated to accelerated sketching is **potentially** valuable, especially if it leads to a better computational-statistical tradeoff in large dimensions. But the paper stops well short of establishing that. The theoretical results are interesting, but the paper does not yet make a convincing case that the gains of the accelerated estimator are large enough to justify the study. The paper asks "How does accelerated sketching affect statistical efficiencyof inference in online sketched Newton methods?" But **doesn't** really answer it. I also think Proposition 4.4 is a useful step toward comparing the inferential behavior of exact, unaccelerated, and accelerated methods.

1. The paper does not provide a direct table comparing accelerated and unaccelerated sketching. That comparison is mostly figure-based (QQ plots), while the tables mainly compare the proposed method against exact Newton. A direct table with runtime and inference metrics would make the benefit of acceleration much clearer. You can add the unaccelerated and exact newton to the table 1. Can you also please add the computation time to the table 1? Currently the computational savings are claimed (and can be seen from the theory) but not established in empirical studies.

2. The paper would be clearer, punchier, and better motivated if the authors gave a more direct comparison of the three limiting covariance induced by exact Newton, unaccelerated sketching, and accelerated sketching. proposition 4.4 starts to move in this direction, but it does not really go all the way, and I would like to see that comparison developed more fully. If this is hard to do in full generality, then showing it clearly in the empirical setups would already be very helpful. Also, right now, neither the abstract nor the conclusion clearly says how these three methods differ in their limiting variances, which leaves the main comparison feeling too vague to be very informative.

3. In table 1, for Toeplitz and Equi-corr, the results seem a little weird that $\tau = \infty$ the MAE is higher but the ave coverage is very high with much smaller ave len. I would have thought if the MAE is higher, to maintain good coverage the ave len has to be larger. Can you help me understand what I am missing? Especially since the asymptotic distribution is Gaussian.

4. I think this is a strong motivation for studying accelerated sketching: *while Kuang et al. (2025) employs at least 10 sketching steps to achieve competitive performance in large dimensions, our method attains comparable or even superior results with at most 5 sketching steps.* However, the current draft does not provide direct evidence for this claim. Some supporting empirical evidence would help convince me that this is a worthwhile estimator to study.

---

> ### Author Rebuttal · Authors · 2026-03-26
>
> **Thank you for careful reading and constructive comments. All your questions are valuable and can be addressed properly; we will revise the paper accordingly to reflect our responses. That said, we sincerely hope the reviewer can *re-evaluate* the paper again based on our responses, its clear significance, and technically highly involved, novel analysis already presented, rather than penalizing it for not performing prior work's experiments as extensively as expected. Your key question of comp-stats trade-off is answered explicitly in our response to Reviewer QAUF.**
> ### 1, Response to W2&Q1&Q4
> **First**, we will add results of **USN** to Tables 1 and 2. However, in the rebuttal phase, we have prioritized the common points of **Reviewers hYL1 & EmcX** to reduce the computation of **ASN** by using periodic parameter updates. This is because experiments on USN are standard and have already been extensively studied in prior work under similar settings, while periodic parameter updates are enabled only by our new theory.
>
> That said, please note that our current experiments are already comparable in scope to existing published work in this area, and the insights from results of USN are reflected by what is already shown and discussed in the presented tables and figures **(i.e., acceleration does not hurt statistical efficiency compared to unacceleration (figures), and exhibits a clear comp-stats trade-off compared to EN (tables))**. More results on existing USN, not the paper's focus anyway, do not change the story.
>
> **Second**, we want to calibrate the paper scope with the reviewer to avoid overreach. As nicely summarized by Reviewers hYL1 & EmcX, **we do not propose any new estimators.** The accelerated Newton estimator already exists in the literature; our contribution is to analyze its statistical efficiency and to connect and compare it with unaccelerated estimators. We are therefore not obligated to justify its computational superiority and *"convince you this is a worthwhile estimator" as requested* by exploring, e.g., reduced running time, faster convergence, and robustness to parameters, as this is the job already been done in Gower et al., 2018, Section 5 for linear and logistic regressions. Having them is great, but not required in our study, we believe. Devaluing our contributions of a new analysis of the accelerated solver, new and comprehensive results on statistical efficiency, and new connections to existing efficiency results, simply because not emphasizing computation as Gower et al. (2018), is unfair!
>
> We indeed concern comp-stats trade-offs, but computational gains of **ASN** have already been analyzed theoretically and shown empirically in prior work (we do not claim contributions either). Our job is to understand the statistics side and know how to use it for online inference, which we have explicitly clarified in response to Reviewer QAUF due to space limits. **We respectfully hope the reviewer not devalue our work on this basis.** See Reviewer hYL1 comments as well.
>
> **Third**, following this discussion, we can complement Tables 1 and 2 with **USN**, but we will not compare running time. The latter is very misleading: for a fixed $\tau$, the unaccelerated solver involves fewer operations and thus shorter runtime than the accelerated solver.
> ### 2, Response to W1&W3&Q2
> Please refer to our response to Reviewer QAUF for a detailed comparison of the three covariances $\Sigma_{asn}^\star, \Sigma_{usn}^\star, \Omega^\star$. This will be clarified in the final version rigorously with one extra page allowed (though we strongly believe the present paper is already quite technically involved and novel). We appreciate the reviewer’s recognition that directly comparing $\Sigma_{asn}^\star, \Sigma_{usn}^\star$ in full generality is challenging. Except for this direct comparison, we have addressed all missing questions, supported by empirical evidence (Figures 1, 2, 3). In short, **accelerated sketching improves computational efficiency without sacrificing statistical efficiency (in the sense of upper-bound comparison)**.
> ### 3, Response to Q3
> MAE measures the iterate error, while AveLen measures the variability; they capture different aspects and are not consequences of one another. (1) EN exhibits a higher MAE than ASN in these cases, but the difference is not substantial considering the problem dimension. Moreover, there is no theory suggesting EN should have a lower MAE than ASN. (2) EN yields shorter AveLen, as predicted by our theory: the variability of EN (that AveLen precisely measures) is smaller than that of ASN, i.e., $\Sigma_{asn}^\star \succeq \Omega^\star$ (comp-stats trade-off). (3) EN achieves a high coverage rate despite higher MAE and shorter CIs, simply because its CIs, while shorter than those of ASN, are still sufficiently wide to cover the true solution. In contrast, ASN has a smaller MAE, but due to its higher variability, its CIs must be longer to achieve comparable coverage.

---

> > ### Author Rebuttal · Reviewer_n436 · 2026-04-03
> >
> > I have increased my rating thanks for the clarification.

---

> > > ### Author Response · Authors · 2026-04-03
> > >
> > > We sincerely appreciate your positive assessment and support of our work. We are grateful for your careful review and constructive suggestions, which have helped improve both the clarity and positioning of the paper. Thank you.

---

### Official Review · Reviewer_QAUF · 2026-03-18

**Soundness:** 3
**Presentation:** 3
**Significance:** 3
**Originality:** 3
**Overall Recommendation:** 4
**Confidence:** 3

**Summary:**

This paper proposes a second-order method for stochastic optimization based on an online Newton scheme with Hessian averaging. The proposed method achieves the same complexity order as first-order methods. The authors establish global almost sure convergence and prove the asymptotic normality of the last iterate.

**Compliance With Llm Reviewing Policy:**

Affirmed.

**Final Justification:**

My concerns have been largely addressed, and I will maintain my current score.

**Key Questions For Authors:**

1. What are the empirical and theoretical convergence rates of the proposed method?

2. What are the computational–statistical trade-offs associated with the proposed method?

3. Please include numerical comparisons with standard benchmarks.

**Limitations:**

yes

**Strengths And Weaknesses:**

## Strengths

The paper is clearly written. The proposed method is computationally efficient, achieving the same complexity as first-order methods. The authors establish global almost-sure convergence and derive asymptotic normality.


## Weaknesses

The numerical evaluation is too limited. In particular, the paper lacks numerical comparisons with standard benchmarks such as SGD. Since the proposed approach is a second-order method, one would also expect a formal analysis of its superlinear convergence rate, but this is not provided. In addition, the claimed computational–statistical trade-offs are not clearly clarified.

---

> ### Author Rebuttal · Authors · 2026-03-25
>
> **Thank you for your careful reading and positive opinion of the paper. All your questions are reasonable and can be addressed properly, so we hope you can support the paper more affirmatively based on our responses.**
> ### 1, Response to W2&Q1
> **Theoretical rate.**  Theorem 3.8 shows global convergence $x_t \rightarrow x^\star$; Theorem 4.3 shows $1/\sqrt{\varphi_t}(x_t - x^\star) \rightarrow N(0, \Sigma^\star)$, which implies the convergence rate $\mathbb{E}[\|\|x_t - x^\star\|\|^2] = O(\varphi_t)$; Lemma 4.5 strengthens this to $\mathbb{E}[\|\|x_t - x^\star\|\|^4] = O(\varphi_t^2)$ when gradient and Hessian estimates have finite 4th moment. Additional convergence rates for sketching parameters and the covariance matrix estimator are also established in Lemma 4.2 and Theorem 4.6.
>
> **Empirical rate.**  We have applied our method to construct valid confidence intervals in experiments following the rule in Line 365; the empirical coverage rate is close to the nominal 95\%. This immediately suggests that the iterate error indeed achieves $O_p(\sqrt{\varphi_t})$ in practice, matching the theory (if the rate is wrong, our CIs cannot be valid and close to 95%).
>
> **Note.**  We emphasize that we study an online Newton method with a decaying stepsize. **In contrast to deterministic Newton methods with local unit stepsize, superlinear convergence should not be expected in this setting.** The sublinear rate $O_p(\sqrt{\varphi_t})$ is the proper benchmark in this regime ($\varphi_t = 1/t$ yields the optimal $\sqrt{t}$-consistency). See also the presentation suggestion of **Reviewer hYL1**. We will add clarification in Line 43 to avoid this confusion in the final version.
> ### 2, Response to W3&Q2
> We will revise Line 89 and add additional results after Line 372 to make comp-stats trade-offs more explicit. Existing literature has compared unaccelerated sketched Newton (USN) and exact Newton (EN) to illustrate this trade-off (Na & Mahoney, 2025; Kuang et al., 2025): USN gains computational efficiency due to sketching, but sacrifices statistical efficiency as its limiting covariance satisfies $\Sigma^\star_{usn}\succeq \Omega^\star$; in contrast, EN attains optimal covariance $\Omega^\star$ (with $\varphi_t = 1/t$). This paper studies accelerated sketched Newton (ASN), which leads to two additional comparisons. The following explanations have either been discussed in the paper (can be re-emphasized for clarity) or can be rigorously complemented in the final version.
>
> **ASN v.s. EN.** The comp-stats trade-off precisely follows that of **USN v.s. EN**. ASN gains computational efficiency while sacrificing statistical efficiency with $\Sigma_{asn}^\star\succeq \Omega^\star$.
>
> **ASN v.s. USN.** ASN gains computational efficiency since Nesterov acceleration improves the convergence rate of the sketching solver from $\rho_{usn} = 1-\mu$ to $\rho_{asn}=1-\sqrt{\mu/\nu}$ (Line 177). This faster rate allows the solver to attain comparable accuracy with fewer sketching steps (see Section 5 in Gower et al., 2018). For statistical efficiency, although we cannot directly compare $\Sigma_{asn}^\star$ and $\Sigma_{usn}^\star$ (which remains a challenging open problem), **we can use the optimal covariance $\Omega^\star$ as a baseline and show that $\|\|\Sigma_{asn}^\star-\Omega^\star\|\| =O(\rho_{asn}^\tau)$, which is smaller than $\|\|\Sigma_{usn}^\star-\Omega^\star\|\| =O(\rho_{usn}^\tau)$ as proved in Na & Mahoney, 2025** (since $\rho_{asn}\leq \rho_{usn}$). In other words, the statistical efficiency gap is controlled by the approximation error of the sketching solver. Although we compare two upper bounds of efficiency gaps, this result is still promising and sheds light on a future direct comparison.
>
> **Summary.** Comp efficiency: ASN $>$ USN $>$ EN; Stats efficiency: EN $>$ ASN "$>$" USN, where the last "$>$" holds in the sense of upper bound comparison.
> ### 3, Response to W1&Q3
> We will follow the suggestion and expand Tables 1 and 2 to include SGD results. See our response to **Reviewer EmcX** for an anonymous link providing these results.
>
> That said, the current experiments are already at least as extensive as those in the existing literature (Na and Mahoney, 2025; Kuang et al., 2025). **In particular, we compare ASN with EN in two big tables and ASN with USN in three figures, and implement different sketching distributions and parameter configurations.** We should mention that adding SGD experiments provides limited additional information, as the under-coverage behavior of SGD online inference is already well documented in prior work (Zhu et al., 2021, Table 1), (Kuang et al., 2025, Table 1). Specifically, averaged SGD with a batch-means covariance estimator converges slowly and exhibits under-coverage even for d= 20, which is also a motivation for our focus on second-order methods.
>
> Nevertheless, we have conducted SGD experiments, and the results confirm the same qualitative under-coverage behavior reported in prior literature.

---

> > ### Author Rebuttal · Reviewer_QAUF · 2026-04-02
> >
> > Thank you for your response. My question regarding convergence was about the algorithmic convergence rate, namely whether the proposed method achieves sublinear, linear, or superlinear convergence per iteration. The results you cited (Theorems 3.8, 4.3, and Lemma 4.5) describe global convergence and asymptotic error rates, but they do not directly clarify the optimization-theoretic rate at which $||x_t - x^\star||$ decreases, nor how this compares with relevant Newton-type baselines. I would therefore appreciate a clearer statement of the per-iteration convergence behavior, together with empirical convergence plots if available.
> >
> > Regarding the computational-statistical trade-off, the current response mainly provides qualitative comparisons among ASN, USN, and EN, but does not make the trade-off itself explicit or quantitative. It would be helpful if the authors could state more clearly what statistical efficiency is lost in exchange for what computational savings, preferably in a form that can be directly interpreted as a concrete trade-off.

---

> > > ### Author Response · Authors · 2026-04-03
> > >
> > > **Thank you for follow-up questions. We clarify the points below.**
> > > ## 1, Convergence rate
> > > In the following discussion, we focus only on stochastic methods with **vanishing stepsize**.
> > >
> > > **(1) Our study.** This paper studies the inference of online sketched Newton methods. All our results closely follow the literature on online inference of SGD, Newton, and their variants (Ruppert, 1988; Polyak and Juditsky, 1992; Chen et al., 2020; Zhu et al., 2021; Duchi and Ruan, 2021; Davis et al., 2024; Jiang et al., 2025; Na and Mahoney, 2025; Kuang et al., 2025). In fact, these papers typically establish either normality or covariance estimation, while we have established both.
> > >
> > > **(2) Rate is asymptotic?** Although an asymptotic rate like $\Delta_t:=x_t-x^\star=O_p(\sqrt{\varphi_t})$ suffices for our inference study, **our stated rates are actually non-asymptotic!** Lemma 4.5 shows there exists $C>0$ such that $\mathbb{E}[\|\|\Delta_t\|\|^4]\leq C\varphi_t^2$ for all $t>0$ (implying $\mathbb{E}[\|\|\Delta_t\|\|^2]\leq C\varphi_t$). These are called $R$-sublinear rates. These non-asymptotic rates in the paper naturally imply asymptotic $O_p(\sqrt{\varphi_t})$ rate for inference, since convergence in $L_2$ implies in probability. Thus, we have indeed already established non-asymptotic rates one can reasonably expect for this method.
> > >
> > > **(3) Per-iteration rate.** The recursion of $\Delta_t$ is of form (B.19): $\Delta_{t+1} = (I - \varphi_t \mathcal{R})\Delta_t + \varphi_t \theta_t + \varphi_t \delta_t$, where $\mathcal{R}$ is a Hessian-related matrix, $\theta_t$ is a martingale difference, and $\delta_t$ collects higher-order terms. Due to this noisy recursion, one cannot control $\Delta_{t+1}$ by $\Delta_t$ alone and establish the $Q$-type (sub/super)linear rate the reviewer expects. Indeed, $\|\|\Delta_t\|\|$ (and $\mathbb{E}[\|\|\Delta_t\|\|]$) is not expected to decrease monotonically, even for SGD. By squaring and taking expectations on both sides (Lines 1375--1383), we obtain the $R$-type sublinear rate from the error recursion, which is standard in our context. See a note (p.30) https://www.cs.ubc.ca/~schmidtm/Courses/540-W19/L11.pdf.
> > >
> > > **(4) Newton-baselines?** See p.19 in the note above. The vanishing stepsize fully determines the convergence rate (i.e., $R$-sublinear rate $\varphi_t$), while preconditioning does not improve/impair it. There is therefore NO Newton-baseline rate to distinguish from SGD (under vanishing stepsize).
> > >
> > > **(5) Empirical results.** We plot the iterate error of a single run and the averaged iterate error in an anonymous link [Figure](https://anonymous.4open.science/r/Testttttttt-92B485L7/Figure1.png). As expected, both errors fluctuate so that no $Q$-type rate is available. The averaged error decays sublinearly in $\sqrt{\varphi_t}$, consistent with our $R$-type sublinear rate.
> > > ## 2, Comp-stats trade-offs
> > > **All > in our explanation are supported by quantitative evidence.** Overall, ASN, USN, and EN exploit different Newton directions (sketched vs. exact), which precisely locate where the trade-off occurs. By trade-off, we mean **one method may apply a computationally cheaper direction than the other, but this direction may also lead the method to a less statistically efficient estimator than the other.**
> > >
> > > **Comp eff: ASN>USN>EN.** The accelerated Newton system solver has a linear rate $\rho_{asn}=1-\sqrt{\mu/\nu}$ faster than $\rho_{usn}=1-\mu$. Thus, to have approximation error $\|\|\Delta x_{t,asn}-\Delta x_t\|\| = O(\rho_{asn}^\tau)\leq\epsilon$, ASN requires fewer sketching steps than USN, $\tau_{asn}\leq\tau_{usn}$. Since the per-iteration cost of ASN and USN is comparable, the total computational cost satisfies ASN<USN. For a more quantitative comparison, consider $s=1$ with Kaczmarz sketching: as discussed in (Gower et al., 2018, Sec. 3.4 (17)) and (Na and Mahoney, 2025, Remark 4.5), both methods have per-iteration cost $O(d)$ and require $\tau = O(d)$ iterations, leading to an overall cost of $O(d^2)$, with ASN exhibiting better Hessian condition number dependence in the constant of $O(\cdot)$ (with case-by-case improvement). By contrast, EN incurs $O(d^3)$ cost to solve for exact $\Delta x_t$.
> > >
> > > **Stats eff: EN>ASN>USN.** Stats eff is measured by asymptotic variance (smaller is better). By Duchi and Ruan (2021); Davis et al. (2024), the minimax optimal variance among all possible estimators is $\Omega^\star$, which EN attains. Due to sketched directions, $\Sigma_{usn},\Sigma_{asn}\succeq\Omega^\star$, so both are less statistically efficient than EN. This loss is quantitatively attributed to the sketching solver randomness (Prop 4.4) characterized by the Lyapunov equation (4.2). Further, the efficiency gap $\|\|\Sigma_{usn}-\Omega^\star\|\| = O(\rho_{usn}^\tau)$ is controlled by the approximation error of the sketching solver (Na and Mahoney, 2025); since $\rho_{asn}\leq\rho_{usn}$, the efficiency gap of ASN is smaller than USN, yielding ASN>USN. Everything here is quantitative.

---

### Decision · Program_Chairs · 2026-04-30

**Decision:**

Accept (regular)

**Comment:**

This paper studies an instance of online Newton method where the Newton step is approximately implemented using an accelerated sketch-and-project method. The reviewers are all in agreement that the work is technically strong and well-written, and that it provides important contributions in extending earlier last-iterate normality results and covariance estimation frameworks to an accelerated setting. Hence, it is recommended that this work be accepted to the conference.